# Antarctic temperature and $CO_2$: near-synchrony yet variable phasing during the last deglaciation

Jai Chowdhry Beeman[a,1], Léa Gest[a,1], Frédéric Parrenin[a], Dominique Raynaud[a], Tyler J. Fudge[b], Christo Buizert[c], and Edward J. Brook[c]

[a]Univ. Grenoble Alpes, CNRS, IRD, IGE, F-38000 Grenoble, France
[b]Department of Earth and Space Sciences, University of Washington, Seattle, WA 98195, USA
[c]College of Earth, Ocean, and Atmospheric Sciences, Oregon State University, Corvallis, OR 97331, USA
[1]Jai Chowdhry Beeman and Léa Gest contributed equally to this work.

*Correspondence to:* Frédéric Parrenin frederic.parrenin@univ-grenoble-alpes.fr

**Abstract.** The last deglaciation, which occurred from 18,000 to 11,000 years ago, is the most recent large natural climatic variation of global extent. With accurately dated paleoclimate records, we can investigate the timings of related variables in the climate system during this major transition. Here, we use an accurate relative chronology to compare temperature proxy data and global atmospheric $CO_2$ as recorded in Antarctic ice cores. In addition to five regional records, we compare a $\delta^{18}O$ stack, representing Antarctic climate variations, with the high-resolution, robustly dated WAIS Divide $CO_2$ record. We assess the $CO_2$ / Antarctic temperature phase relationship using a stochastic method to accurately identify the probable timings of changes in their trends. Four coherent changes are identified for the two series, and synchrony between $CO_2$ and temperature is within the 95% uncertainty range for all of the changes except the end of Termination 1. During the onset of the last deglaciation at 18 ka BP and the deglaciation end at 11.5 ka BP, Antarctic temperature most likely led $CO_2$ by several centuries (by 570 years, within a range of 127 to 751 years, 68% probability, at the T1 onset; and by 532 years, within a range of 337 to 629 years, 68% probability, at the deglaciation end). At 14.4 ka BP, the onset of the Antarctic Cold Reversal (ACR) period, our results do not show a clear lead or lag (Antarctic temperatre leads by 50 years, within a range of -137 to 376 years, 68% probability). The same is true at the end of the ACR ($CO_2$ leads by 65 years, within a range of 211 to 117 years, 68% probability). However, the timings of changes in trends for the individual proxy records show variations from the stack, indicating regional differences in the pattern of temperature change, particularly in the WAIS Divide record at the onset of the deglaciation; the Dome Fuji record at the deglaciation end; and the EDML record after 16 ka BP. In addition, two significant changes, one at 16 ka BP in the $CO_2$ record and one after the ACR onset in three of the isotopic temperature records, do not have significant counterparts in the other record. The likely-variable phasings we identify testify to the complex nature of the mechanisms driving the carbon cycle and Antarctic temperature during the deglaciation.

## 1   Introduction

Glacial-interglacial transitions, or deglaciations, mark the paleorecord approximately every 100,000 years over the past million years or so (Jouzel et al., 2007; Lisiecki and Raymo, 2005; Williams et al., 1997). The last deglaciation, often referred to as

glacial termination 1 (T1), offers a case study for a large global climatic change, very likely in the 3 - 8°C range on the regional scale (Masson-Delmotte et al., 2013), and thought to be initiated by an orbitally driven insolation forcing (Berger, 1978; Hays et al., 1976; Kawamura et al., 2007). The canonical interpretation of this apparent puzzle is that insolation acts as a pacemaker of climatic cycles and the amplitude of glacial - interglacial transitions is mainly driven by two strong climatic feedbacks:

atmospheric $CO_2$ and continental ice surface - albedo changes. However, the mechanisms that control the $CO_2$ rise are still a matter of debate. Accordingly, reconstructing the phase relationship (leads and lags) between climate variables and $CO_2$ during the last termination has become of importance, and has a substantial history in ice core research (Barnola et al., 1991; Caillon et al., 2003; Parrenin et al., 2013; Pedro et al., 2012; Raynaud and Siegenthaler, 1993).

Global temperature has been shown to lag $CO_2$ on average during T1 (Shakun et al., 2012), supporting the importance of

$CO_2$ as an amplifier of orbitally-driven global-scale warming. But Antarctic temperature and $CO_2$ concentrations changed much more coherently as T1 progressed. Indeed, midway through the glacial-interglacial transition, Antarctic warming and $CO_2$ increase slowed and even reversed during a period of about 2000 years, coinciding with a warm period in the North called the Bølling–Allerød (B/A). The respective period of cooling in Antarctica is called the Antarctic Cold Reversal (ACR). A period of cooling in the Northern Hemisphere known as the Younger Dryas (YD), followed the B/A, coinciding with a period

of warming in the SH.

High-latitude Southern Hemisphere paleotemperature series–including Southern Ocean temperature–varied similarly to Antarctic temperature during T1 (Shakun et al., 2012; Pedro et al., 2016). Upwelling from the Southern Ocean is thought to have played an important role in the deglacial $CO_2$ increases (Anderson et al., 2009; Burke and Robinson, 2012; Rae et al., 2018; Schmitt et al., 2012). The Atlantic Meridional Overturning Current, or AMOC, a major conduit of heat between

the Northern and Southern Hemispheres and component of the bipolar seesaw, the umbrella term encompassing the mechanisms thought to control the seemingly alternating variations of Northern and Southern Hemisphere temperature (Stocker and Johnsen, 2003; Pedro et al., 2018), is thought to have influenced Southern Ocean upwelling during the deglaciation (Anderson et al., 2009; Skinner et al., 2010). A weakening of the oceanic biological carbon pump appears to have dominated the deglacial $CO_2$ increase until 15.5 ka BP, when rising ocean temperature likely began to play a role as well (Bauska et al., 2016).

Ice sheets are exceptional archives of past climates and atmospheric composition. Local temperature is recorded in the isotopic composition of snow/ice (Jouzel et al., 2007; NorthGRIP Project Members, 2004) due to the temperature dependent fractionation of water isotopes (Lorius and Merlivat, 1975; Johnsen et al., 1989). The concentration of continental dust in ice sheets is a proxy of continental aridity, atmospheric transport intensity and precipitation. Finally, air bubbles enclosed in ice sheets are near-direct samples of the past atmosphere. However, the age of the air bubbles is younger than the age of the

surrounding ice, since air is locked in at the base of the firn (on the order of 70 m below the surface on the West Antarctic Ice Sheet (WAIS) Divide) at the Lock-In Depth (LID) (Buizert and Severinghaus, 2016). The firn, from top to bottom, is composed of a convective zone (CZ) where the air is mixed vigorously, and a diffusive zone (DZ) where molecular diffusion dominates transport. Firn densification models can be used to estimate the LID and the corresponding age difference (Sowers et al., 1992).

Atmospheric $CO_2$ concentrations, recorded in the air bubbles enclosed in ice sheets, are better preserved in Antarctic ice

than in Greenland ice, because the latter has much higher concentrations of organic material and carbonate dust (Raynaud

et al., 1993; Anklin et al., 1995). Measured on the Vostok and EPICA Dome C ice cores, the long-term history of $CO_2$ (Lüthi et al., 2008) covers the last 800 ka.

Early studies suggested that at the initiation of the termination around 18 ka BP (kiloyears before 1950 A.D.), just after the Last Glacial Maximum (LGM), Antarctic temperature started to warm $800 \pm 600$ yr before $CO_2$ began to increase (Monnin et al., 2001), a result that was sometimes misinterpreted to mean that $CO_2$ was not an important amplification factor of the deglacial temperature increase. This study used measurements from the EPICA Dome C (EDC) ice core (Jouzel et al., 2007) and a firn densification model to determine the air chronology. However, this firn densification model was later shown to be in error by several centuries for low accumulation sites such as EDC during glacial periods (Loulergue et al., 2007; Parrenin et al., 2012).

Two more recent works (Pedro et al., 2012; Parrenin et al., 2013), used stacked temperature records and improved estimates of the age difference between ice and air records to more accurately estimate the relative timing of changes in Antarctic temperature and atmospheric $CO_2$ concentration. In the first of these studies, measurements from the higher accumulation ice cores at Siple Dome and Byrd Station were used to decrease the uncertainty in the ice-air age shift, and indicated that $CO_2$ lagged Antarctic temperature by 0-400 yr on average during the last deglaciation (Pedro et al., 2012). The second study (Parrenin et al., 2013) used measurements from the low accumulation EDC ice core but circumvented the use of firn densification models by using the nitrogen isotope ratio $\delta^{15}N$ of $N_2$ as a proxy of the DZ height, assuming that the height of the CZ was negligible during the study period. $CO_2$ and Antarctic temperature were found to be in phase at the beginning of TI ($-10 \pm 160$ years) and at the end of the ACR period ($-60 \pm 120$ years), but $CO_2$ was found to lag Antarctic temperature by several centuries at the beginning of the Antarctic Cold Reversal ($260 \pm 130$ years) and at the end of the deglacial warming in Antarctica ($500 \pm 90$ years). The end of the deglacial warming in Antarctica occured roughly two centuries after the onset of the Holocene period, dated at 11.7 ka BP according to the International Commission on Stratigraphy. However, the assumption that $\delta^{15}N$ reflects DZ height is imperfect, as it may underestimate the DZ height for sites with strong barometric pumping and layering (Buizert and Severinghaus, 2016).

A new $CO_2$ record of unprecedented high resolution (Marcott et al., 2014) from the WAIS Divide (WD) ice core merits the reopening of this investigation. The air chronology of WAIS Divide is well constrained thanks to a relatively high accumulation rate and to accurate nitrogen-15 measurements (Buizert et al., 2015). The WAIS record evidences centennial-scale changes in the global carbon cycle during the last deglaciation superimposed on more gradual, millenial-scale trends that bear resemblance to Antarctic temperature (Marcott et al., 2014).

The deglacial temperature rise seen at WD is structurally similar to that at other Antarctic sites. However, West Antarctic warming may have been greater in magnitude than East Antarctic warming by up to 3 degrees, and the rise in West Antarctic temperature shows an early warming starting around 21 ka BP, following local insolation (Cuffey et al., 2016). This early warming trend is much more gradual in records from East Antarctic ice cores. The difference between the two records may be related to sea ice conditions around East and West Antarctica, and perhaps to elevation changes (Cuffey et al., 2016; WAIS Divide Project Members, 2013). The temperature record at WAIS Divide shows an acceleration in warming around 18 ka BP which is also present in East Antarctic records (WAIS Divide Project Members, 2013).

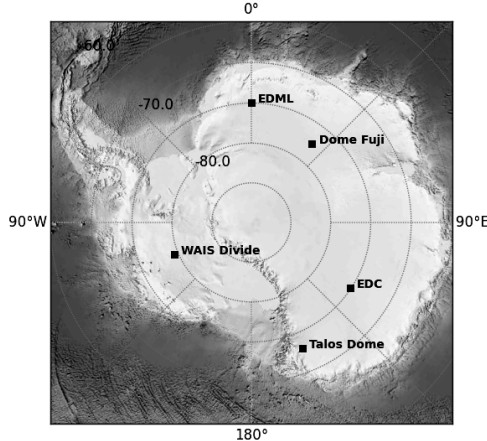

**Figure 1.** Drilling locations of the ice cores from which the $CO_2$ and isotopic paleotemperature records included in this study were measured.

On the much shorter timescales of the satellite era, Jones et al. (2016) note differing temperature trends at the drilling sites of the five cores used in this study. On the other hand, the interpretation of individual isotopic records can prove complicated, as local effects, including those of ice sheet elevation change and sea ice extent, are difficult to correct.

In the present work we refine our knowledge of leads and lags between Antarctic temperature and $CO_2$. We use a stack of accurately synchronized Antarctic temperature records (Buizert et al., 2018) to reduce local signals, placed using volcanic matching on the WAIS Divide chronology (WD2014). We then compare the temperature stack to the high resolution WAIS Divide $CO_2$ record by determining the probable timings of changes in trend, and calculate probable change point timings for the five individual isotope-derived records used in the stack as well.

## 2 Methods and data

### 2.1 Temperature stack and ice chronology

We use the $\delta^{18}O$ stack developed by Buizert et al. (2018) (referred to hereafter as Antarctic Temperature Stack 3, or ATS3) to represent Antarctic Temperature. The use of the stack allows us to remove local influences and noise in the individual records to the greatest extent possible. The stack contains five records: EDC, Dome Fuji (DF), Talos Dome (TALDICE), EPICA Dronning Maud Land (EDML) and WAIS Divide (WD). Volcanic ties between WD and EDC, WD and TD, and WD and EDML are developed in Buizert et al. (2018); previously published volcanic ties were used between EDC and DF (Fujita et al., 2015), placing all of the records on the WD2014 chronology (Buizert et al., 2015). Notably, the Vostok record, included in the stack used by Parrenin et al. (2013) is excluded from the Buizert et al. (2018) stack: it contains additional chronological uncertainty as it is derived using records from two drilling sites. We take the quadratic sum of the synchronization error from Buizert et al.

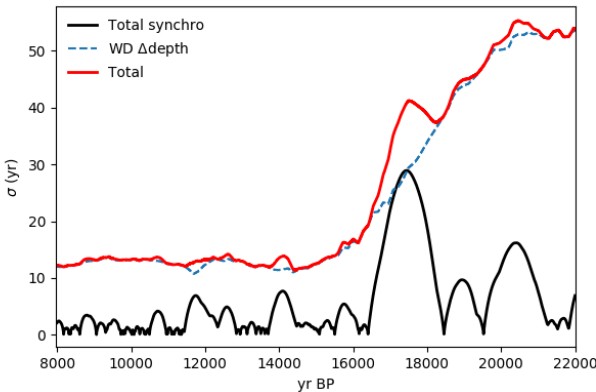

**Figure 2.** Relative chronological uncertainty between ATS3 and the WD $CO_2$ record (red line), calculated as the quadratic sum of the synchronization error from Buizert et al. (2018) (black line) and the $\Delta$ age uncertainty from WD2014 (blue dashed line).

.

(2018) and the $\Delta$ age uncertainty from WD2014 to calculate the relative chronological error between ATS3 and the WD $CO_2$ record.

## 2.2 $CO_2$ and air chronology

We use atmospheric $CO_2$ data from the WD ice core (Marcott et al., 2014) which consist of 1,030 measurements at 320 depths that correspond to ages between 23,000 and 9,000 years B1950 with a median resolution of 25 years. At WD, the age offset between the ice and air (trapped much later) at a given depth, $\Delta$age, is calculated using a firn densification model, which is constrained using nitrogen-15 data, a proxy for firn column thickness (Buizert et al., 2015). $\Delta$age ranges from 500±100 yr at the last glacial maximum, to 200±30 yr during the Holocene. $\Delta$age uncertainty is added to cumulative layer counting uncertainty to determine the total uncertainty of the air chronology.

## 2.3 Identifying changes in trend

We identify likely change points using piecewise linear functions. Residuals are calculated between the raw data and linear functions with a fixed number of stochastically proposed change points, which are free to explore the entire temporal range of the time series (similarly to Parrenin et al. (2013)). These residuals are summed to form a cost function, which allows us to perform a Bayesian analysis of the probable timing of change points. At the base of our method is a parallelized Metropolis-Hastings (MH) procedure (Goodman and Weare, 2010; Foreman-Mackey et al., 2013). Therefore, we do not present a single "best fit" but rather analyze the ensemble of fits accepted by the routine. We plot two histograms: an upward-oriented histogram for concave-up change points, and a downward-oriented histogram for concave-down change points. We use these histograms as probabilistic locators of changes in slope (Figure 3).

The change point representations of the ATS3 and $CO_2$ time series are composed of a set of $n$ specified change points $\{X_i = (x_i, y_i) |\ i = 1, \ldots, n\}$. We denote the vector of $m$ time series observations $o$ at time $t$ $\{O_l = (t_l, o_l) |\ l = 1, \ldots, m\}$, and the scalar residual term $J$ between observations and the linear interpolation between change points $f_y$:

$$J(X_i) = R^T C^{-1} R;\ r_l = (f_y(t_l) - o_l)_l \tag{1}$$

where $R$ is the vector of residuals at each data point with components $r_l$ and $C$ is the covariance matrix of the residuals. The ATS3 series contains 1412 data points, and the WD $CO_2$ series contains 320, each of which is considered in the residuals.

We fix $x_0 = t_0$ and $x_n = t_l$; i.e. the x-values of the first and last change points are fixed to the first and last x-values of the observation vector, with the y-values allowed to vary. The remaining points are allowed to vary freely in both dimensions.

### 2.3.1    Estimating the covariance matrix C: treating uncertainty and noise

Our method fits time series with piecewise linear functions, and the residual vector thus accounts for any variability that cannot be represented by these fits. Paleoclimate time series, like the $CO_2$ and ATS3 series used here, typically contain autocorrelated noise (see Mudelsee (2002), for example) which cannot be accurately represented by a piecewise linear function. Weighting the residuals of a cost-function based formulation by a properly estimated inverse covariance matrix ensures that this autocorrelated noise is not overfitted, and can improve the balance of precision and accuracy of the fits.

Our time series contain two potential sources of uncertainty: measurement or observational uncertainty, related with the creation of the data series, and modeling uncertainty, related to the formulation of the fitting function. We formulate a separate covariance matrix to account for each source of uncertainty. These matrices are then summed to form $C$. We assume the measurement uncertainty to be uncorrelated in time (i.e. a white noise process). Thus, the associated covariance matrix $C_{meas}$ is diagonal, and the diagonal elements $C_{jj}$ are each equal to the variance of observation $o_j$, $\sigma_j^2$, as estimated during the measurement process.

The covariance matrix of the modeling uncertainty, which we denote $C_{mod}$, is more complicated, since the residual vector contains any autocorrelated noise in the time series that is not accounted for by the piecewise linear fits. Additionally, the time series contain outliers with respect to these linear fits, and these can impact any non-robust estimate of covariance. Finally, an initial idea of the model must be used to calculate residuals, and thus estimate their covariance. These challenges can be circumvented when data resolution is low enough to assume that residuals are uncorrelated, as in Parrenin et al. (2013), however, including the covariance matrix allows us to make use of noisy, high-resolution data.

We arrive at an initial model by running a MH simulation in which $C$ is assumed equal to the identity matrix, and select the best fit of this run. Note that $C_{meas}$ is not taken into account at this point, since we require an independent estimate of $C_{mod}$. At this point, covariance could be estimated directly, but tests indicated that this method was not robust, making the covariance matrix estimate sensitive to outliers and to the initial model fit. Our $CO_2$ data are unevenly spaced in time, and developing a covariance matrix using the traditional covariance estimator would require some form of interpolation, which can introduce substantial error.

The residuals with respect to the initial model are instead used to fit an AR(1) model (Robinson, 1977; Mudelsee, 2002) which treats the autocorrelation between a pair of residuals $r_i$ and $r_{i-1}$ as a function of the separation between the two data points in time $t_i - t_{i-1}$. The Robinson (1977) / Mudelsee (2002) model is expressed as follows:

$$r_i = r_{i-1} \cdot a^{t_i - t_{i-1}} \tag{2}$$

where the constant $a$ determines the correlation between two residuals separated by $t_i - t_{i-1}$ units of time, and minimizing the loss function:

$$S(a) = \sum_{i=1}^{n} \left\{ r_i - r_{i-1} \cdot a^{t_i - t_{i-1}} \right\} \tag{3}$$

allows us to estimate $a$. We do so using a nonlinear least-squares estimate with L1-norm regularization to provide a robust estimate (Chang and Politis, 2016). We test the validity of the AR(1) hypothesis by comparing $r_i$ with $r_{i-1} \cdot a^{t_i - t_{i-1}}$ (Supplement). Given that the AR(1) hypothesis cannot be rejected, we can use $a$ to calculate the theoretical correlation between two residuals, and construct a correlation matrix $\mathbf{K}$ and the model covariance matrix $\mathbf{C}_{mod}$ as follows:

$$\mathbf{C}_{mod} = \sigma^2_{mod} \, \mathbf{K} \; ; \; \mathbf{K}_{ij} = a^{t_j - t_i} \tag{4}$$

where $\sigma^2_{mod}$ is the variance of the modeling error, assumed constant and estimated using a robust estimator based on the Interquartile Range (IQR), calculated as $(IQR(\mathbf{R})/1.349)^2$ (Ghosh, 2018; Silverman, 1986). Finally, the covariance matrix of the residuals $C$ is calculated as:

$$\mathbf{C} = \mathbf{C}_{mod} + \mathbf{C}_{meas}. \tag{5}$$

Rather than inverting the covariance matrix, we use Cholesky and Lower/Upper (LU) decompositions to solve for the cost function value $J$, as in Parrenin et al. (2015).

## 2.4 Estimating the posterior probability density

In general, the probability density of the change points cannot be assumed to follow any particular distribution, as short-timescale variations of the time series may lead to multiple modes or heavy tails, for example. Thus, stochastic methods, which are best adapted to exploring general probability distributions (for example, Tarantola (2005)), are suited to our problem.

To tackle the large computation time required for traditional MH sampling, we apply the ensemble sampler developed by Goodman and Weare (2010) (GW) as implemented in the Python emcee library (Foreman-Mackey et al., 2013). This sampler adapts the MH algorithm so that multiple model walkers can explore the probability distribution at once, making the algorithm

parallelizable. It has the advantage of being affine invariant: that is, steps are adapted to the scale of the posterior distribution in a given direction.

The final task in our piecewise linear analysis is to identify the number of change points to best represent the two series we wish to analyze. The choice should reflect our goal of accurately investigating millenial-scale variability. Further, we aim for parsimony in the representation. To best balance these two goals, we apply the Bayesian Information Criterion (BIC, Schwarz et al. (1978)) to the number of points we allow to fit the two series.

We apply a joint BIC–normalizing the cost function for each series by its lowest value–and arrive at the conclusion that the two series are best compared by fitting 8 points. The histograms created for fits of 7, 8 and 9 points of the two series are remarkably similar, and we assess that our choice of 8 points does not add significant uncertainty to the timing of change points, with the exception of the change point at ACR onset. We include histograms of fits between 5 and 9 points in the supplementary materials. We also include change point timings and lead-lag estimates calculated using 7-point fits in the supplement.

The most probable timings are identified by probability peaks, or modes–for a fit of $n$ points, we analyze the $n$ time periods with greatest contiguous cumulative probability. Thus, we analyze a coherent number of change points, and avoid setting artificial probability thresholds. We avoid comparing incoherent modes by separating changes by the sign of the change in slope of the fits. If the slope decreases at a change point, the change in slope is negative, or concave-down. These changes are indicated by the downward-facing part of the histogram graphs. Note that while this part of the histogram appears 'negative' probabilities cannot be negative, and this simply indicates that the probability is for a concave-down change point. If the slope increases at a change point, the change in slope is positive, or concave-up. The probabilities of these changes are indicated by the upward-facing, or 'positive' part of the histogram. When we calculate leads and lags, we only do so for either a region in which there is a probability peak for a concave-down change point in both series, or for a concave-up change point in both series, but we do not treat concave-up probability and concave-down probability together.

## 2.5 Phasing

We estimate $\rho_{lead}^{ATS3}$, the probability that ATS3 leads $CO_2$ over a given interval, as

$$\rho_{lead}^{ATS3} = (\rho_x^{ATS3} \circ \rho_x^{CO_2}) \star \rho^{chron}, \tag{6}$$

where $\rho_x^{ATS3}$ is the probability of a change point at time $x$ for ATS3, $\rho_x^{CO_2}$ is the probability of a change point at time $x$ for $CO_2$, $\circ$ is the cross-correlation operator, which is used to calculate the probability of the difference between two variables, and $\star$ is the convolution operator, which is used to calculate the probability of the sum of two variables. $\rho^{chron}$ is the probability distribution of the chronological uncertainty between the two records, which we take to be Gaussian centered on 0, with standard deviation $\sigma = \sigma_{chron}$ (shown in Figure 3). The intervals associated with each change point are given in Figure 6.

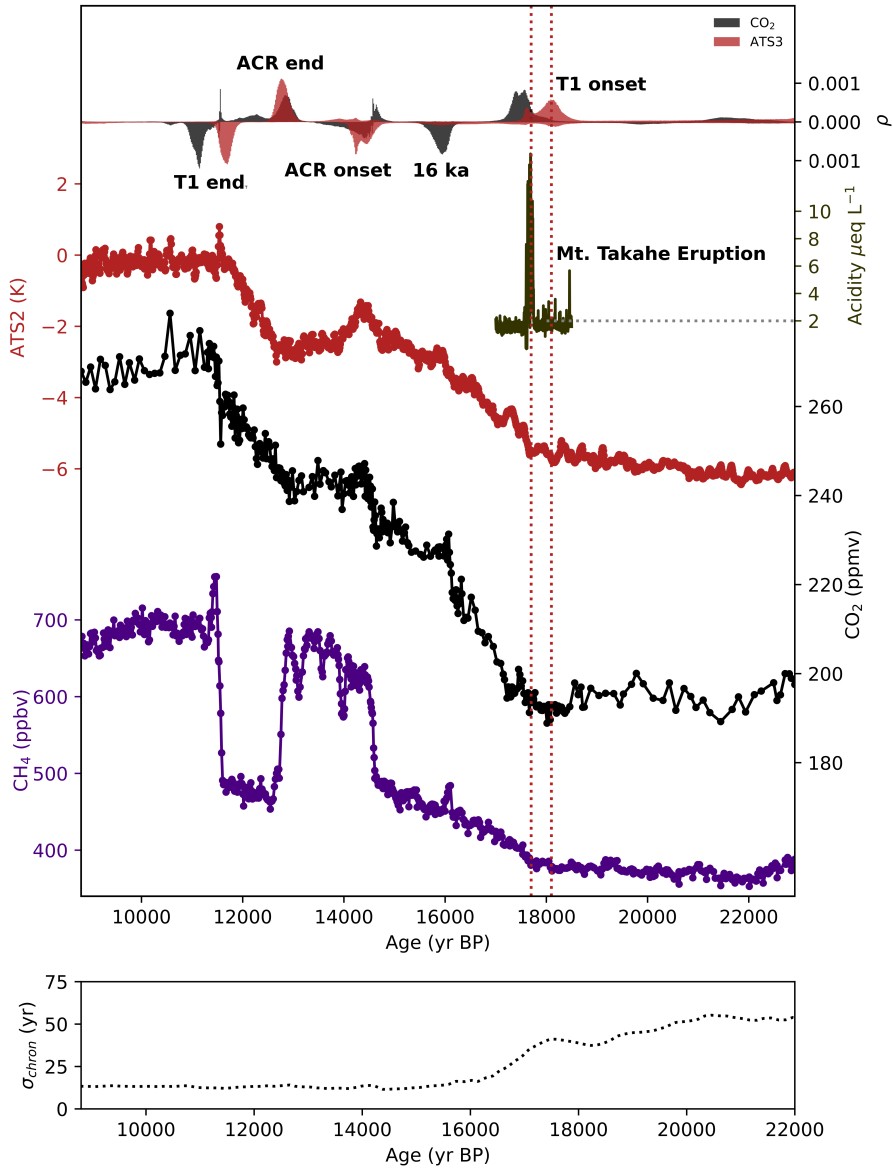

**Figure 3.** Upper panel: Atmospheric $CO_2$ (black) and ATS3 (red) placed on a common time scale, with the normalized histograms of probable change points (8 point simulations, allowed to fit 6 points and two endpoints per series). Histograms are plotted downward-oriented when the rate of change decreases and upward-oriented when it increases (same colors, y-axis not shown). Probabilities are normalized so that the integrated probability for a given histogram sums to 1. In four distinct time intervals, both series show concurrent probable change points. We also plot WD Acidity (green) and WD $CH_4$ (violet) series. Vertical lines are plotted to highlight select change point modes for the $CO_2$ (black) and ATS3 (red) series. $CH_4$ tracks changes in Northern Hemisphere climate. $CO_2$ modes correspond with rapid changes in $CH_4$ at the ACR end, ACR onset, 16 ka BP rise, and the rapid rise preceding the T1 end. Lower panel: Chronological uncertainty, taken as the sum of the Δage uncertainties and the uncertainty estimate for our volcanic synchronization.

# 3 Results and discussion

## 3.1 Change point timings

The change point histograms for the ATS3 and $CO_2$ time series in Figure 3 confirm that the millenial-scale changes in the two series were largely coherent. We focus on four major changes in trend which are common to both series: the onset of the deglaciation from 18.2 to 17.2 ka BP; the onset of the Antarctic Cold Reversal (ACR) at around 14.5 ka BP, the ACR end beween 12.9 and 12.65, and the end of the deglaciation, at approximately 11.5 ka BP. For each of these four changes, we calculate the probability of a lead or lag over the time interval that encompasses the continuous peaks in the two histograms. Two $CO_2$ change points, one centered at approximately 16 ka BP, and one just before the ACR onset at 14.4 ka BP, do not have high-probability counterparts in the ATS3 series. A low-probability change point for the temperature series after the ACR onset, centered at 14 ka BP, does not have a counterpart in the $CO_2$ series.

The deglaciation onset begins with a large, postive change point mode for Antarctic temperature, centered around 18.1 ka BP. The corresponding change point for the $CO_2$ series is centered around 17.6 ka BP.

The $CO_2$ rise peaks at around 16 ka BP, identified by a downward-oriented probability peak, which has no significant counterpart in the temperature series. This peak is followed by a brief plateau in $CO_2$ concentrations, before a gradual, accelerating resumption of the increase.

At the onset of the Antarctic Cold Reversal, an upward-facing $CO_2$ change point at 14.4 ka BP is followed by a broad, downward-facing $CO_2$ change point which peaks at around 14.25 ka BP. The first peak appears to reflect a centennial-scale change (identified by Marcott et al. (2014)), and the broadness of the second peak reflects further methodological uncertainty with respect to the timing of the millenial-scale change in $CO_2$. An unambiguous negative temperature change also occurs at around 14.3 ka BP, roughly concurrent with the downward $CO_2$ change point. Antarctic temperature began to descend after the ACR onset, and it is worth mentioning the low-probability, concave-up change point mode centered on 13.9 ka BP, particularly because this point is much more probable for some of the individual isotopic record. No corresponding change point is detected for $CO_2$.

The ACR termination is represented by significant modes in both series. An increase in $CO_2$ began at the peak occuring around 12.85 ka BP, while the ATS3 increase is centered at 12.78 ka BP, appproximately, reaching its maximum around 12.7 ka BP.

The end of the deglacial warming in Antarctica is well-defined in the ATS3 series, with a large mode reaching its maximum at 11.7 ka BP. The corresponding $CO_2$ mode reaches its maximum at around 11.15 ka BP. Visually, we might question why the CO2 change point is deemed to most probably occur at 11.15 ka, when a kink in the series appears to occur closer to 11.5 ka. It is important to note that two minor spikes in probability appear to fit the rapid rise that occurs close to 11.5 ka, and that the downward spike at the end of this rise indeed indicates that a small number of accepted iterations do indeed fit a change point here.

Since the resolution of the WAIS Divide dataset decreases considerably after the Holocene onset, and only three points account for the majority of the rapid rise that occurs before the Holocene onset, most of the weight is given to the obvious line

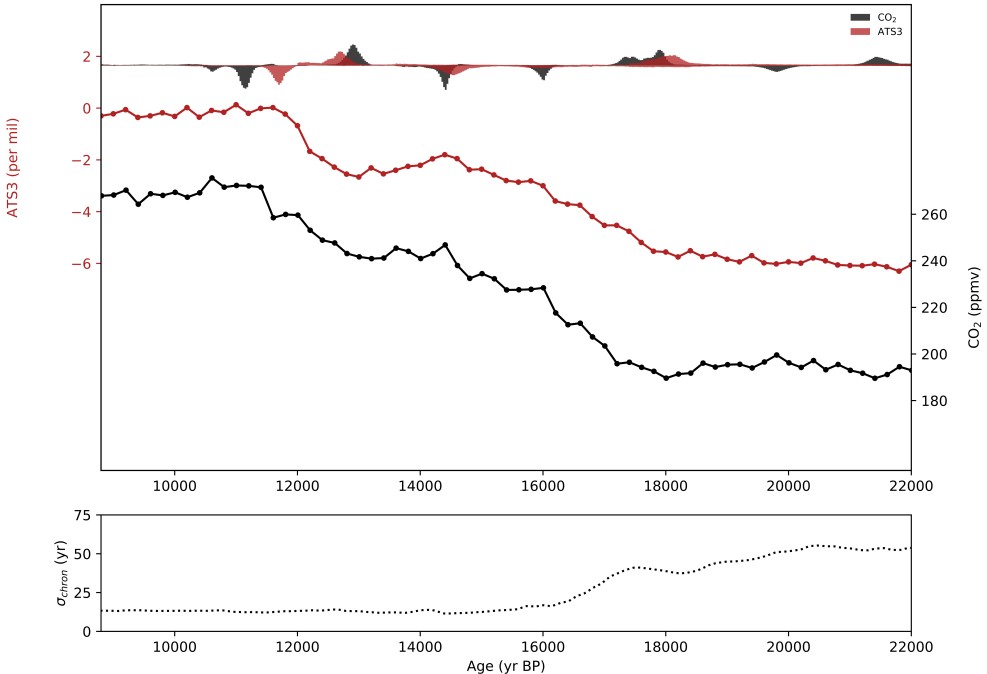

**Figure 4.** Upper panel: Savitsky-Golay filtered atmospheric $CO_2$ (black) and ATS3 (red) placed on a common time scale, with the normalized histograms of probable change points (8 point simulations, allowed to fit 6 points per series). Histograms are plotted downward-oriented when the rate of change decreases and upward-oriented when it increases (same colors, y-axis not shown, probabilities range from 0 (center) to 0.0024 (top/bottom)). Lower panel: Chronological uncertainty, taken as the sum of the $\Delta$age uncertainties and the uncertainty estimate for our volcanic synchronization.

beginning at the ACR end. Adding an additional point should allow us to fit this slightly better–the CO2 series is slightly better fit with 9 points, according to the individual BIC values, and there is more probability mass around the rapid rise in the 9-point fit, though the peak at 11.15 ka is still dominant. In any case, some methodological uncertainty exists regarding the location of this point, and the probability estimate is possibly biased by the quick change in resolution. Better resolution around this point

5   will help identify the true location of the change.

As a second test of the timings of millenial-scale events, we use our method to fit filtered versions of the ATS3 and WAIS Divide $CO_2$ data. A Savitsky-Golay filter, designed to have an approximate cutoff periodicity of 500 years, is applied to the two records. Fitting change points to these two series allows us to verify that our leads and lags are not overly influenced by sub-millenial scale noise in the original records.

10   Figure 4 shows the Savitsky-Golay filtered $CO_2$ and ATS2 time series, and the corresponding change point histograms. The four major changes identified in both series, at the T1 onset, the ACR onset, the ACR end, and T1 end, are similar in shape and center to the change points identified for the raw data. However, there are two notable differences between the two fits.

First, the histograms are smoother and have broader peaks. This is not surprising, given that the Savitsky-Golay filters are designed to remove all variability with periodicities less than 500 years, whereas the covariance matrix applied to the fits of the raw data only treats an approximation of AR(1) correlated noise. Second, the pre-ACR change in $CO_2$ is removed from the filtered series, which is again reasonable as it appears to mark a centennial-scale event. Savitsky-Golay filtering has its own drawbacks–data reinterpolation is required, for example, and propagating measurement uncertainty becomes difficult. However, the similarity of the two results supports our fits of the raw data.

## 3.2 Change point timings for individual temperature records

Histograms calculated for each of the regional $\delta^{18}O$ records are shown in figure 5. These histograms should still be interpreted cautiously, as additional information included in the isotopic records here assumed to represent temperature–the signal of ice sheet elevation change, for example–are not corrected for. The comparison of these histograms provides an initial, exploratory picture of potential regional differences in climate change during the last termination.

Of the four changes identified as coherent between the temperature stack and $CO_2$, those at the deglaciation onset, the ACR end, and the T1 end are expressed as significant probability peaks in all five records. Some ambiguity appears to exist about the timing of the ACR onset in the EDML record. It is expressed by a rather broad, non-significant probability mode extending between 16 ka BP and 14 ka BP, though a significant spike at 14 ka BP marks the downturn seen in the other records. The ACR onset is significant and well-defined in all of the five other records. Three of the records–TD, EDC and WD–show a marked stabilization in temperature after the ACR onset, near 13.8 ka BP, which appears as a region of much lower probability in the ATS3 stack.

The WAIS Divide record is, notably, the only isotopic record in the stack from the West Antarctic ice sheet. We could thus reasonably expect it to show considerably different trends from the other records. Indeed, a changes in the WD temperature record occur at 22 ka BP and 20 ka BP. This earlier change in the isotopic record was identified and confirmed to indeed be a temperature signal by Cuffey et al. (2016) using a borehole temperature record, though their study places the change at 21 ka BP. We confirm that the onset of the deglacial temperature rise in West Antarctica likely began as much as 4 ka BP before the onset of temperature rise in East Antarctica. Interestingly, the WD record also shows a temperature change point around 17.8 ka BP, expressed slightly later than in the other records and more synchronous with $CO_2$. This apparent acceleration of the temperature rise is followed by a significant downward-facing change point not seen in any of the other records. A difference appears to exist in timing at the T1 end as well, with temperature change at WD apearing to precede the East Antarctic records, and the DF temperature change, centered at 11.2 ka BP, occuring more synchronously with $CO_2$.

## 3.3 Leads and lags

The probability densities of leads and lags at the coherent change points between ATS3 and $CO_2$ are shown in Figure 6. We then report the central 68% and 95% probability intervals for each histogram, thus avoiding any misinterpretation resulting from Gaussian approximation (mean and standard deviation). These values are grouped in Table 1.

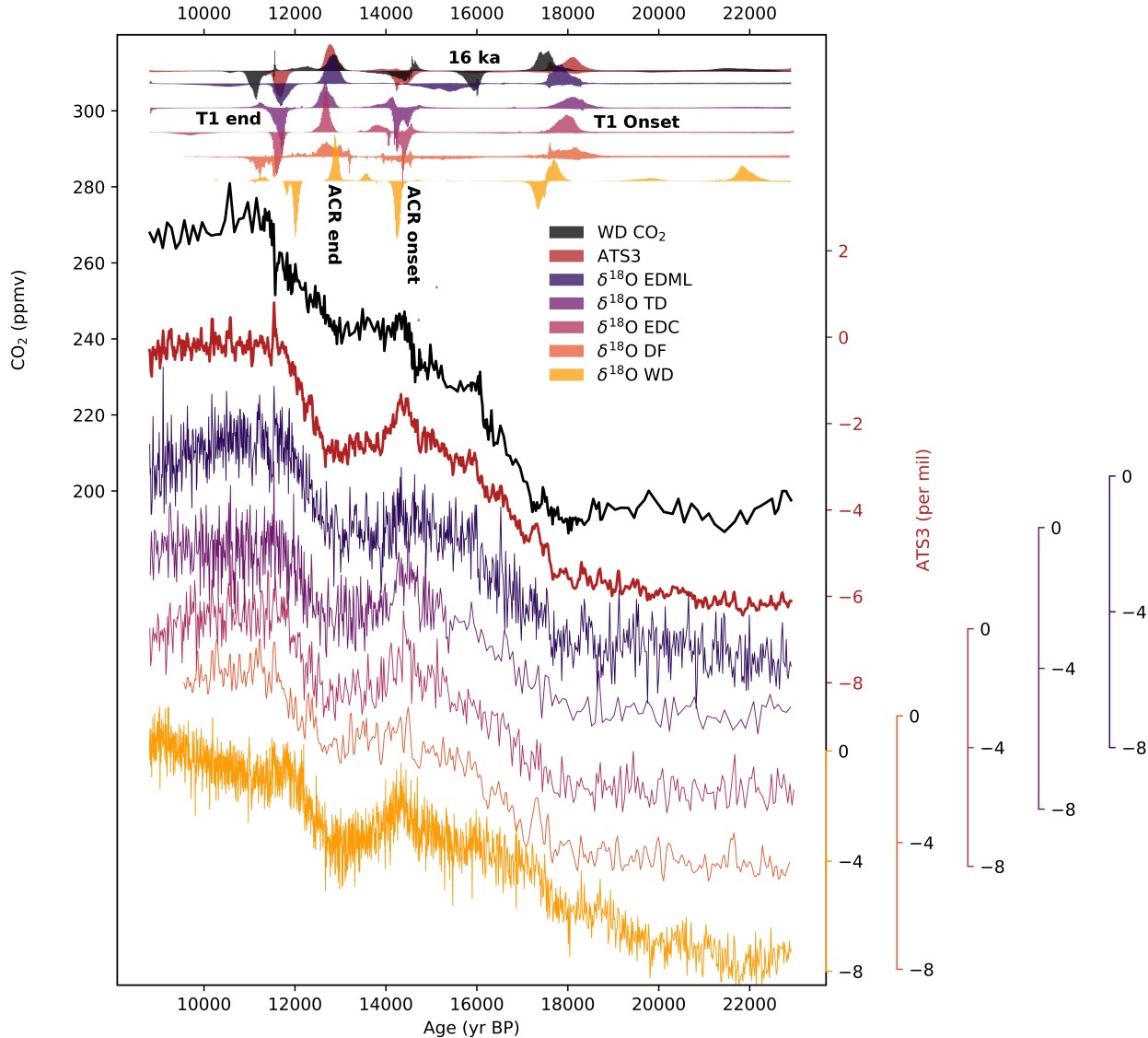

**Figure 5.** Atmospheric $CO_2$ (black) and individual $\delta^{18}O$ records placed on a common time scale, with the normalized histograms of probable change points (8 points) for each ice core used in the ATS3 stack; the locations of the drill sites are shown in the top right corner. Details of the histogram plots are as in Figure 3. Maximum probability estimates and 68/95% probability intervals for timings of the individual records are provided in the supplement. The $\delta^{18}O$ records are given in per mil anomalies with respect to the last 200 years, as is the ATS3 stack.

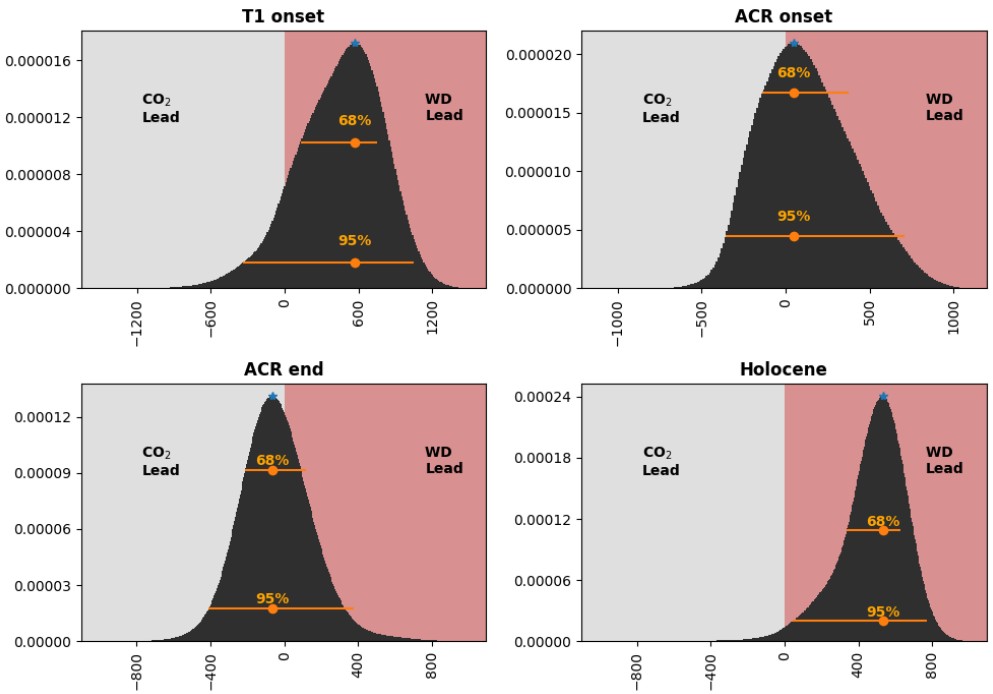

**Figure 6.** Probability density $\rho$ (y-axis, normalized) of an ATS lead (x-axes, in years) at each of the selected change point intervals (noted on subfigures). Negative x-axis values indicate a $CO_2$ lead. The maximum probability lead/lag and 68% / 95 % central probability intervals are indicated by the orange dot and lines on each histogram.

| | 95 % (lower) | 68 % (lower) | Maximum | 68 % (upper) | 95 % (upper) | Parrenin et al. (2013) | $\sigma$ |
|---|---|---|---|---|---|---|---|
| **T1 Onset** | -338 | 127 | 570 | 751 | 1045 | -10 | 160 |
| **ACR Onset*** | -357 | -137 | 50 | 376 | 708 | 260 | 130 |
| **ACR End** | -410 | -211 | -65 | 117 | 375 | -60 | 120 |
| **T1 End**[†] | 45 | 337 | 532 | 629 | 773 | 500 | 90 |

**Table 1.** Maximum probabilities and central probability intervals for leads and lags at each of the selected change point intervals. Negative x-axis values indicate a $CO_2$ lead. [†] Note the small, but distinct probability that the $CO_2$ change point occurs closer to 11.5 ka would indicate a lag of 174 years, with a 68% central probability range of 65 to 280 years. *The phasing at the ACR onset is sensitive to whether 7 or 8 points are used. Timings with 7 and 8 point fits, and for the individual isotopic records, are made available in the Supplementary Materials.

ATS3 led $CO_2$ by 570 years, (within a 68% interval of 128 to 751 years) at the T1 onset. Given the large range of uncertainty, though, we cannot exclude the possibility of synchrony at the 95% level, which, interestingly, appears to be the case for the Dome Fuji record. At the ACR onset, we are not able to identify a clear lead or lag. At this point, phasing is sensitive to the number of points used to make the calculation: with 7 points, we calculate a 240 year lead of ATS3, and with 8 points, we

calculate a 50 year lead. In neither of these cases can we exclude synchrony within 95 % probability, and with 8 points, it is well within 68 %. At the ACR end, $CO_2$ led ATS3, by 65 $\pm$ years within a 221% range of 228 years to -117 years (a temperature lead) and so again, the possibility of synchrony cannot be excluded within 68 % probability.

At the T1 end, a $CO_2$ lag is certain. Calculating the phasing between 12.0 ka BP and 11.0 ka BP, we obtain an ATS3 lead
of 532 years, with a 68% probability range of 337 to 629 years. This estimate is complicated, though, if we consider the small possibility that the true $CO_2$ change point occurs closer to 11.5 ka BP, at the end of the rapid rise. In this case, the phasing is reduced to 174 years (68% central probability range of 65 to 280 years) and synchrony is within the 95% central probability interval (-71 to 411 years).

## 3.4   Discussion

Our results refine and complicate the timings and leads and lags identified by the most recent comparable studies (Parrenin et al., 2013; Pedro et al., 2012). We identify a $CO_2$ change point not treated in these studies at 16 ka BP, and one before the ACR onset, associated with the centennial-scale rapid rises identified by Marcott et al. (2014). We also treat regional isotopic records, and identify a change point occuring at 13.9 ka BP in three of the records. None of these change points have a marked counterpart in the other series.

During the major, multi-millenial scale changes which occur at T1 onset and T1 end, Antarctic temperature likely led $CO_2$ by several centuries. However, during the complex, centennial-scale change at the ACR onset, we cannot calculate a clear lead of either ATS or $CO_2$, and at the end of the ACR, $CO_2$ leads temperature. Further, we do not identify a temperature analog for the $CO_2$ change at 16 ka BP, or an analog in $CO_2$ of the temperature stabilization in ATS3 after the ACR onset, itself not present in all of the regional $\delta^{18}O$ records, indicating at least some degree of decoupling during these changes. Additionally,
the $CO_2$ changes at the ACR onset and T1 end are overlayed with centennial-scale substructures. Finally, synchrony is within the $2\sigma$ uncertainty range for each of the phasings, with the exception of the T1 end.

The changes in $CO_2$ occurring at the ACR onset, ACR end and the T1 end have been identified to correspond with changes in $CH_4$ (Marcott et al., 2014), which are thought to originate in tropical wetland sources (Chappellaz et al., 1997; Fischer et al., 2008; Petrenko et al., 2009) and are indicative of Northern Hemisphere and low-latitude temperature changes during
the deglaciation (Shakun et al., 2012). Indeed, the $CO_2$ modes appear to demarcate the rapid changes in the WD $CH_4$ record, shown in Figure 3.

The beginning of a gradual rise in $CH_4$ at around 18 ka BP appears to be near-synchronous with the T1 onset rise in Antarctic temperature. This rise is not seen in Greenland paleotemperature records, where it may have been masked by AMOC-driven wintertime cooling (Buizert et al., 2017) but it appears as well in proxy temperature stacks spanning both the Northern and
Southern 0° to 30° latitude bands (Shakun et al., 2012).

Tephras from Mt. Takahe, a stratovolcano located in West Antarctica, have been detected in Antarctic ice cores during a 192 year interval around 17.7 ka BP. It has been postulated that this eruption may have provoked changes to large-scale SH circulation via ozone depletion, possibly triggering the transition between the gradual SH temperature rise beginning well before 18 ka BP and the more rapid rise marking the deglaciation (McConnell et al., 2017). The $CO_2$ mode we find at the

deglaciation is coeval with this event within the range of dating uncertainty (Figure 3), and $CH_4$ visually appears to accelerate concurrently. However, the cumulative probability of the ATS3 change point is much greater before 17.7 ka BP than after (approximately 80% of the probability density occurs before, see Figure 3); hence our results do not support McConnell et al. (2017)'s proposed volcanic forcing of the temperature change.

Though the T1 onset and the ACR end are both thought to originate in AMOC reductions (Marcott et al., 2014), our results allow for the $CO_2$-ATS3 phasing to be different during the two events, with the maximum probabilities reversed in directionality (i.e. with temperature leading at T1 and $CO_2$ leading at the ACR end, though zero phasing is within 95% error). Though $CH_4$ appears to change alongside $CO_2$ during both intervals, the phasings between $CO_2$ and ATS3 are opposite in direction and different in slope. This hints at a complex coupling, depending on conditions defined by multiple other variables

and mechanisms, between $CO_2$ and Antarctic Temperature. Bauska et al. (2016), for example, hypothesize that an earlier rise of $CO_2$ at 12.9 ka BP, driven by land carbon loss or SH westerly winds, might have been superimposed on the millenial-scale trend.

The apparent decoupling between $CO_2$ and ATS3 at 16 ka BP also merits further discussion. We do not detect a significant change-point for any of the isotopic records at 16 ka BP, but the EDML record contains extremely broad uncertainty associated

with the significant ACR onset peak, stretching to 16 ka BP, which indicates that this portion of the EDML time series is indeed notably different in shape from the other records, even if a clear signal is not identified at 16 ka BP by our method. EDML indeed appears to record changes in AMOC differently than the other isotopic records (Landais et al., 2018; Buizert et al., 2018). $CO_2$ and ATS3 are similarly apparently decoupled at the temperature change point centered at 14 ka BP, and this point could be indicative of variability specific to the Pacific/Eastern Indian Ocean sectors, as it is present only in the TALOS Dome

and EDC records, and slightly later, around 13.7 ka BP, in the WAIS Divide record, indicating a cooling trend after the ACR onset which is not clear in the DF or EDML series.

Within the range of uncertainty, the mean of our lead-lag estimates is consistent with the boundaries proposed by Pedro et al. (2012). Our results are consistent with those of Parrenin et al. (2013) for three out of the four change points addressed, but differ considerably at the T1 onset.

The considerable difference at T1 between our result and that of Parrenin et al. (2013) is most likely due to the much higher resolution of the WD $CO_2$ time series. It is also possible that the result of Parrenin et al. (2013) was limited to a local probability maximum of this change point in the $CO_2$ series. The addition of the WD paleotemperature record and removal of the Vostok record from ATS3, the updated atmospheric $CO_2$ dataset, and our more generalized methodology are all, in part, responsible for the differences in computed time delays (SI). This testifies to the importance of data resolution, methodological development,

and chronological accuracy in the determination of leads and lags.

## 4   Conclusions

Our study is a follow-up of the studies by Pedro et al. (2012) and Parrenin et al. (2013) on the leads and lags between atmospheric $CO_2$ and Antarctic temperature during the last deglacial warming. We refine the results of these studies by using the

high resolution $CO_2$ record from WD; using $\Delta$age computed on WD; using a new Antarctic Temperature Stack composed of 5 volcanically synchronized ice core isotope records, developed by Buizert et al. (2018); and using a more precise and complete probabilistic estimate to determine change points. Our methodology detects four major common break points in both time series. The phasing between $CO_2$ and Antarctic climate is small but variable, with phasing ranging from a centennial-scale $CO_2$ lead, to synchrony, to a multi-centennial-scale lead of Antarctic climate. This variability of phasings indicates that the mechanisms of coupling are complex. We propose three possibilities: I) the mechanisms by which $CO_2$ and Antarctic Temperature were coupled were consistent throught the deglaciation, but can be modulated by external forcings or background conditions that impact heat transfer and oceanic circulation (and hence $CO_2$ release); II) these mechanisms can be modulated by internal feedbacks that change the response timings of the two series; and/or III) multiple, distinct mechanisms might have provoked similar responses in both series, but with accordingly different lags.

We also explore the hypothesis of regional differences in temperature change in Antarctica. Though the use of individual isotopic temperature records is complicated by influences other than regional temperature, including localized variations in source temperature and ice sheet elevation change we confirm that the deglacial temperature rise did not occur homogeneously accross the Antarctic continent, with significant differences existing between the WAIS Divide and East Antarctic records at the onset of the termination, and smaller potential differences occuring between the East Antarctic records, including a considerably later end of the deglacial warming in the Dome Fuji record.

Hypotheses of relationships between these events should now be reinvestigated with modeling studies. The relationship between $CO_2$ and Antarctic temperature on longer timescales and during other periods of rapid climate change is also of interest. Additional high-resolution West Antarctic paleotemperature records would allow for a robust investigation of regional differences between West and East Antarctica, and our analysis at the T1 end could be improved with continued high-resolution $CO_2$ measurements through the beginning of the Holocene. Finally, the continued measurement of high-resolution ice core $CO_2$ records is essential to understand the relationship between $CO_2$ and global and regional temperature during the last 800,000 years.

*Code and data availability.*

*Acknowledgements.* We thank Michael Sigl, Jinhwa Shin, Emmanuel Witrant and Amaelle Landais for their support and great help discussing this work and Mirko Severi for his EDC data and support with the volcanic synchronisation. This work is supported by the Fondation Ars et Cuttoli, and by the LEFE IceChrono and $CO_2$Role projects.

*Competing interests.* The authors declare that no competing interests are present for this study.

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
