# Peer review of "Antarctic temperature and CO2: near-synchrony yet variable phasing during the last deglaciation"

_Climate of the Past, 2018_

## Referee Comment (RC1) · Anonymous Referee #1 · 22 May 2018

Summary:

Beeman et al. investigate the phase relationship between atmospheric CO2 concentration and Antarctic temperature during the last deglaciation. This question has been investigated various times over the last decades as it can illuminate on the role of CO2 forcing and carbon cycle changes in the deglaciation. In comparison to earlier studies, Beeman et al. use a CO2 dataset (WAIS) that is better resolved in time, and better constrained in terms of its delta age uncertainty. In addition, they include new water isotope data from the WAIS divide ice core. Furthermore, they do not just estimate a mean lag between CO2 and temperature over the entire study period (e.g., Pedro et

al. 2012) but investigate the relative timing of change-points in both timeseries, similar to (Parrenin et al. 2013). Their results indicate that CO2 and temperature change synchronously except at the end of the Antarctic cold reversal (CO2 leads) and the onset of the Holocene (CO2 lags). The availability of the new high resolution CO2 data from WAIS and the approach to investigate change-points rather than a mean lag make this paper interesting and timely. The results will be a valuable contribution to the discussion of causes and effects of deglacial climate changes. However, I have some reservations about aspects of the manuscript and method that warrant further testing and/or justification.

General Comments:

i) The authors use a stack of five Antarctic ice core records as their temperature estimate. Their argument is that this reduces local noise. While this may be true, it also removes local differences in the temperature sensitivity to CO2 and/or sea ice changes. For recent climate change, Jones et al. (2016) showed that the local temperature trends vary strongly across Antarctica, including cooling in some regions. If we use figure 1 in Jones et al. as a potential analogue, than 4 out of the 5 cores used by Beeman et al. fall into regions where temperature hasn't warmed or is cooling since 1979. Thus, from recent observations one can expect the response of Antarctic temperature to CO2 forcing to be spatially heterogeneous. Hence, instead of stacking the isotope records, I think it is more informative to determine the phase relationship for each ice core. Because: If we want to determine cause and effect between CO2 and Antarctic climate, it is the first robust temperature rise at any ice core that is informative and not their mean. Obviously, this will increase the noise in the temperature dataset, but ideally a change-point detection method can handle noise (see also specific comments). If all ice cores lead to internally consistent estimates of temperature change-points, their likelihood estimates can be combined to reduce the errors – but this consistency would first need to be shown. And in case of inconsistencies this needs to be discussed. The stacking may also influence the detection of fast tempera-

ture rises (such as around 16k) where already minor synchronization uncertainties and differences in resolution can lead to smoothing.

ii) The authors state throughout the manuscript that they detect abrupt temperature rises corresponding to the rapid $CO_2$ rises discussed by Marcott et al. (2014). However, their method actually only detects the stabilization afterwards and not rises themselves (at least in temperature).

iii) In fact, this hints at another problem: The applied change-point detection method requires user input regarding how many change-points it is allowing for. The authors chose a number of 8 (and tested 7 for comparison). However, just by visual inspection of the $CO_2$ record one can identify more change-points: i) the onset of the deglaciation, ii) the 16k rise, iii) the levelling off after 16k, iv) the rise at 14.6k, v) the levelling off after 14.3k, vi) the end of the ACR, vii) the jump into the Holocene and viii) the levelling off at the onset of the Holocene. Including the start and end point in the series which have to be marked as change points in the method, this yields 10 change-points. In fact, all of these points are discussed in the manuscript. This creates the inconsistency, that the method only allows for 8 (minus 2, i.e., start  end) change points, but 10 (minus 2) are discussed – a solution that cannot be true in any of the realizations of the method: Some of these points must exclude each other if the maximum is set to 8. I think the authors should test their method allowing for more change points (e.g. 10).

iv) It is unclear, how the change-points that are actually discussed are chosen. The authors discuss the $CO_2$ rise at 16k, which has a very low probability peak, but do not discuss the large probability peak of a $CO_2$ increase at the onset of the ACR. How is this justified? I agree that the rise at 16k is the relevant one, but the method obviously implies a higher likelihood of a $CO_2$ increase at 14.3k? What does this imply about the methods ability to detect the correct change points and infer their timing? The authors should discuss more clearly, how they evaluate the likelihood of a given point to be a change-point at all, before discussing its timing.

Specific Comments:

PP1,L13: "Multimodal timings" – what do you mean? Multimodal probability distributions of change point estimates? I suppose the real change cannot be multimodal.

PP2, L6: "Consistently": Replace with "on average", since Shakun et al. do not discuss whether this is consistent over the entire time.

PP2, L8: Also global T did not increase continuously according to Shakun et al.

PP2, L20-21: Please rephrase the colloquial "thanks to the so-called isotopic paleothermometer". Possibly: "... due to temperature dependent fractionation of water isotopes during condensation"?

PP2, L24-25: Please replace the $CO_2$ lock in depth with the corresponding estimate from WAIS and a reference to (Buizert et al. 2015). Generally it is not clear throughout the manuscript, whether EDC $CO_2$ is used at all, and if so, how it is spliced together with WAIS $CO_2$. If it is not used, please shorten the methodological discussion of EDC $CO_2$ as it is misleading the reader to believe EDC $CO_2$ data was used too.

PP3, L10: $d15N$ of $N_2$ as a proxy for DZ height: Refer to (Buizert and Severinghaus 2016) who propose some uncertainty to this assumption?

PP4, L14: "stack": Please describe how this is generated, so that other people can reproduce this. Is this an average over all records? Are they resampled to equal resolution beforehand? Are they standardized or kept in degC? If kept in degC, which slope (per mille/degC) is used? Do all cores have similar amplitudes across the deglaciation (in degC or per mille) or can the stack be biased by single records with exceptional amplitude? Later on, the fitting procedure requires an error in K for each data-point: How is this derived? Does it include uncertainty from the isotope temperature conversion and other (e.g., circulation) influences on isotopes?

PP4, L17: "previously published ties": The list in (Parrenin et al. 2013) includes a lot of isotopic tie-points between the ice cores as well. Are these used here too? They

introduce some circularity in the approach as they will reinforce the structure and timing of isotope (temperature) changes in the ice core that is used as a target. Please clearly state, whether all tie-points are volcanic or not.

PP4, L17-18: Synchronization: Please provide a list of tie-points as well as the ECM data for both cores in the supplementary, so that people can reproduce the analysis.

PP4, L23: "1,030 measurements": Earlier (PP3, L14) it is stated that there are only 320 points? Are these replicates? If so, please state this as it is slightly confusing.

PP4, L27-28: "At EDC...": Again – is this used at all? If not, remove. If yes, elaborate how $CO_2$ records of such different resolution are stacked.

PP5, Figure2 caption: "Ratio of the age difference between two consecutive tie points.." ... and what? Not clear what is plotted here.

Section 2.3 – 2.5: I encourage the authors to have a look at these sections again, and try to rewrite them more clearly. As it is now, it is near impossible to really understand what's going on. The authors elaborate on how the MH sampler is working. This may be nice for saving computing time, but hopefully doesn't affect the results. A reference could be enough? At the same time the authors do not discuss more relevant aspects of the method: How does the method deal with irregular sampling resolution in the records. They just say it becomes less precise (PP6, L3), but is that really so, or can it become biased? Similarly, it is not discussed, how the uncertainty for $CO_2$ or ATS is derived. Are the residuals/uncertainties treated as independent or correlated? Since the method only detects linear trends, any other internal climate variability would basically be a correlated uncertainty (i.e., red noise, as opposed to white measurement noise)?

PP7, L13: "...and corresponding rise in temperature around 16ka": Is that true? Looking at figure 3, there is no detection of a ATS increase around 16ka. Only the stabilization. In principle I agree, that there appears to be an ATS increase. However, there

is a similar increase around 17kaBP in ATS without a corresponding change in CO2, implying that this may just be internal variability/noise. And in any case: The method does not detect either of these increases in ATS.

PP7, L17: see previous comment. The ATS rise at 16k is not actually detected by the method.

PP7, L18-21: Multimodality: This paragraph is written as if the CO2/ATS increase was multimodal. However, each realization of the method probably only picks one or the other mode as a possible change point, and never both. Hence, the multimodality reflects an uncertainty in the change-point identification, not the identification of two separate change-points. Is this correct? If so, please rephrase.

PP7, L22: "a small positive probability peak". The probability of this point being a change-point is very low, much less than for example the positive peak for CO2 around 14.4ka which is not discussed. How do the authors choose which peak to discuss? How trustworthy is a change-point that is apparently only used in a small number of iterations?

PP7, L24: See comment above. The ATS probability peak is very low. How reliable is the inference of a change point there (and its timing)?

PP7, L 24-25: This section illustrates my concerns about how well the method deals with noise, and how subjectively some probability peaks are discussed while others are not. Why is the positive probability peak in CO2 at 14.64ka discussed, while the bigger positive peak at 14.42 isn't? I agree that the rise at 14.64ka and the stabilization around 14.42ka are the relevant change points, but the statistical method doesn't. To me this highlights, that the method may underestimate noise in the CO2 and ATS data, and hence, depict potentially erroneous change-points, which also have a seemingly high degree of certainty (in terms of timing). Please comment.

PP8, "leads and lags": Generally, I think the results should be more explicitly compared

to (Parrenin et al. 2013), who applied a largely similar method to a different CO2 (and slightly different ATS) dataset. Their estimates could be shown in figure 4 for comparison. Are the results consistent for each change point?

PP8, L2: "...it is not obvious." Change to: "our method doesn't detect it."

PP8, L7: "either the histogram peak around 12.9 or ...". I don't think that you can interpret single peaks in the histogram like this. The timing of the rise in CO2 at the end of the ACR is detected as a broad probability distribution and not just by two minor peaks in the histogram. All values within the probability distribution are the possible "true" value with a given likelihood. Even if the peaks are the most likely single values, the cumulative probability of the true value not being these peaks is higher (cf. it is very unlikely to actually draw the exact mean value of a normal distribution).

PP8, L6: "which loses data resolution.." See earlier comments: How does the method deal with this?

PP8, L6-8: Again: Can all these modes really be interpreted in other terms than uncertainty? The authors mention "later peaks, where data resolution is lower, to be likely indicative of higher frequency variability or noise". If the method cannot handle those, how good are the timing estimates for the other change-points? Please elaborate.

PP9, L1-2: "Applying the cross correlation operator...". How is it handled when the method indicates near equal probabilities of a CO2 rise and fall like around the ACR onset?

PP9, L13-14: See earlier comments on the reliability of the method and the handling of noise and resolution.

PP9, L14: "Calculating the phasing between 12-11.5..." How is this done? Are certain values excluded from the histogram for CO2? How?

PP10, L5-6: "We identify a coherent ATS2 change-point..." See general comments. I don't think this is the case.

PP 10, L11-12: "minor modes". See earlier comments. Can these really be interpreted?

PP10, L27-29: "Mt. Takahe" I do not understand why the cumulative probability of the ATS2 change-point is relevant here. The method does not detect whether there are multiple change points there, but only a single one with a given uncertainty in timing. Mt. Takahe falls into the uncertainty range of the detected change point at that time. Correct?

PP11, L1: "Here we confirm". See general comments. The method does not detect ATS rises coinciding with the rapid CO2 rises.

PP12, L 3: "we identify change points". See earlier comments.

Technical Comments:

PP2, L14: "... is thought to have..." (not "haved")

PP2, L19: "... and atmospheric composition..." (not "atmosphere")

PP4, L18: "Sigl et al. 2015": Change to 2016.

PP4, L17-18: "the offset in between ice and the air trapped much later at a given depth": Convoluted, please rephrase. Possibly: "The age difference between trapped gas and the surrounding ice matrix, delta age, ..."

PP7, L22: "a small positive probability peak". Probability is always positive. Please rephrase.

PP9, L4: "2 sigma": The numbers in the following paragraph (PP9, L7-9) match the numbers in figure 4. However, the caption of figure 4 says this was 1 sigma? Please check.

PP9, L8: "At the peak of the 16ka rise". Since you do not detect the 16k rise in ATS a better formulation could be: "CO2 and ATS stop rising synchronously at the onset of

the ACR" or similar.

PP10, L4: replace "in AMOC" with "from AMOC"

References:

Buizert, C., K. M. Cuffey, J. P. Severinghaus, D. Baggenstos, T. J. Fudge, E. J. Steig, B. R. Markle, M. Winstrup, R. H. Rhodes, E. J. Brook, T. A. Sowers, G. D. Clow, H. Cheng, R. L. Edwards, M. Sigl, J. R. McConnell and K. C. Taylor (2015). "The WAIS Divide deep ice core WD2014 chronology - Part 1: Methane synchronization (68–31 ka BP) and the gas age–ice age difference." Clim. Past 11(2): 153-173.

Buizert, C. and J. P. Severinghaus (2016). "Dispersion in deep polar firn driven by synoptic-scale surface pressure variability." The Cryosphere 10(5): 2099-2111.

Jones, J. M., S. T. Gille, H. Goosse, N. J. Abram, P. O. Canziani, D. J. Charman, K. R. Clem, X. Crosta, C. de Lavergne, I. Eisenman, M. H. England, R. L. Fogt, L. M. Frankcombe, G. J. Marshall, V. Masson-Delmotte, A. K. Morrison, A. J. Orsi, M. N. Raphael, J. A. Renwick, D. P. Schneider, G. R. Simpkins, E. J. Steig, B. Stenni, D. Swingedouw and T. R. Vance (2016). "Assessing recent trends in high-latitude Southern Hemisphere surface climate." Nature Clim. Change 6(10): 917-926.

Marcott, S. A., T. K. Bauska, C. Buizert, E. J. Steig, J. L. Rosen, K. M. Cuffey, T. J. Fudge, J. P. Severinghaus, J. Ahn, M. L. Kalk, J. R. McConnell, T. Sowers, K. C. Taylor, J. W. White and E. J. Brook (2014). "Centennial-scale changes in the global carbon cycle during the last deglaciation." Nature 514(7524): 616-619.

Parrenin, F., V. Masson-Delmotte, P. Köhler, D. Raynaud, D. Paillard, J. Schwander, C. Barbante, A. Landais, A. Wegner and J. Jouzel (2013). "Synchronous Change of Atmospheric $CO_2$ and Antarctic Temperature During the Last Deglacial Warming." Science 339(6123): 1060-1063.

Pedro, J. B., S. O. Rasmussen and T. D. van Ommen (2012). "Tightened constraints on the time-lag between Antarctic temperature and CO2 during the last deglaciation."

Climate of the Past 8(4): 1213-1221.

---

## Referee Comment (RC2) · Anonymous Referee #2 · 12 Jun 2018

**Overview**

Chowdhry Beeman et al. investigate the time relationship between Antarctic temperature and atmospheric CO2 during the last deglaciation. The question is of importance for our understanding of climate-carbon cycle feedbacks and has been tackled in multiple previous papers, most recently Parrenin et al 2013 and Pedro et al 2012. Chowdhry-Beeman et al. is distinguished from these previous studies by its use of the high time resolution and low delta-age WDC ice core CO2 record and a new regional Antarctic temperature stack. The consensus emerging from previous studies is that there is little-to-no significant time delay between CO2 rise and Antarctic temperature rise throughout most of the deglaciation. Chowdhry-Beeman et also find synchronous

changes within uncertainties (excepting the Holocene onset), however they place more attention than others on centennial-scale signals in the temperature and CO2 series and their purported relationship. The techniques used to analyse of the phase relationship are more complex than used in previous studies. Although I have no reason to doubt the techniques, I do have some concern about their rather qualitative and selective interpretation. In particular the conclusion that there is a significant change in Antarctic temperature corresponding to the abrupt CO2 change around 16ka is not convincing. My overall impression is that the complex technique used to generate the PD histograms is out of proportion to their qualitative interpretation.

Missing is some clear hypothesis testing and a more sceptical view of whether the centennial scale signals detected in ATS and CO2 are really meaningfully related. In general the approach appears very promising but the interpretation is still requiring quite some work. I would support publication after revisions to address the concerns below.

**Major Comments**
Section 3.1: The technique applied to study the time relationship between ATS2 and WDC CO2 is more complex than previous studies and difficult (at least for this reviewer) to follow. My concern is that despite the complex approach the interpretation of phasing, in the end rests on a rather qualitative assessment of the change point histograms presented in Figure 3. Adding to my concern is that there is never a clear description of what makes a mode distinct and worthy of discussion or of the precise criteria for defining a significant change point. The caption of Fig 3 says that the y-axis range for the probability histograms is 0 to 0.0024. It's not clear to me precisely what is meant here. Can you show horizontal lines marking e.g. 0.05 probability cut-offs, which could then be used to judge which modes are significant? Or am I entirely missing the point? If the latter, then please work on a better explanation of how to interpret these PD histograms. Some more specific examples of my concern about this section and approach as follows:

We are told (pp7 line 18) that the degalacial CO2 rise features 'two modes' (17.63 ka and 17.30 ka) separated by a 'distinct anti-mode'. On what basis is the anti-mode distinct? Could this be over-interpretation of noisy data?

Further down ( line 24) the authors describe 'a broad low probability peak' in ATS2 at 15.96 ka. It's not explained how 15.96 is selected as the centre of this peak or why this peak is considered significant, given there are similar amplitude peaks elsewhere in the deglacial CO2 record that are not discussed at all. The same can be said for the 'small upward probability peak' in CO2 at 16.15ka. This ambiguity about what is a significant feature and what is not continues throughout the section.

To give another example (line pp8 line 7), the author's describe 'two larger modes in CO2 at 11.12 and 11.01 ka.' as being 'indicative of higher-frequency variability or noise.' It's not clear on what basis these peaks are considered noise whereas the (smaller) peaks around 16 ka are considered meaningful and related to ATS2. My overall impression is that the complex technique used to generate the PDs is out of proportion to their qualitative interpretation.

I suggest the authors revise Section 3.1 to be shorter and more quantitative. I think part of the reason the some of the interpretation is unconvincing is that the authors do not appear to use their results to test any specific hypotheses. Instead we get a rather post-hoc interpretation of the leads and lags. Reframing the introduction to set up some specific hypotheses for testing could make the discussion more convincing.

Section 3.2: The methodology here appears good, however the section rests on the selection in Section 3.1 of five 'common' change points in ATS2 and CO2. As above its not clear by what criteria these 5 are selected. Please clarify.

p10 line 5. The significance of the ATS2 change point at 16k is not convincing. Please be more clear about the criteria for its selection over and above other peaks in the PDs that are not discussed at all.

P10 line 11. " during the complex, centennial-scale changes associated with the 16 ka rapid rise and the ACR onset, ATS was most likely synchronous with CO2 ". This is not convincing given the +- 340 yr uncertainty and the ambiguity of the 16ka peak in ATS2. I'd suggest a more cautious interpretation: centennial scale variability in both series (possibly physically related, possibly not) restricts ability to make any clear statement on significant leads or lags during this interval.

P10 line 22 to 29. McConnell et al suggested that accelerated warming was triggered by the Mt Takahe eruption. The finding here, that accelerated warming begins *before* the Mt. Takahe eruption, contradicts the McConnell hypothesis. The spin about "additional forcings beginning to accelerate warming before Takahe" is very unconvincing.

p11 line 1. The authors claim here to 'confirm' an imprint on Antarctic temperature of ice berg discharge to the Sth Ocn *and* Nth Atlantic around 16ka". This is not convincing at all. First, as above, the ATS2 signal around16k is questionable given other similar sized peaks in the PDs that are not discussed. Second, as the authors well know, correlation in timing does not prove of a casual relationship. Third, what is the imprint supposed to be (warming, cooling, stabilization?) and how did the icebergs drive it? Revise.

p11 line 5. The 'reversal in phasing' between T1 and the ACR end is not convincing. The phasing at T1 is 292+-343 yrs (1 sigma!), thus spanning from CO2 lag to CO2 lead. How can the phasing reverse if it is not distinct at T1? A simpler interpretation is that the ATS2 and CO2 are roughly synchronous with the exact lead-lag varying between change points due to centennial scale variability in both series.

p 11 line 9: "Centennial-scale variability may have been superimposed on coherent millenial scale trends, for example". Performing a similar analysis on band-pass filtered versions of the two series could be used to test this idea and would add a substantial new result.

p12 line 18. Comparison between east and west Antarctic temperature and CO2 could

already be done by making an east Antarctic and west Antarctic stack. The authors might consider doing this in revisions, it would add a substantial new result to the lead and lag discussion.

**Technical comments**

p10 line 5. It should be mentioned that within uncertainties the results are consistent with Parrenin et al and Pedro et al.

Figure 4. Important typo. I think the phasing at the ACR end should read *-*250 +- 188.

p 11 line 9: "Centennial-scale variability may have been superimposed on coherent millenial scale trends, for example". Performing a similar analysis on band-pass filtered versions of the two series could be used to test this idea and would add a substantial new result.

P 11 line 11. It's very difficult to follow this sentence. Please revise.

P12 line 3. " Notably, we identify change points in ATS2 that are associated with rapid rises in CO2." Which change points exactly? The previous paragraph comments that rapid change in CO2 and ATS2 around the ACR are not clearly in common. And my concerns remain about the significance of any signal in ATS2 around the rapid 16ka signal in CO2. Without further evidence this conclusion of related abrupt changes in not convincing and not justified to include as a major conclusion here or in the abstract.

P12 line 13-15: "This variability suggests complex mechanisms of coupling. Indeed, perhaps different mechanisms of ATS2 and CO2 rises, some coupled, others decoupled, were activated and deactivated (Bauska et al., 2016) throughout the deglaciation." This statement so encompasses all possibilities that it is almost meaningless.

Please advise where the new Antarctic temperature stack will be made publicly accessible upon publication.

---

## Author Comment (AC1) · 1 Sep 2018

Dear reviewers, dear editor,

We are grateful for the detailed suggestions provided by both anonymous reviewers, and believe that incorporating these suggestions will considerably improve our study. Below, we address the comments provided by each reviewer individually. Reviewer comments are included in italics.

On behalf of all co-authors,

Cordially,

[Figure]

Jai Chowdhry Beeman

Please also note the supplement to this comment:
https://www.clim-past-discuss.net/cp-2018-33/cp-2018-33-AC1-supplement.pdf

―――――――――――――――――

**Supplement:**

Dear reviewers, dear editor,

We are grateful for the detailed suggestions provided by both anonymous reviewers, and believe that  incorporating these suggestions will considerably improve our study. Below, we address the comments provided by each reviewer individually. Reviewer comments are included in italics.

On behalf of  all co-authors,

Cordially,

Jai Chowdhry Beeman

**Reviewer 1.**

*Summary: Beeman et al. investigate the phase relationship between atmospheric CO2 concentration and Antarctic temperature during the last deglaciation. This question has been investigated various times over the last decades as it can illuminate on the role of CO2 forcing and carbon cycle changes in the deglaciation. In comparison to earlier studies, Beeman et al. use a CO2 dataset (WAIS) that is better resolved in time, and better constrained in terms of its delta age uncertainty. In addition, they include new water isotope data from the WAIS divide ice core. Furthermore, they do not just estimate a mean lag between CO2 and temperature over the entire study period (e.g., Pedro et C1 al. 2012) but investigate the relative timing of change-points in both timeseries, similar to (Parrenin et al. 2013). Their results indicate that CO2 and temperature change synchronously except at the end of the Antarctic cold reversal (CO2 leads) and the onset of the Holocene (CO2 lags). The availability of the new high resolution CO2 data from WAIS and the approach to investigate change-points rather than a mean lag make this paper interesting and timely. The results will be a valuable contribution to the discussion of causes and effects of deglacial climate changes. However, I have some reservations about aspects of the manuscript and method that warrant further testing and/or justification.*

*General Comments: i) The authors use a stack of five Antarctic ice core records as their temperature estimate. Their argument is that this reduces local noise. While this may be true, it also removes local differences in the temperature sensitivity to CO2 and/or sea ice changes. For recent climate change, Jones et al. (2016) showed that the local temperature trends vary strongly across Antarctica, including cooling in some regions. If we use figure 1 in Jones et al. as a potential analogue, than 4 out of the 5 cores used by Beeman et al. fall into regions where temperature hasn't warmed or is cooling since 1979. Thus, from recent observations one can expect the response of Antarctic temperature to CO2 forcing to be spatially heterogeneous. Hence, instead of stacking the isotope records, I think it is more informative to determine the phase relationship for each ice core. Because: If we want to determine cause and effect between CO2 and Antarctic climate, it is the first robust temperature rise at any ice core that is informative and not their mean. Obviously, this will increase the noise in the temperature dataset, but ideally a change-point detection method can handle noise (see also specific comments). If all ice cores lead to internally consistent estimates of temperature change-points, their likelihood estimates can be combined to reduce the errors – but this consistency would first need to be shown. And in case of inconsistencies this needs to be discussed. The stacking may also influence the detection of fast temperature rises (such as around 16k) where already minor synchronization uncertainties and differences in resolution can lead to smoothing.*

We have calculated timings for each ice core for the revised manuscript (See figures 1-5). We greatly appreciate this suggestion, which we think significantly improves our study.

The timings for the temperature records are generally consistent. The features analyzed for the stack in the original versions of the manuscript are not absent from any of the individual records, though they are assigned different probabilities. However, there are several differences between the temperature records, perhaps indicating the regional character of certain events.

[Figure]

**Figures 1-5. Histograms made for all five cores included in the stack.**

These new results merit discussion, and we propose to include the following text:

In the introduction:

"Previous work on the relative timing of CO2 and Antarctic Temperature did not take into account the possible regional differences in climate. Differences between West Antarctic and East Antarctic temperature during the last deglaciation have been noted (Cuffey et al., 2016). On much shorter timescales, (Jones et al., 2016) note differing temperature trends at the drilling sites of the five cores used in this study over the period for which direct temperature observations exist (beginning in the mid-20th century). On the other hand, the interpretation of individual isotopic records can prove complicated, as local effects, including those of ice sheet elevation change and sea ice extent, are difficult to correct.

We provide change point timings and lead-lag estimates for the five individual isotope-derived records used in our stack as well."

In the results section:

"The change points for the five individual isotopic records are largely coherent with those identified for ATS2. However, some major differences do appear, and these differences merit discussion.

In the WAIS divide record, a small probability peak appears around 22ka. This peak does not surpass the 95% probability threshold (but does surpass the 90% threshold). We find it worth mentioning because it represents a much earlier beginning of temperature rise in the WAIS record. A stabilization after the 18ka temperature acceleration surpasses the 95% probability threshold at WAIS divide as well.

The temperature stabilization after the 14ka event is not significant in the WAIS record. Indeed, this stabilization is only a significant event in the EDC and Talos Dome records.

In the Dome Fuji record, the beginning of the temperature acceleration at the T1 onset appears to occur considerably before 18ka. We have no clear explanation for the subsequent temperature excursion. In the other four cores, this rise appears to occur more concurrently with the rise in CO2. It should be noted that the temporal resolution of the Dome Fuji temperature record is lower during the T1 onset than the other records.

Also of note in the Dome Fuji record is the abrupt jump in temperature around 14.7 ka, concurrent with a similar rise in CO2. No similar event is detected in any of the other regional temperature series.

Finally, a stabilization around 16ka is detected in the EDML core, concurrent with a similar stabilization in CO2. This stabilization is not detected in any of the other series. The probability of change around the ACR onset, on the other hand, does not pass the significance threshold in the EDML core."

In the conclusions:

"Some of the major millennial-scale changes in isotopic temperature records do not occur concurrently across all five cores. The T1 onset, ACR end, and Holocene onset do appear as significant changes in all five isotopic records. The ACR onset appears as a significant change in four out of five records. The temperature stabilization following the ACR onset appears in two

cores. Finally, the 14ka jump only appears in the Dome Fuji record, and the 16ka stabilization only in EDML.

The inter-core differences in detected events, particularly those that appear to correspond with CO2 variations, pose challenging questions. Could these events indicate a relationship between regional isotopic signals and the mechanisms of CO2 release and uptake? These questions will remain to be answered by future studies."

*The authors state throughout the manuscript that they detect abrupt temperature rises corresponding to the rapid CO2 rises discussed by Marcott et al. (2014). However, their method actually only detects the stabilization afterwards and not rises themselves (at least in temperature).*

This comment is correct, and any phrasing which implies that the beginning of a rise is detected where it does not surpass the 95 % threshold (see below) in the revised manuscript.

*In fact, this hints at another problem: The applied change-point detection method requires user input regarding how many change-points it is allowing for. The authors chose a number of 8 (and tested 7 for comparison). However, just by visual inspection of the CO2 record one can identify more change-points: i) the onset of the deglaciation, ii) the 16k rise, iii) the levelling off after 16k, iv) the rise at 14.6k, v) the levelling off after 14.3k, vi) the end of the ACR, vii) the jump into the Holocene and viii) the levelling off at the onset of the Holocene. Including the start and end point in the series which have to be marked as change points in the method, this yields 10 change-points. In fact, all of these points are discussed in the manuscript. This creates the inconsistency, that the method only allows for 8 (minus 2, i.e., start end) change points, but 10 (minus 2) are discussed – a solution that cannot be true in any of the realizations of the method: Some of these points must exclude each other if the maximum is set to 8. I think the authors should test their method allowing for more change points (e.g. 10).*

We may not have been clear enough about a subtle point of our method, which we think likely makes it less sensitive to the number of change points. When we analyze a time series with $n$ points, these points may be proposed anywhere in the time interval of the series, as long as the x-values increase monotonically. Then, all the accepted points are considered in the calculation of one probability distribution (rather than one distribution per point). Because a fit need not be perfect to be accepted in the Markov Chain Monte Carlo simulation, we may estimate more peaks of high probability than the number of points used in the linear representation. Thus, 10 peaks of probability for an 8-point simulation is not inconsistent. This will be emphasized in the revised manuscript.

We have performed a test using 10 change points, to be included in the supplementary materials for revised manuscript (Figure 6, below). This test shows convergence to approximately the same distribution as the 8-point test.

[Figure]

**Figure 6. 10-point test.**

*It is unclear, how the change-points that are actually discussed are chosen. The authors discuss the CO2 rise at 16k, which has a very low probability peak, but do not discuss the large probability peak of a CO2 increase at the onset of the ACR. How is this justified? I agree that the rise at 16k is the relevant one, but the method obviously implies a higher likelihood of a CO2 increase at 14.3k? What does this imply about the methods ability to detect the correct change points and infer their timing? The authors should discuss more clearly, how they evaluate the likelihood of a given point to be a change-point at all, before discussing its timing.*

We implement a probability threshold to discuss the significance of change points, and propose to include the following text on page 6, line 4, in addition to a figure :

 "To estimate the significance of change points, a probability threshold is implemented. For each of the histograms, more than 95\% of the bins have (normalized) values below 0.0004—the value we select for the threshold. This threshold does not evaluate significance in the sense of comparison with a null hypothesis. We perform such a test by creating linear fits using 8 random samples of the linear interpolations between data points of each series, and do not apply a Metropolis-Hastings type criterion, but rather accept all fits. The bin values of the resulting normalized histograms do not surpass 0.0002."

[Figure]

**Figure 7. 95 % confidence threshold at 0.0004 (dashed blue lines)**

See our discussion about the autocorrelation of residuals for more explanation of the large peak around 14.3 ka...

*Specific comments:*

*PP1,L13: "Multimodal timings" – what do you mean? Multimodal probability distributions of change point estimates? I suppose the real change cannot be multimodal.*

Omitted from the abstract.

*PP2, L6: "Consistently": Replace with "on average", since Shakun et al. do not discuss whether this is consistent over the entire time.*

Accepted.

*PP2, L8: Also global T did not increase continuously according to Shakun et al.*

This line has been changed to read:

"But Antarctic temperature and CO2 concentrations changed much more coherently as T1 progressed."

*PP2, L20-21: Please rephrase the colloquial "thanks to the so-called isotopic paleothermometer". Possibly: ". . . due to temperature dependent fractionation of water isotopes during condensation"?*

This line has been rephrased to "...due to the temperature dependent fractionation of water isotopes..."

*PP2, L24-25: Please replace the CO2 lock in depth with the corresponding estimate from WAIS and*

*a reference to (Buizert et al. 2015). Generally it is not clear throughout the manuscript, whether EDC CO2 is used at all, and if so, how it is spliced together with WAIS CO2. If it is not used, please shorten the methodological discussion of EDC CO2 as it is misleading the reader to believe EDC CO2 data was used too.*

Replaced with:

"However, the age of the air bubbles is younger than the age of the surrounding ice, since air is locked in at the base of the firn (on the order of 70 m below the surface on the West Antarctic Ice Sheet (WAIS) Divide) at the Lock-In Depth (LID) (Buizert and Severinghaus, 2016)."

EDC CO2 data were not used at all. The later line on P4:

"At EDC, Delta-depth, the depth shift between synchronous air and ice levels, is calculated using an estimate of the LID based on nitrogen-15 data (Parrenin et al., 2013) that assumes negligible convective zone height."

has been omitted.

*PP3, L10: d15N of N2 as a proxy for DZ height: Refer to (Buizert and Severinghaus 2016) who propose some uncertainty to this assumption?*

Buizert and Severinghaus (2016), do not propose a quantitative uncertainty, we propose to include the following line.

"However, the assumption that d15N reflects DZ height is imperfect, as it may underestimate the DZ height for sites with strong barometric pumping and layering (Buizert and Severinghaus, 2016), generally those closer to the coast."

PP4, L14: "stack": Please describe how this is generated, so that other people can reproduce this. Is this an average over all records? Are they resampled to equal resolution beforehand? Are they standardized or kept in degC? If kept in degC, which slope (per mille/degC) is used? Do all cores have similar amplitudes across the deglaciation (in degC or per mille) or can the stack be biased by single records with exceptional amplitude? Later on, the fitting procedure requires an error in K for each data-point: How is this derived? Does it include uncertainty from the isotope temperature conversion and other (e.g., circulation) influences on isotopes?

We now include:

"The individual isotopic records are converted to temperature (C) and are corrected for source temperature (Bintanja et al., 2013), resampled to a timestep of 20 years, and averaged. The standard deviation of the records at each timestep is assumed to be representative of the uncertainty concerning the conversion from isotopes to temperature, and of the uncertainty rooted in the geographic distribution of the stack."

The spreadsheet used to calculate the stack will be made publicly available.

*PP4, L17: "previously published ties": The list in (Parrenin et al. 2013) includes a lot of isotopic tie-points between the ice cores as well. Are these used here too? They introduce some circularity in the approach as they will reinforce the structure and timing of isotope (temperature) changes in the*

*ice core that is used as a target. Please clearly state, whether all tie-points are volcanic or not.*

We only use the volcanic tie points from Parrenin et al. (2013) + new EDC-DF and EDC-WD volcanic tie points. The tie-points themselves will be made available.

The line now reads

"We use previously published volcanic ties..."

*PP4, L17-18: Synchronization: Please provide a list of tie-points as well as the ECM data for both cores in the supplementary, so that people can reproduce the analysis.*

We will make these available in the supplementary materials and the appropriate paleoclimate databases.

*PP4, L23: "1,030 measurements": Earlier (PP3, L14) it is stated that there are only 320 points? Are these replicates? If so, please state this as it is slightly confusing.*

Rephrased to "1,030 measurements at 320 depths..."

*PP4, L27-28: "At EDC. . .": Again – is this used at all? If not, remove. If yes, elaborate how CO2 records of such different resolution are stacked.*

EDC was not used at all, this has been removed.

*PP5, Figure2 caption: "Ratio of the age difference between two consecutive tie points.." . . . and what? Not clear what is plotted here.*

Will rephrase to "Ratio of the age differences between two consecutive **pairs of** tie points..."

*Section 2.3 – 2.5: I encourage the authors to have a look at these sections again, and try to rewrite them more clearly. As it is now, it is near impossible to really understand what's going on. The authors elaborate on how the MH sampler is working. This may be nice for saving computing time, but hopefully doesn't affect the results. A reference could be enough?*

Only section 2.4 treats the MH sampler. Lines 15-23 are moved to the supplement.

The other two sections are important for reproducibility. Section 2.3 indicates how the goodness of individual fits to the time series is assessed, and section 2.5 details the formal calculation of leads and lags.

*At the same time the authors do not discuss more relevant aspects of the method: How does the method deal with irregular sampling resolution in the records. They just say it becomes less precise (PP6, L3), but is that really so, or can it become biased? Similarly, it is not discussed, how the uncertainty for CO2 or ATS is derived. Are the residuals/uncertainties treated as independent or correlated? Since the method only detects linear trends, any other internal climate variability would basically be a correlated uncertainty (i.e., red noise, as opposed to white measurement noise)?*

The residuals are corrected using the inverse of an autocorrelation matrix (we assume the residuals to be correlated), which is estimated after an initial sampling run. This is treated on P6, lines 0-5.

We add, on P6, line 5:

"Some limitations of this method should be made clear: first, where data are more sparse (i.e. in the CO2 series after the Holocene onset) change point identification becomes less precise and may be biased by the lack of data. Second, the method is not designed to treat variations that depart significantly from a linear shape, and accelerating trends can lead to noise in the histogram representations."

*PP7, L13: ". . .and corresponding rise in temperature around 16ka": Is that true? Looking at figure 3, there is no detection of a ATS increase around 16ka. Only the stabilization. In principle I agree, that there appears to be an ATS increase. However, there is a similar increase around 17kaBP in ATS without a corresponding change in CO2, implying that this may just be internal variability/noise. And in any case: The method does not detect either of these increases in ATS.*

Eliminated.

*PP7, L17: see previous comment. The ATS rise at 16k is not actually detected by the method.*

Eliminated.

*PP7, L18-21: Multimodality: This paragraph is written as if the CO2/ATS increase was multimodal. However, each realization of the method probably only picks one or the other mode as a possible change point, and never both. Hence, the multimodality reflects an uncertainty in the change-point identification, not the identification of two separate change-points. Is this correct? If so, please rephrase.*

[Figure]

**Figure 8. Two change points in a small region, 18-19 ka.**

Two change points can be detected, see the figure for reference. However, fits may also only contain one point.

The text is changed to:

"The deglaciation onset begins with a large, postive change point mode for Antarctic temperature,

centered around 18.08 ka. A second mode follows, at 17.70 ka. Similarly, there are two modes for the $CO_2$ series, centered at 17.63 ka and 17.30 ka. In both series, the two modes are upward-oriented. Investigation of the ensembles of fits indicates that the modes can represent an initial, more gradual increase in the rate of change, followed by a sharper acceleration in both series. However, since the probability peak here is continuous, we take it to be representative of a single change point."

*PP7, L22: "a small positive probability peak". The probability of this point being a change-point is very low, much less than for example the positive peak for $CO_2$ around 14.4ka which is not discussed. How do the authors choose which peak to discuss? How trustworthy is a change-point that is apparently only used in a small number of iterations?*

This is resolved by the implementation of the 95 % probability threshold. Reworded to:

"$CO_2$ began to change more quickly at around 16.15 ka, though we do not calculate a significant change point here. This rise abruptly peaked at 16.07 ka, and finally stabilized at 15.9 ka. These events are both identified by downward-oriented probability peaks."

*PP7, L24: See comment above. The ATS probability peak is very low. How reliable is the inference of a change point there (and its timing)? PP7, L 24-25: This section illustrates my concerns about how well the method deals with noise, and how subjectively some probability peaks are discussed while others are not. Why is the positive probability peak in $CO_2$ at 14.64ka discussed, while the bigger positive peak at 14.42 isn't? I agree that the rise at 14.64ka and the stabilization around 14.42ka are the relevant change points, but the statistical method doesn't. To me this highlights, that the method may underestimate noise in the $CO_2$ and ATS data, and hence, depict potentially erroneous change-points, which also have a seemingly high degree of certainty (in terms of timing). Please comment.*

The apparent rise at 14.64 ka indicates that the method attempts to treat the rapid rise as a step-change, which is probably not correct. Sensitivity tests to the autocorrelation matrix taken into account when estimating the residuals indicate that if the values of this matrix are artificially lowered, the step-change can be removed.

However, perhaps counter-intuitively, lowering the values of the autocorrelation matrix (which is normally estimated empirically) amounts to allowing the model to over-fit with respect to noise. The baseline values of the histograms become higher in the regions between peaks, and the peaks themselves lower, indicating that change points are allowed to represent what is essentially white noise.

Reworded to:

"A second abrupt $CO_2$ rise preceded the Antarctic Cold Reversal. During this rapid rise, $CO_2$ experienced major sub-centennial scale variations, and the corresponding probability peaks are noisy and large. Two narrow spikes in probability, one at 14.64 ka, and one at 14.42 ka, mark its beginning and end. The second peak is significant in both directions; investigation of the time series shows that a $CO_2$ jump of around 9 ppm occurs here in the data series."

*PP8, "leads and lags": Generally, I think the results should be more explicitly compared to (Parrenin et al. 2013), who applied a largely similar method to a different $CO_2$ (and slightly different ATS) dataset. Their estimates could be shown in figure 4 for comparison. Are the results*

*consistent for each change point?*

A new version of figure 4 will be included, and is shown here:

**Figure 9. Revised leads and lags**

*PP8, L2: ". . .it is not obvious." Change to: "our method doesn't detect it."*

Accepted.

*PP8, L7: "either the histogram peak around 12.9 or . . .". I don't think that you can interpret single peaks in the histogram like this. The timing of the rise in CO2 at the end of the ACR is detected as a broad probability distribution and not just by two minor peaks in the histogram. All values within the probability distribution are the possible "true" value with a given likelihood. Even if the peaks are the most likely single values, the cumulative probability of the true value not being these peaks is higher (cf. it is very unlikely to actually draw the exact mean value of a normal distribution).*

Accepted, these peaks are likely not distinct. Changed to:

"The ACR terminated with an increase in CO2, beginning at the peak occuring around 12.9-12.8 ka. ATS2 most likely began to increase at 12.66 ka."

*PP8, L6: "which loses data resolution.." See earlier comments: How does the method deal with this? PP8, L6-8: Again: Can all these modes really be interpreted in other terms than uncertainty? The authors mention "later peaks, where data resolution is lower, to be likely indicative of higher*

*frequency variability or noise". If the method cannot handle those, how good are the timing estimates for the other change-points? Please elaborate.*

The modes are still interpreted as method-related uncertainty, and are still included in the lead-lag calculation. The line "later peaks, where data resolution is lower, to be likely indicative of higher frequency variability or noise" is confusing, as it implies we do not include this period in the calculation. We omit this line.

*PP9, L1-2: "Applying the cross correlation operator. . .". How is it handled when the method indicates near equal probabilities of a CO2 rise and fall like around the ACR onset?*

The cross-correlation operator is applied only to the probability of a fall. We pick only the coherent direction.

The equality around the ACR onset is an error of scale, this will be changed in the figure.

Text changed to:

"The probability that one variable leads the other over a given time period can be calculated by applying the cross-correlation operator to the histograms of the two variables over a given time period and given direction (we make calculations only for the coherent change direction)"

*PP9, L13-14: See earlier comments on the reliability of the method and the handling of noise and resolution.*

*PP9, L14: "Calculating the phasing between 12-11.5. . ." How is this done? Are certain values excluded from the histogram for CO2? How?*

Rephrased to:

At the Holocene onset, a CO2 lag is certain. Calculating the phasing between 12.0 ka and 11.0 ka, we obtain an ATS2 lead of 574 +- 143 years. However, the decrease in data resolution here may bias the estimate.
We could consider the initial stabilization in both series to be coherent, and the following CO2 modes to be representative of bias resulting from the change in data resolution. Calculating the phasing between 12.0 ka and 11.5 ka, we obtain an ATS2 lead of 195 +- 62 years. Note that this timing appears as a minor mode of the phasing calculated between 12.0 and 11.0 ka. However, this calculation can only be performed making considerable assumptions about the data, and the estimate of 574 +- 143 is statistically more appropriate.

*PP10, L5-6: "We identify a coherent ATS2 change-point. . ." See general comments. I don't think this is the case.*

Accepted. Changed to:

We identify a CO2 change point not treated in these studies at 16 ka, associated with the centennial-scale rapid rise...

*PP 10, L11-12: "minor modes". See earlier comments. Can these really be interpreted?*

Changed to:

Additionally, many of the changes in CO2 are overlayed with centennial-scale substructures. The CO2 modes around the ACR onset are major and statistically significant.

*PP10, L27-29: "Mt. Takahe" I do not understand why the cumulative probability of the ATS2 change-point is relevant here. The method does not detect whether there are multiple change points there, but only a single one with a given uncertainty in timing. Mt. Takahe falls into the uncertainty range of the detected change point at that time. Correct?*

The cumulative probability is important when assessing whether one event occurred before or after another. Reviewer 2 comments:

"P10 line 22 to 29. McConnell et al suggested that accelerated warming was triggered by the Mt Takahe eruption. The finding here, that accelerated warming begins *before* the Mt. Takahe eruption, contradicts the McConnell hypothesis. The spin about "additional forcings beginning to accelerate warming before Takahe" is very unconvincing."

We agree that the eruption falls within the histogram, but it falls far to one side! (For a gaussian-shaped distribution, which this peak is not necessarily, we could calculate whether or not it falls outside the 1-sigma range).

*PP11, L1: "Here we confirm". See general comments. The method does not detect ATS rises coinciding with the rapid CO2 rises.*

Changed to: Here, we confirm that the ends of two of these events correspond with stabilizations in the Antarctic temperature record.

*PP12, L 3: "we identify change points". See earlier comments.*

Accepted. (Deleted)

*Technical Comments: PP2, L14: ". . . is thought to have. . ." (not "haved")*

Accepted.

*PP2, L19: ". . . and atmospheric composition. . ." (not "atmosphere")*

Accepted.

*PP4, L18: "Sigl et al. 2015": Change to 2016.*

Accepted.

*PP4, L17-18: "the offset in between ice and the air trapped much later at a given depth": Convoluted, please rephrase. Possibly: "The age difference between trapped gas and the surrounding ice matrix, delta age, . . ."*

Accepted.

*PP7, L22: "a small positive probability peak". Probability is always positive. Please rephrase.*

This peak does not meet our probability threshold, will be eliminated.

We could generally refer to downward-oriented and upward-oriented changes.

*PP9, L4: "2 sigma": The numbers in the following paragraph (PP9, L7-9) match the numbers in figure 4. However, the caption of figure 4 says this was 1 sigma? Please check.*

*PP9, L8: "At the peak of the 16ka rise". Since you do not detect the 16k rise in ATS a better formulation could be: "CO2 and ATS stop rising synchronously at the onset of the ACR" or similar.*

Accepted. Changed to:

"At the peak of the 16 ka CO2 rise, a rise is not detected with statistical certainty in the temperature series."

*PP10, L4: replace "in AMOC" with "from AMOC"*

"originate from" seems less correct than "originate in". Decision left to the editor.

**Reviewer 2.**

*Chowdhry Beeman et al. investigate the time relationship between Antarctic temperature and atmospheric CO2 during the last deglaciation. The question is of importance for our understanding of climate-carbon cycle feedbacks and has been tackled in multiple previous papers, most recently Parrenin et al 2013 and Pedro et al 2012. Chowdhry-Beeman et al. is distinguished from these previous studies by its use of the high time resolution and low delta-age WDC ice core CO2 record and a new regional Antarctic temperature stack. The consensus emerging from previous studies is that there is little-to-no significant time delay between CO2 rise and Antarctic temperature rise throughout most of the deglaciation. Chowdhry-Beeman et also find synchronous changes within uncertainties (excepting the Holocene onset), however they place more attention than others on centennial-scale signals in the temperature and CO2 series and their purported relationship. The techniques used to analyse of the phase relationship are more complex than used in previous studies. Although I have no reason to doubt the techniques, I do have some concern about their rather qualitative and selective interpretation. In particular the conclusion that there is a significant change in Antarctic temperature corresponding to the abrupt CO2 change around 16ka is not convincing. My overall impression is that the complex technique used to generate the PD histograms is out of proportion to their qualitative interpretation. Missing is some clear hypothesis testing and a more sceptical view of whether the centennial scale signals detected in ATS and CO2 are really meaningfully related. In general the approach appears very promising but the interpretation is still requiring quite some work. I would support publication after revisions to address the concerns below.*

*Major Comments Section 3.1: The technique applied to study the time relationship between ATS2 and WDC CO2 is more complex than previous studies and difficult (at least for this reviewer) to follow. My concern is that despite the complex approach the interpretation of phasing, in the end rests on a rather qualitative assessment of the change point histograms presented in Figure 3. Adding to my concern is that there is never a clear description of what makes a mode distinct and worthy of discussion or of the precise criteria for defining a significant change point.*

We propose to address this criticism as proposed by the reviewer, below.

*The caption of Fig 3 says that the y-axis range for the probability histograms is 0 to 0.0024. It's not clear to me precisely what is meant here.*

Probabilities are normalized, and thus the integrals of the histograms over the entire study period should sum to one. We will make this more clear in the caption.

*Can you show horizontal lines marking e.g. 0.05 probability cut-offs, which could then be used to judge which modes are significant?*

For each of the histograms (concave-up and concave-down change points for ATS2 and $CO_2$), more than 95% of the bins have (normalized) values below 0.0004—the value we select for the threshold.

Since defining a threshold does not evaluate significance in the sense of comparison with a null hypothesis, we perform a second test—we create linear fits by using 8 random samples of the linear interpolations between data points of each series, and do not apply a Metropolis-Hastings type criterion, but rather accept all fits. These fits should still approximate the shape of the series, but the points will not converge to major changes. The bin values of the resulting normalized histograms do not surpass 0.0002.

Of the events we discuss, this threshold notably eliminates the discussion of a possible temperature increase around 16ka. (See Figure 7, response to reviewer 1).

The following text is included:

" To estimate the significance of change points, a probability threshold is implemented. For each of the histograms, more than 95\% of the bins have (normalized) values below 0.0004—the value we select for the threshold. This threshold does not evaluate significance in the sense of comparison with a null hypothesis. We perform such a test by creating linear fits using 8 random samples of the linear interpolations between data points of each series, and do not apply a Metropolis-Hastings type criterion, but rather accept all fits. The bin values of the resulting normalized histograms do not surpass 0.0002."

*We are told (pp7 line 18) that the degalacial $CO_2$ rise features 'two modes' (17.63 ka and 17.30 ka) separated by a 'distinct anti-mode'. On what basis is the anti-mode distinct? Could this be over-interpretation of noisy data?*

The description of the antimode is removed from the text.

*Further down ( line 24) the authors describe 'a broad low probability peak' in ATS2 at 15.96 ka. It's not explained how 15.96 is selected as the centre of this peak or why this peak is considered significant, given there are similar amplitude peaks elsewhere in the deglacial $CO_2$ record that are not discussed at all. The same can be said for the 'small upward probability peak' in $CO_2$ at 16.15ka. This ambiguity about what is a significant feature and what is not continues throughout the section.*

This ambiguity is removed by the introduction of a probability threshold. This peak will no longer be discussed.

*To give another example (line pp8 line 7), the author's describe 'two larger modes in $CO_2$ at 11.12 and 11.01 ka.' as being 'indicative of higher-frequency variability or noise.' It's not clear on what basis these peaks are considered noise whereas the (smaller) peaks around 16 ka are considered*

*meaningful and related to ATS2. My overall impression is that the complex technique used to generate the PDs is out of proportion to their qualitative interpretation.*

The large modes in $CO_2$ at 11.12 and 11.02 ka are still included in the calculation of leads and lags. However, we cannot be certain if they are representative of individual changes, or rather of the severe reduction in the resolution of the $CO_2$ series at the onset of the Holocene; we thus refrain from further commentary. This section is now reworded to:

"At the Holocene onset, a CO2 lag is certain. Calculating the phasing between 12.0 ka and 11.0 ka, we obtain an ATS2 lead of 574 +-143 years. However, the decrease in data resolution here may bias the estimate.
We could consider the initial stabilization in both series to be coherent, and the following CO2 modes to be representative of bias resulting from from the change in data resolution. Calculating the phasing between 12.0 ka and 11.5 ka, we obtain an ATS2 lead of 195 +-62 years. Note that this timing appears as a minor mode of the phasing calculated between 12.0 and 11.0 ka. However, this calculation can only be performed making considerable assumptions about the data, and the estimate of 574 +- 143 is statistically more appropriate."

*I suggest the authors revise Section 3.1 to be shorter and more quantitative. I think part of the reason the some of the interpretation is unconvincing is that the authors do not appear to use their results to test any specific hypotheses. Instead we get a rather post-hoc interpretation of the leads and lags. Reframing the introduction to set up some specific hypotheses for testing could make the discussion more convincing.*

We accept this suggestion for the revised manuscript. The main working hypothesis is that CO2 and ATS2 are synchronous and coherent. An important hypothesis that can be tested is included in the comments of reviewer 1, with respect to regional differences in temperature series.

*Section 3.2: The methodology here appears good, however the section rests on the selection in Section 3.1 of five 'common' change points in ATS2 and CO2. As above its not clear by what criteria these 5 are selected. Please clarify.*

These points are now using a probability threshold, as mentioned above.

*p10 line 5. The significance of the ATS2 change point at 16k is not convincing. Please be more clear about the criteria for its selection over and above other peaks in the PDs that are not discussed at all.*

We are not convinced by this point either-it does not meet the probability threshold. This point will no longer be included.

*P10 line 11. " during the complex, centennial-scale changes associated with the 16 ka rapid rise and the ACR onset, ATS was most likely synchronous with CO2 ". This is not convincing given the +- 340 yr uncertainty and the ambiguity of the 16ka peak in ATS2. I'd suggest a more cautious interpretation: centennial scale variability in both series (possibly physically related, possibly not) restricts ability to make any clear statement on significant leads or lags during this interval.*

We agree with this interpretation, reworded to:

"However, during the complex, centennial-scale change at the ACR onset, ATS was most likely

synchronous with CO2..."

*P10 line 22 to 29. McConnell et al suggested that accelerated warming was triggered by the Mt Takahe eruption. The finding here, that accelerated warming begins \*before\* the Mt. Takahe eruption, contradicts the McConnell hypothesis. The spin about "additional forcings beginning to accelerate warming before Takahe" is very unconvincing.*

We accept this clearer rephrasing of our findings. However, it is important to note that our findings do not fully contradict the McConnell hypothesis (see reviewer 1's comment as well), but rather find it improbable to a certain degree. The line about additional forcings is eliminated, and the paragraph now reads:

"However, the cumulative probability of the ATS2 change point is much greater before 17.7 ka than after."

*p11 line 1. The authors claim here to 'confirm' an imprint on Antarctic temperature of ice berg discharge to the Sth Ocn \*and\* Nth Atlantic around 16ka". This is not convincing at all. First, as above, the ATS2 signal around16k is questionable given other similar sized peaks in the PDs that are not discussed. Second, as the authors well know, correlation in timing does not prove of a casual relationship. Third, what is the imprint supposed to be (warming, cooling, stabilization?) and how did the icebergs drive it? Revise.*

This paragraph is removed.

*p11 line 5. The 'reversal in phasing' between T1 and the ACR end is not convincing. The phasing at T1 is 292+-343 yrs (1 sigma!), thus spanning from CO2 lag to CO2 lead. How can the phasing reverse if it is not distinct at T1? A simpler interpretation is that the ATS2 and CO2 are roughly synchronous with the exact lead-lag varying between change points due to centennial scale variability in both series.*

We accept this interpretation. This now reads

"Though the T1 onset and the ACR end are both roughly synchronous, the most likely directionality of phasing is reversed.. These two events are structurally similar, and it has been postulated that both originate in AMOC reductions (Marcott et al., 2014). It appears that centennial-scale variability in the two series can modulate the timing of the effect of an AMOC change on CO2, Antarctic temperature, or both."

*p 11 line 9: "Centennial-scale variability may have been superimposed on coherent millenial scale trends, for example". Performing a similar analysis on band-pass filtered versions of the two series could be used to test this idea and would add a substantial new result.*

We propose to include the following in the results section:

"We apply a Savitsky-Golay filter designed to have a cutoff periodicity of approximately 500 years to the two series, resample the series to a 200 year timestep, and perform the fits again, assuming residuals to be uncorrelated because of the lower resolution. We identify change points in the same regions, though the distributions of these points are much broader and smoother. This indicates sensitivity to the centennial-scale variability superimposed on major millennial-scale changes (but not, importantly, to periodic centennial-scale variability in general)."

Figure 10, which represents this fit, will be included in the supplement.

[Figure]

**Figure 10. Fits to Savitsky-Golay filtered series.**

*p12 line 18. Comparison between east and west Antarctic temperature and CO2 could already be done by making an east Antarctic and west Antarctic stack. The authors might consider doing this in revisions, it would add a substantial new result to the lead and lag discussion.*

As suggessted by Reviewer 1, we have applied our method to the isotopic records from each ice core. See Figures 2-6 above.

*p10 line 5. It should be mentioned that within uncertainties the results are consistent with Parrenin et al and Pedro et al.*

Accepted. Now reads

"Within the range of uncertainty, our lead-lag estimates are consistent with those of Pedro et al. (2012) and Parrenin et al (2013). The addition of the WD paleotemperature record and removal of the Vostok record from ATS2, the updated atmospheric CO$_2$ dataset, and our more generalized methodology are all, in part, responsible for the differences in computed time delays (SI)."

*Figure 4. Important typo. I think the phasing at the ACR end should read \*-\*250 +- 188.*

Accepted, likewise for the ACR onset.

*P 11 line 11. It's very difficult to follow this sentence. Please revise*

Reword to :

Bauska et al. (2016), for example, hypothesize that an earlier rise 10 in CO2 at 12.9 ka, driven by land carbon loss or SH westerly winds, might have been superimposed on the millenial-scale trend.

*P12 line 3. " Notably, we identify change points in ATS2 that are associated with rapid rises in CO2." Which change points exactly? The previous paragraph comments that rapid change in CO2 and ATS2 around the ACR are not clearly in common. And my concerns remain about the significance of any signal in ATS2 around the rapid 16ka signal in CO2. Without further evidence this conclusion of related abrupt changes in not convincing and not justified to include as a major conclusion here or in the abstract.*

Accepted.

*P12 line 13-15: "This variability suggests complex mechanisms of coupling. ==Indeed, perhaps different mechanisms of ATS2 and CO2 rises, some coupled, others decoupled, were activated and deactivated (Bauska et al., 2016) throughout the deglaciation.==" This statement so encompasses all possibilities that it is almost meaningless*

The second sentence (highlighted) will be omitted.

*Please advise where the new Antarctic temperature stack will be made publicly accessible upon publication.*

The stack is already available on the linked github page (https://github.com/Jai-Chowdhry/LinearFit-2.0-beta/tree/v0.0) as ATS2-new-sigma2.txt. It will be made available on Pangaea/NOAA Paleoclimate upon publication.

---

## Author Response (AR1)

Dear editor, dear reviewers,

We have made substantial revisions to our manuscript, *Antarctic temperature and CO$_2$: near-synchrony yet variable phasing during the last deglaciation* following the two reviews and subsequent editorial comments.

A major concern shared by the two reviewers and the editor was the treatement of uncertainty in the methodology. One reviewer expressed concern that our method might overfit correlated noise in the data series. We have therefore revised our treatment of the inverse covariance matrix that weights residuals between the two series and the piecewise linear fits used to estimate change points. This covariance matrix is designed to account for autocorrelation between residuals, but we found several weaknesses in our previous estimate, which used interpolation to calculate an empirical covariance estimate on regular intervals. Our CO$_2$ series is irregularly sampled in time, and as such interpolation adds spurious information to the covariance estimate. Additionally, the empirical covariance estimate is not robust to outliers. We thus implement an AR(1) model designed for autocorrelated noise in unevenly spaced time series (Mudelsee, 2002) combined with a robust fitting approach (Chang and Politis, 2016). A test of this robust procedure shows it to appropriately model the noise in the CO$_2$ series.

The resulting change point timing histograms are located similarly to the previous estimates, but contain considerably less noise and are often broader. The new procedure is also applied to the fits of individual isotopic temperature records from the five cores used in the study, as requested by the reviewers. These histograms contain significantly less noise as well.

We consider this methodological correction to be rather important, and have thus replaced the old results in the manuscript with the new results. This constitutes a rather major change to the manuscript. As such, we reassess our point-by-point responses to the editor's and reviewers' major comments below, which we think are better answered using the new method (all minor comments have still been accepted).

Below these point-by-point responses is a marked-up version of the manuscript. In order to maintain readability, we do not mark-up minor changes, which have all been accepted unless the relevant text has been deleted entirely, but show all major changes (in red) and all changes to the text corresponding to the methodological change (in italics).

All the best, on behalf of all co-authors,

Jai Chowdhry Beeman

**10    Point-by-point responses**

**Editorial comments**

The wording stabilization is misleading, essentially you mean a significant kink in the record, i.e. a change in the slope. Please change wording here and throughout, where you used stabilization.

*The wording "stabilization" is removed when referring to a change in slope, and replaced with change in slope or change point.*

[With respect to Dome Fuji fits] This example clearly shows the limits of your methods and may be related to the strong
20  autocorrelation of the record in this time interval? Please choose a wording that clearly expresses these limitations.

*As mentioned above, autocorrelation in the records is treated more appropriately by our new formulation of the covariance matrix. In the individual case of the Dome Fuji record, the Deglaciation mode is now centered on 18ka, in accordance with the other records, in spite of the highly correlated data variation just afterward. The very high correlation here appears to*
25  *still not be completely accounted for, and is fitted by a narrow peak in probability, but large sub-millenial scale variations like these should indeed not be entirely smoothed when using our AR(1) modeled covariance matrix, which is designed to rather be robust to them. Since the Dome Fuji result is not significantly different from the others at T1 using the new fits, this discussion has been eliminated from the text.*

30  I wonder in how far the noise level of the individual records affects the results. It is clear that due to the different accumu-lation rates and the different sampling strategies, the noise levels are different in all temperature records. You may want to downsample the higher records to the DF resolution to allow for a fair comparison.

*Since a unique AR(1) model is used to make the covariance matrix for each record, varying noise levels should not greatly*
35  *affect the results. Under the previous formulation, we could expect the error due to interpolation during the covariance matrix calculation to be larger for low-noise, low-resolution records like Dome Fuji, Talos Dome and EDC, since the true amount of information at the lowest time step would be quite low. These records indeed appear to have the noisiest histograms in the older formulation, while the smoothness of the histograms in the newer formulation is relatively similar.*

*A limitation remaining in the new formulation is the treatment of very highly correlated residuals, treated as outliers by the*
5  *robust AR(1) model. In the case of the $CO_2$ series, these are the centennial-scale jumps at the ACR and Holocene onset, such a jump is also visible in the DF temperature record at the T1 onset. However, using a non-robust fitting of the AR(1) model would improperly account for the remaining correlations in the series, and thus bias the positions of the millenial scale change points.*

[The significance test]...is not clear...Please explain more thouroughly. ...explain in more detail why the longer phase appears
10  to be more correct in your opinion.

*Testing the significance of probabilities calculated in an ostensibly bayesian framework is not a standardly applied proce-dure. We can choose two approaches: comparing the probabilities with the probabilities resulting from a null hypothesis, or comparing the probabilities among themselves.*
15  *For the first approach, we choose the null hypothesis that any point chosen randomly from a uniform distribution is equally probable to be a change point. To correctly treat this hypothesis, we calculate the directionality (concave up/down) of slope changes at the change points for 1000 fits distributed uniformly in time, with y-values chosen by interpolating between data points in the series. None of the normalized histogram bins made this way for either series surpasses 0.0002.*

*For the second approach, we search for a probability value below which lie approximately 95 % of histogram bins for our fits. 94.9 % of the bins for our $CO_2$/ATS2 fits fall below 0.0003. Since this value is higher than the value calculated using the first approach, we use it to define our probability threshold.*

*Note that since the histograms are normalized, and in order to treat the series consistently, we do not caluclate different thresholds for the $CO_2$ or ATS2 fits, or according to directionality, but rather assess all histograms cumulatively.*

5     *We include the above note in the supplement; in the main text, it is treated as follows:*

    *We implement a probability threshold to select significant change points. 94.9% of the histogram bins have (normalized) values below 0.0003—the value we select for the threshold. This threshold does not, on the other hand, evaluate significance in the sense of comparison with a null hypothesis. A simple null hypothesis would be that the series is equally well-represented by segments placed anywhere on the interval, in time, with y-axis values approximately corresponding to the data. We randomly*

10     *generate 1000 points along the time intervals for both series, and calculate y-axis values for each point by linearly interpolating between data points at the respective x-value. We can thus create upward-facing and downward-facing histograms that reflect the approximate slopes of the series at any given time, but that effectively consider any change point timing to be appropriate. The bin values of the resulting normalized histograms do not surpass 0.0002. We choose the higher of these two estimates of significance.*

    [With respect to multimodality at the Holocene onset ...] ...does not meet the reveiwer concern. Please elaborate more carefully in the revised text.

    *The multimodality of this change point is reduced by the new covariance matrix formulation, but is still present. We treat it*

20     *in the text as follows : A second, broad mode, representing further methodological uncertainty about the timing of the change of the long-term, millenial scale trend. Our method cannot specify which mode better represents the change, and both must be considered.*

    *The figure with fits to the Savitsky-Golay filtered series is now included, as requested, in the main text.*

**Reviewer 1 Comments**

    Summary: Beeman et al. investigate the phase relationship between atmospheric CO2 concentration and Antarctic temperature during the last deglaciation. This question has been investigated various times over the last decades as it can illuminate

30    on the role of CO2 forcing and carbon cycle changes in the deglaciation. In comparison to earlier studies, Beeman et al. use a CO2 dataset (WAIS) that is better resolved in time, and better constrained in terms of its delta age uncertainty. In addition, they include new water isotope data from the WAIS divide ice core. Furthermore, they do not just estimate a mean lag between CO2 and temperature over the entire study period (e.g., Pedro et C1 al. 2012) but investigate the relative timing of change-points in both timeseries, similar to (Parrenin et al. 2013). Their results indicate that CO2 and temperature change synchronously except

5    at the end of the Antarctic cold reversal (CO2 leads) and the onset of the Holocene (CO2 lags). The availability of the new high resolution CO2 data from WAIS and the approach to investigate change-points rather than a mean lag make this paper interesting and timely. The results will be a valuable contribution to the discussion of causes and effects of deglacial climate changes. However, I have some reservations about aspects of the manuscript and method that warrant further testing and/or justification.

10    General Comments: i) The authors use a stack of five Antarctic ice core records as their temperature estimate. Their argument is that this reduces local noise. While this may be true, it also removes local differences in the temperature sensitivity to CO2 and/or sea ice changes. For recent climate change, Jones et al. (2016) showed that the local temperature trends vary strongly across Antarctica, including cooling in some regions. If we use figure 1 in Jones et al. as a potential analogue, than 4 out of the 5 cores used by Beeman et al. fall into regions where temperature hasn't warmed or is cooling since 1979. Thus, from recent observations one can expect the response of Antarctic temperature to CO2 forcing to be spatially heterogeneous. Hence, instead of stacking the isotope records, I think it is more informative to determine the phase relationship for each ice core. Because: If we want to determine cause and effect between CO2 and Antarctic climate, it is the first robust temperature rise at any ice core that is informative and not their mean. Obviously, this will increase the noise in the temperature dataset,

5    but ideally a change-point detection method can handle noise (see also specific comments). If all ice cores lead to internally consistent estimates of temperature change-points, their likelihood estimates can be combined to reduce the errors – but this consistency would first need to be shown. And in case of inconsistencies this needs to be discussed. The stacking may also influence the detection of fast temperature rises (such as around 16k) where already minor synchronization uncertainties and

differences in resolution can lead to smoothing.

*We have calculated timings for each ice core for the revised manuscript (See figure 5). We greatly appreciate this suggestion, which we think significantly improves our study.*

*The timings for the temperature records are generally consistent. However, there are several differences between the temperature records, as discussed in the text in section 3.2.*

The authors state throughout the manuscript that they detect abrupt temperature rises corresponding to the rapid CO2 rises discussed by Marcott et al. (2014). However, their method actually only detects the stabilization afterwards and not rises themselves (at least in temperature).

20 *This comment is correct, and any phrasing which implies that the beginnings of rapid rises in temperature are detected has been removed.*

In fact, this hints at another problem: The applied change-point detection method requires user input regarding how many change-points it is allowing for. The authors chose a number of 8 (and tested 7 for comparison). However, just by visual inspec-
25 tion of the CO2 record one can identify more change-points: i) the onset of the deglaciation, ii) the 16k rise, iii) the levelling off after 16k, iv) the rise at 14.6k, v) the levelling off after 14.3k, vi) the end of the ACR, vii) the jump into the Holocene and viii) the levelling off at the onset of the Holocene. Including the start and end point in the series which have to be marked as change points in the method, this yields 10 change-points. In fact, all of these points are discussed in the manuscript. This creates the inconsistency, that the method only allows for 8 (minus 2, i.e., start end) change points, but 10 (minus 2) are discussed –
30 a solution that cannot be true in any of the realizations of the method: Some of these points must exclude each other if the maximum is set to 8. I think the authors should test their method allowing for more change points (e.g. 10).

*We may not have been clear enough about a subtle point of our method, which we think likely makes it less sensitive to the number of change points. When we analyze a time series with n points, these points may be proposed anywhere in the time interval of the series, as long as the x-values increase monotonically. Then, all the accepted points are considered in*
5 *the calculation of one probability distribution (rather than one distribution per point). Because a fit need not be perfect to be accepted in the Markov Chain Monte Carlo simulation, we may estimate more peaks of high probability than the number of points used in the linear representation. Thus, 10 peaks of probability for an 8-point simulation is not inconsistent. This is emphasized in the revised manuscript in section 2.4, line 600.*

*We have performed a test using 10 change points, included in the supplementary materials for revised manuscript. This test*
10 *shows convergence to approximately the same distribution as the 8-point test.*

It is unclear, how the change-points that are actually discussed are chosen. The authors discuss the CO2 rise at 16k, which has a very low probability peak, but do not discuss the large probability peak of a CO2 increase at the onset of the ACR. How is this justified? I agree that the rise at 16k is the relevant one, but the method obviously implies a higher likelihood of a CO2
15 increase at 14.3k? What does this imply about the methods ability to detect the correct change points and infer their timing? The authors should discuss more clearly, how they evaluate the likelihood of a given point to be a change-point at all, before discussing its timing.

*We implement a probability threshold to justify our choice of change points, which is discussed in detail in section 2.4, lines*
20 *606-614.*

**Specific comments:**

PP1,L13: "Multimodal timings" – what do you mean? Multimodal probability distributions of change point estimates? I
25 suppose the real change cannot be multimodal.
*Omitted from the abstract.*

PP2, L6: "Consistently": Replace with "on average", since Shakun et al. do not discuss whether this is consistent over the entire time.

*Accepted.*

PP2, L8: Also global T did not increase continuously according to Shakun et al.

5    *This line has been changed to read: "But Antarctic temperature and $CO_2$ concentrations changed much more coherently as T1 progressed."*

PP2, L20-21: Please rephrase the colloquial "thanks to the so-called isotopic paleothermometer". Possibly: ". . . due to temperature dependent fractionation of water isotopes during condensation"?

*This line has been rephrased to "...due to the temperature dependent fractionation of water isotopes..."*

10    PP2, L24-25: Please replace the $CO_2$ lock in depth with the corresponding estimate from WAIS and a reference to (Buizert et al. 2015). Generally it is not clear throughout the manuscript, whether EDC $CO_2$ is used at all, and if so, how it is spliced together with WAIS $CO_2$. If it is not used, please shorten the methodological discussion of EDC $CO_2$ as it is misleading the reader to believe EDC $CO_2$ data was used too.

*Replaced with: "However, the age of the air bubbles is younger than the age of the surrounding ice, since air is locked in*
15    *at the base of the firn (on the order of 70 m below the surface on the West Antarctic Ice Sheet (WAIS) Divide) at the Lock-In Depth (LID) (Buizert and Severinghaus, 2016)."*

*EDC $CO_2$ data were not used at all. The later line on P4:*

*"At EDC, Delta-depth, the depth shift between synchronous air and ice levels, is calculated using an estimate of the LID based on nitrogen-15 data (Parrenin et al., 2013) that assumes negligible convective zone height."*

20    *has been omitted.*

PP3, L10: d15N of N2 as a proxy for DZ height: Refer to (Buizert and Severinghaus 2016) who propose some uncertainty to this assumption?

*Buizert and Severinghaus (2016), do not propose a quantitative uncertainty, we propose to include the following line. "However, the assumption that d15N reflects DZ height is imperfect, as it may underestimate the DZ height for sites with strong*
25    *barometric pumping and layering (Buizert and Severinghaus, 2016), generally those closer to the coast."*

PP4, L14: "stack": Please describe how this is generated, so that other people can reproduce this. Is this an average over all records? Are they resampled to equal resolution beforehand? Are they standardized or kept in degC? If kept in degC, which slope (per mille/degC) is used? Do all cores have similar amplitudes across the deglaciation (in degC or per mille) or can the stack be biased by single records with exceptional amplitude? Later on, the fitting procedure requires an error in K for each data-point: How is this derived? Does it include uncertainty from the isotope temperature conversion and other (e.g., circulation) influences on isotopes?

5    *We now include: "The individual isotopic records are converted to temperature (C) and are corrected for source isotopic variations (Bintanja et al., 2013), resampled to a timestep of 20 years, and averaged. The standard deviation of the records at each timestep is assumed to be representative of the uncertainty concerning the conversion from isotopes to temperature, and of the uncertainty rooted in the geographic distribution of the stack." The spreadsheet used to calculate the stack will be made publicly available.*

10    PP4, L17: "previously published ties": The list in (Parrenin et al. 2013) includes a lot of isotopic tie-points between the ice cores as well. Are these used here too? They introduce some circularity in the approach as they will reinforce the structure and timing of isotope (temperature) changes in the ice core that is used as a target. Please clearly state, whether all tie-points are volcanic or not.

*We only use the volcanic tie points from Parrenin et al. (2013) + new EDC-DF and EDC-WD volcanic tie points. The tie-points themselves will be made available.*

5    *The line now reads*

*"We use previously published volcanic ties..."*

PP4, L17-18: Synchronization: Please provide a list of tie-points as well as the ECM data for both cores in the supplementary, so that people can reproduce the analysis.

*We will make these available in the supplementary materials and the appropriate paleoclimate databases.*

10    PP4, L23: "1,030 measurements": Earlier (PP3, L14) it is stated that there are only 320 points? Are these replicates? If so, please state this as it is slightly confusing.

*Rephrased to "1,030 measurements at 320 depths..."*

PP4, L27-28: "At EDC. . .": Again – is this used at all? If not, remove. If yes, elaborate how $CO_2$ records of such different resolution are stacked.

*EDC was not used at all, this has been removed.*

PP5, Figure2 caption: "Ratio of the age difference between two consecutive tie points.." . . . and what? Not clear what is plotted here.

*Will rephrase to "Ratio of the age differences between two consecutive pairs of tie points..."*

Section 2.3 – 2.5: I encourage the authors to have a look at these sections again, and try to rewrite them more clearly. As it is now, it is near impossible to really understand what's going on. The authors elaborate on how the MH sampler is working. This may be nice for saving computing time, but hopefully doesn't affect the results. A reference could be enough?

*Only section 2.4 treats the MH sampler. Lines 15-23 are moved to the supplement.*

*The other two sections are important for reproducibility. Section 2.3 indicates how the goodness of individual fits to the time series is assessed, and section 2.5 details the formal calculation of leads and lags.*

At the same time the authors do not discuss more relevant aspects of the method: How does the method deal with irregular sampling resolution in the records. They just say it becomes less precise (PP6, L3), but is that really so, or can it become biased? Similarly, it is not discussed, how the uncertainty for $CO_2$ or ATS is derived. Are the residuals/uncertainties treated as independent or correlated? Since the method only detects linear trends, any other internal climate variability would basically be a correlated uncertainty (i.e., red noise, as opposed to white measurement noise)?

*The residuals are corrected using the inverse of an autocorrelation matrix, the treatment of which has been considerably modified. This is now discussed in detail in section 2.3.1*

PP7, L13: ". . .and corresponding rise in temperature around 16ka": Is that true? Looking at figure 3, there is no detection of a ATS increase around 16ka. Only the stabilization. In principle I agree, that there appears to be an ATS increase. However, there is a similar increase around 17kaBP in ATS without a corresponding change in $CO_2$, implying that this may just be internal variability/noise. And in any case: The method does not detect either of these increases in ATS.

*Eliminated.*

PP7, L17: see previous comment. The ATS rise at 16k is not actually detected by the method.

*Eliminated.*

PP7, L18-21: Multimodality: This paragraph is written as if the $CO_2$/ATS increase was multimodal. However, each realization of the method probably only picks one or the other mode as a possible change point, and never both. Hence, the multimodality reflects an uncertainty in the change-point identification, not the identification of two separate change-points. Is this correct? If so, please rephrase.

*Eliminated (due to new histograms)*

PP7, L22: "a small positive probability peak". The probability of this point being a change-point is very low, much less than for example the positive peak for $CO_2$ around 14.4ka which is not discussed. How do the authors choose which peak to discuss? How trustworthy is a change-point that is apparently only used in a small number of iterations?

*Eliminated*

PP7, L24: See comment above. The ATS probability peak is very low. How reliable is the inference of a change point there (and its timing)? PP7, L 24-25: This section illustrates my concerns about how well the method deals with noise, and how subjectively some probability peaks are discussed while others are not. Why is the positive probability peak in $CO_2$ at 14.64ka discussed, while the bigger positive peak at 14.42 isn't? I agree that the rise at 14.64ka and the stabilization around 14.42ka are the relevant change points, but the statistical method doesn't. To me this highlights, that the method may underestimate noise in the $CO_2$ and ATS data, and hence, depict potentially erroneous change-points, which also have a seemingly high degree of certainty (in terms of timing). Please comment. *A considerable amount of noise is removed from these probability spikes by the new treatment of the covariance matrix. Only two (significant) spikes remain to represent the rapid rise, and these are much lower in probability.*

*However, we must still make the choice of whether to include the 'tail' of the ACR onset in $CO_2$ in the calculation of the uncertainty. It arguably represents uncertainty about whether the change is best represented by a short (centennial-scale) line or a longer variation, and therefore should be included.*

PP8, "leads and lags": Generally, I think the results should be more explicitly compared to (Parrenin et al. 2013), who applied a largely similar method to a different $CO_2$ (and slightly different ATS) dataset. Their estimates could be shown in figure 4 for comparison. Are the results consistent for each change point?

*A new version of figure 4 is included (now figure 6*

PP8, L2: ". . .it is not obvious." Change to: "our method doesn't detect it."

*Accepted.*

PP8, L7: "either the histogram peak around 12.9 or . . .". I don't think that you can interpret single peaks in the histogram like this. The timing of the rise in $CO_2$ at the end of the ACR is detected as a broad probability distribution and not just by two minor peaks in the histogram. All values within the probability distribution are the possible "true" value with a given likelihood. Even if the peaks are the most likely single values, the cumulative probability of the true value not being these peaks is higher (cf. it is very unlikely to actually draw the exact mean value of a normal distribution).

*Removed as a result of new fits*

PP8, L6: "which loses data resolution.." See earlier comments: How does the method deal with this? PP8, L6-8: Again: Can all these modes really be interpreted in other terms than uncertainty? The authors mention "later peaks, where data resolution is lower, to be likely indicative of higher frequency variability or noise". If the method cannot handle those, how good are the timing estimates for the other change-points? Please elaborate.

*This comment is still relevant in spite of the new fits. The multiple modes should be interpreted as methodological uncertainty.*

PP9, L1-2: "Applying the cross correlation operator. . .". How is it handled when the method indicates near equal probabilities of a $CO_2$ rise and fall like around the ACR onset?

*The cross-correlation operator is applied only in the coherent direction.*

*The equally large, bi-directional spikes at the ACR onset are not present in the new fits.*

PP9, L13-14: See earlier comments on the reliability of the method and the handling of noise and resolution.

PP9, L14: "Calculating the phasing between 12-11.5. . ." How is this done? Are certain values excluded from the histogram for $CO_2$? How?

*We no longer separate probabilities at the Holocene Onset, but consider the two modes together.*

PP10, L5-6: "We identify a coherent ATS2 change-point. . ." See general comments. I don't think this is the case.

*Accepted. The probability in ATS2 no longer appears.*

PP 10, L11-12: "minor modes". See earlier comments. Can these really be interpreted?

*Changed to:*

*[Changes in $CO_2$] are overlayed with centennial-scale substructures.*

PP10, L27-29: "Mt. Takahe" I do not understand why the cumulative probability of the ATS2 change-point is relevant here. The method does not detect whether there are multiple change points there, but only a single one with a given uncertainty in timing. Mt. Takahe falls into the uncertainty range of the detected change point at that time. Correct?

*The cumulative probability is important when assessing whether one event occurred before or after another. In the new fits, the Takahe eruption occurs more clearly after the temperature rise.*

PP11, L1: "Here we confirm". See general comments. The method does not detect ATS rises coinciding with the rapid $CO_2$ rises.

*Changed to: Here, we confirm that the ends of two of these events correspond with stabilizations in the Antarctic temperature record.*

PP12, L 3: "we identify change points". See earlier comments.

*Accepted. (Deleted)*

Technical Comments: PP2, L14: ". . . is thought to have. . ." (not "haved")

*Accepted.*

PP2, L19: ". . . and atmospheric composition. . ." (not "atmosphere")

*Accepted.*

PP4, L18: "Sigl et al. 2015": Change to 2016.

*Accepted.*

PP4, L17-18: "the offset in between ice and the air trapped much later at a given depth": Convoluted, please rephrase. Possibly: "The age difference between trapped gas and the surrounding ice matrix, delta age, . . ."

*Accepted.*

PP7, L22: "a small positive probability peak". Probability is always positive. Please rephrase.

*We now refer to downward-oriented and upward-oriented changes.*

5    PP9, L8: "At the peak of the 16ka rise". Since you do not detect the 16k rise in ATS a better formulation could be: "$CO_2$ and ATS stop rising synchronously at the onset of the ACR" or similar.

*Accepted (reworded since there is no ACR change probability at 16k in the new fits)*

PP10, L4: replace "in AMOC" with "from AMOC"

*Accepted*

**Reviewer 2 Comments**

Chowdhry Beeman et al. investigate the time relationship between Antarctic temperature and atmospheric CO2 during the last deglaciation. The question is of importance for our understanding of climate-carbon cycle feedbacks and has been tackled in multiple previous papers, most recently Parrenin et al 2013 and Pedro et al 2012. Chowdhry-Beeman et al. is distinguished from these previous studies by its use of the high time resolution and low delta-age WDC ice core CO2 record and a new regional Antarctic temperature stack. The consensus emerging from previous studies is that there is little-to-no significant time

5    delay between CO2 rise and Antarctic temperature rise throughout most of the deglaciation. Chowdhry-Beeman et also find synchronous changes within uncertainties (excepting the Holocene onset), however they place more attention than others on centennial-scale signals in the temperature and CO2 series and their purported relationship. The techniques used to analyse of the phase relationship are more complex than used in previous studies. Although I have no reason to doubt the techniques, I do have some concern about their rather qualitative and selective interpretation. In particular the conclusion that there is a

10   significant change in Antarctic temperature corresponding to the abrupt CO2 change around 16ka is not convincing. My overall impression is that the complex technique used to generate the PD histograms is out of proportion to their qualitative interpretation. Missing is some clear hypothesis testing and a more sceptical view of whether the centennial scale signals detected in ATS and CO2 are really meaningfully related. In general the approach appears very promising but the interpretation is still requiring quite some work. I would support publication after revisions to address the concerns below.

Major Comments Section 3.1: The technique applied to study the time relationship between ATS2 and WDC CO2 is more complex than previous studies and difficult (at least for this reviewer) to follow. My concern is that despite the complex approach the interpretation of phasing, in the end rests on a rather qualitative assessment of the change point histograms presented in Figure 3. Adding to my concern is that there is never a clear description of what makes a mode distinct and worthy of dis-

20   cussion or of the precise criteria for defining a significant change point.

*We address this criticism as proposed by the reviewer, below.*

The caption of Fig 3 says that the y-axis range for the probability histograms is 0 to 0.0024. It's not clear to me precisely what is meant here.

*Probabilities are normalized, and thus the integrals of the histograms over the entire study period should sum to one.*

Can you show horizontal lines marking e.g. 0.05 probability cut-offs, which could then be used to judge which modes are significant?//

*For each of the histograms (concave-up and concave-down change points for ATS2 and CO2), 94.9% of the bins have*

10   *(normalized) values below 0.0003—the value we select for the threshold. This is discussed further in the response to the editor and in Section 2.4, lines 10-20//*

We are told (pp7 line 18) that the degalacial CO2 rise features 'two modes' (17.63 ka and 17.30 ka) separated by a 'distinct anti-mode'. On what basis is the anti-mode distinct? Could this be over-interpretation of noisy data?

*The description of the antimode is removed from the text.*

Further down ( line 24) the authors describe 'a broad low probability peak' in ATS2 at 15.96 ka. It's not explained how 15.96 is selected as the centre of this peak or why this peak is considered significant, given there are similar amplitude peaks elsewhere in the deglacial CO2 record that are not discussed at all. The same can be said for the 'small upward probability peak' in CO2 at 16.15ka. This ambiguity about what is a significant feature and what is not continues throughout the section.

*Because of the probability threshold and the new covariance matrix, this point is no longer included*

To give another example (line pp8 line 7), the author's describe 'two larger modes in CO2 at 11.12 and 11.01 ka.' as being 'indicative of higher-frequency variability or noise.' It's not clear on what basis these peaks are considered noise whereas the (smaller) peaks around 16 ka are considered meaningful and related to ATS2. My overall impression is that the complex technique used to generate the PDs is out of proportion to their qualitative interpretation.

*With the new covariance matrix, the Holocene onset only has two significant modes in the $CO_2$ series. We accept the reviewer's interpretation, and treat these modes equally.*

I suggest the authors revise Section 3.1 to be shorter and more quantitative. I think part of the reason the some of the interpretation is unconvincing is that the authors do not appear to use their results to test any specific hypotheses. Instead we get a rather post-hoc interpretation of the leads and lags. Reframing the introduction to set up some specific hypotheses for testing could make the discussion more convincing.

35

*We accept this suggestion for the revised manuscript. The main working hypothesis was that CO2 and ATS2 are synchronous and coherent. An important hypothesis that is now tested is included in the comments of reviewer 1, with respect to regional differences in temperature series.*

Section 3.2: The methodology here appears good, however the section rests on the selection in Section 3.1 of five 'common' change points in ATS2 and CO2. As above its not clear by what criteria these 5 are selected. Please clarify.

*The (now four) change points are now using a probability threshold, as mentioned above.*

p10 line 5. The significance of the ATS2 change point at 16k is not convincing. Please be more clear about the criteria for its selection over and above other peaks in the PDs that are not discussed at all.

*We are not convinced by this point either-it does not meet the probability threshold, particularly with the new covariance matrix. It is no longer discussed, we state rather that our method does not show coupling at 16 ka.*

P10 line 11. " during the complex, centennial-scale changes associated with the 16 ka rapid rise and the ACR onset, ATS

15 was most likely synchronous with CO2 ". This is not convincing given the +- 340 yr uncertainty and the ambiguity of the 16ka peak in ATS2. I'd suggest a more cautious interpretation: centennial scale variability in both series (possibly physically related, possibly not) restricts ability to make any clear statement on significant leads or lags during this interval.

*We agree with this interpretation.*

P10 line 22 to 29. McConnell et al suggested that accelerated warming was triggered by the Mt Takahe eruption. The finding here, that accelerated warming begins *before* the Mt. Takahe eruption, contradicts the McConnell hypothesis. The spin about "additional forcings beginning to accelerate warming before Takahe" is very unconvincing.

25 *We accept this clearer rephrasing of our findings. With the covariance-adjusted fits, the probability that warming began after the eruption is even lower*

p11 line 1. The authors claim here to 'confirm' an imprint on Antarctic temperature of ice berg discharge to the Sth Ocn *and* Nth Atlantic around 16ka". This is not convincing at all. First, as above, the ATS2 signal around16k is questionable given other similar sized peaks in the PDs that are not discussed. Second, as the authors well know, correlation in timing does not prove of a casual relationship. Third, what is the imprint supposed to be (warming, cooling, stabilization?) and how did the icebergs drive it? Revise.

*This paragraph is removed.*

p11 line 5. The 'reversal in phasing' between T1 and the ACR end is not convincing. The phasing at T1 is 292+-343 yrs (1 sigma!), thus spanning from CO2 lag to CO2 lead. How can the phasing reverse if it is not distinct at T1? A simpler interpretation is that the ATS2 and CO2 are roughly synchronous with the exact lead-lag varying between change points due to centennial scale variability in both series.

*We accept this interpretation, though the T1 phasing is now more marked. This now reads*
*"Though the T1 onset and the ACR end are both thought to originate in AMOC reductions (Marcott et al., 2014), our results allow for the directionality of $CO_2$-ATS2 phasing to be reversed during the two events. $CH_4$ changes nearly synchronously with $CO_2$ at both points, but the phasings are opposite in direction and different in magnitude."*

p 11 line 9: "Centennial-scale variability may have been superimposed on coherent millenial scale trends, for example". Performing a similar analysis on band-pass filtered versions of the two series could be used to test this idea and would add a substantial new result.

*We include a figure of Savitsky-Golay filtered data, and the corresponding fits (Figure 4), which is discussed in detail in section 3.1, lines 10-17.*

p12 line 18. Comparison between east and west Antarctic temperature and CO2 could already be done by making an east Antarctic and west Antarctic stack. The authors might consider doing this in revisions, it would add a substantial new result to the lead and lag discussion.

*As suggessted by Reviewer 1, we have applied our method to the isotopic records from each ice core. These results are shown in Figure 5, and discussed extensively in section 3.2*

p10 line 5. It should be mentioned that within uncertainties the results are consistent with Parrenin et al and Pedro et al.

*With the new fits, this is not always the case. This is rephrased to "Within the range of uncertainty, our lead-lag estimates are only roughly consistent with those of Pedro et al. (2012) and Parrenin et al. (2013)"*

Figure 4. Important typo. I think the phasing at the ACR end should read *-*250 +- 188.

*Accepted, changed in the new version of this figure as well.*

P 11 line 11. It's very difficult to follow this sentence. Please revise

*Reworded to: Bauska et al. (2016), for example, hypothesize that an earlier rise 10 in CO2 at 12.9 ka, driven by land carbon loss or SH westerly winds, might have been superimposed on the millenial-scale trend.*

P12 line 3. " Notably, we identify change points in ATS2 that are associated with rapid rises in CO2." Which change points exactly? The previous paragraph comments that rapid change in CO2 and ATS2 around the ACR are not clearly in common. And my concerns remain about the significance of any signal in ATS2 around the rapid 16ka signal in CO2. Without further

470    evidence this conclusion of related abrupt changes in not convincing and not justified to include as a major conclusion here or in the abstract.

*Accepted.*

475    P12 line 13-15: "This variability suggests complex mechanisms of coupling. Indeed, perhaps different mechanisms of ATS2 and $CO_2$ rises, some coupled, others decoupled, were activated and deactivated (Bauska et al., 2016) throughout the deglaciation." This statement so encompasses all possibilities that it is almost meaningless

*The second sentence is omitted.*

480

Please advise where the new Antarctic temperature stack will be made publicly accessible upon publication.

*The stack is already available on the linked github page (https://github.com/Jai-Chowdhry/LinearFit-2.0-beta/tree/v0.0) as ATS2-new-sigma2.txt. It will be made available on Pangaea/NOAA Paleoclimate upon publication.*

485

[revised manuscript text omitted]
 $\boldsymbol{K}$ and the model covariance matrix $\boldsymbol{C}_{mod}$ as follows:*

$$\boldsymbol{C}_{mod} = \sigma_{mod}^2 \, \boldsymbol{K} \, ; \, \boldsymbol{K}_{ij} = a^{t_j - t_i} \tag{4}$$

*where $\sigma_{mod}^2$ is the variance of the modeling error, assumed constant and estimated using a robust estimator as $(IQR(\boldsymbol{R})/1.349)^2$. Finally, the covariance matrix of the residuals $C$ is calculated as:*

$$\boldsymbol{C} = \boldsymbol{C}_{mod} + \boldsymbol{C}_{meas}. \tag{5}$$

*Rather than inverting the covariance matrix, we use Cholesky and LU decompositions to solve for the cost function value $J$, as in Parrenin et al. (2015).*

**2.4 Estimating the posterior probability density**

In general, the probability density of the change points cannot be assumed to follow any particular distribution, as short-timescale variations of the time series may lead to multiple modes or heavy tails, for example. Thus, stochastic methods, which are best adapted to exploring general probability distributions (for example, Tarantola (2005)), are suited to our problem.

To tackle the large computation time required for traditional MH sampling, we apply the ensemble sampler developed by Goodman and Weare (2010) (GW) as implemented in the python emcee library (Foreman-Mackey et al., 2013). This sampler adapts the MH algorithm so that multiple model walkers can explore the probability distribution at once, making the algorithm parallelizable. It has the advantage of being affine invariant: that is, steps are adapted to the scale of the posterior distribution in a given direction.

We make histograms of the probable timings of 8 major change points for the WD $CO_2$ and ATS2 series. The choice of 8 points is not entirely arbitrary: it reflects our goal of investigating millenial-scale variability (8 points allows for approximately one point per two millenia over the study period). This choice is subjective, but our method is not particularly sensitive to the number of change points. Since fits need not be perfect to be accepted in the stochastic simulation, we may estimate more or less peaks of high probability than the number of points used in the linear representation, and probability distributions of simulations with 8 and 10 points are rather similar (Supplementary Materials). The results of the 8-point simulation are shown in Figure 3. The most probable timings are identified by probability peaks, or modes. We avoid comparing incoherent modes by separating changes by the sign of the second derivative of the fits. Further details of the simulations are given in the supplement.

*We implement a probability threshold to select significant change points. 94.9% of the histogram bins have (normalized) values below 0.0003—the value we select for the threshold. This threshold does not, on the other hand, evaluate significance in the sense of comparison with a null hypothesis. A simple null hypothesis could be that the series are equally well-represented by segments placed anywhere on the interval, in time, with y-axis values approximately corresponding to the data. We randomly generate 1000 points along the time intervals for both series, and calculate y-axis values for each point by linearly interpolating between data points at the respective x-value. We can thus create upward-facing and downward-facing histograms that reflect the approximate slopes of the series at any given time, but that effectively consider any change point timing to be appropriate. The bin values of the resulting normalized histograms do not surpass 0.0002. We choose the higher of these two estimates of significance, and consider peaks to be significant when the threshold is approximately met (within $\pm 2.5 \cdot 10^{-5}$) or clearly passed. The choice of this threshold over the lower threshold is ultimately arbitrary, and allowing for a buffer zone around the threshold reflects both this arbitrariness and the small uncertainty resulting from the stochastic nature of the histograms.*

**2.5 Phasing**

We estimate $\rho_{lead}^{ATS2}$, the probability that ATS2 leads $CO_2$ over a given interval, as

$$\rho_{lead}^{ATS2} = (\rho_x^{ATS2} \circ \rho_x^{CO_2}) \star \rho^{chron}, \tag{6}$$

where $\rho_x^{ATS2}$ is the probability of a change point at time $x$ for ATS2, $\rho_x^{CO_2}$ is the probability of a change point at time $x$ for $CO_2$, $\circ$ is the cross-correlation operator, which is used to calculate the probability of the difference between two variables, and $\star$ is the convolution operator, which is used to calculate the probability of the sum of two variables. $\rho^{chron}$ is the probability distribution of the chronological uncertainty between the two records, which we take to be Gaussian centered on 0, with standard deviation $\sigma = \sigma_{chron}$ (shown in Figure 3). The intervals associated with each change point are given in Figure 6.

[Figure]

**Figure 3.** Upper panel: Atmospheric $CO_2$ (black) and ATS2 (red) placed on a common time scale, with the normalized histograms of probable change points (8 points). Histograms are plotted downward-oriented when the rate of change decreases and upward-oriented when it increases (same colors, y-axis not shown) Rrobabilities are normalized so that the integrated probability for a given histogram sums to 1, and range from 0 (center) to 0.004 (top/bottom). The 0.0003 probability threshold is marked by dotted blue lines. In four distinct time intervals, both series show concurrent probable change points. Lower panel: Chronological uncertainty, taken as the sum of the Δage uncertainties and the uncertainty estimate for our volcanic synchronization.

690 ## 3 Results and discussion

**3.1 Change point timings**

*The change point histograms for the ATS2 and $CO_2$ time series in Figure 3 confirm that the millenial-scale changes in the two series were largely coherent. We identify four major changes in trend which surpass the 95% confidence interval for both series: the onset of the deglaciation from 18.2 to 17.2 ka B1950; the onset of the Antarctic Cold Reversal (ACR) at around 14.5*
695 *ka, the ACR end beween 12.9 and 12.65, and the Holocene onset, at approximately 11.5 ka. For each of these four changes, we calculate the probability of a lead or lag. Two additional change points, one for $CO_2$ centered at approximately 16 ka, and a second change point for the temperature series after the ACR onset, centered at 14 ka, are also significant but do not have significant counterparts in the other series. Two abrupt, centennial-scale rises in $CO_2$, one at the ACR onset and before the Holocene onset, are visible in the histograms as narrow peaks. These changes have been identified in the WD $CO_2$ record by*
700 *Marcott et al. (2014), though the beginning of a third such rapid change before 16ka is not detected here.*

*The deglaciation onset begins with a large, postive change point mode for Antarctic temperature, centered around 18.1 ka. The corresponding change point for the $CO_2$ series is centered around 17.6 ka.*

*The $CO_2$ rise peaks at around 16 ka, identified by a downward-oriented probability peak, which has no significant counter-part in the temperature series. This peak is followed by a brief plateau in $CO_2$ concentrations, before a gradual, accelerating resumption of the increase.*

*An abrupt $CO_2$ rise preceded the Antarctic Cold Reversal. Two narrow spikes in probability, upward- and downward-facing, near 14.58 ka, appear to represent this rise, the downward peak just reaching the significance threshold. The broad tail of the downward-facing $CO_2$ change point peaks again at around 14.35 ka. On its own, the second mode does not reach the significance threshold, but it appears to reflect further methodological uncertainty with respect to the timing of the millenial-scale change in $CO_2$. An unambiguous negative temperature change also occurs at around 14.35 ka, roughly concurrent with the downward $CO_2$ change point. Antarctic temperature began to descend rapidly after the ACR onset, finally stabilizing at the concave-up change point identified by the mode centered on 14.02 ka. No corresponding change point is detected for $CO_2$.*

*The ACR termination is represented by significant modes in both series. An increase in $CO_2$ began at the peak occuring around 12.88 ka, while the ATS2 increase is centered at 12.65 ka, appproximately.*

*The Holocene onset is well-defined in the ATS2 series, with a large mode centered at 11.7 ka. The probability peaks in the $CO_2$ series are remarkably similar to those identified at the ACR onset. A rapid rise in $CO_2$ is represented by narrow peaks from 11.57 to 11.53 ka. A second, broad mode, representing further methodological uncertainty about the timing of the change of the long-term, millenial scale trend. Our method cannot specify which mode better represents the change, and both must be considered.*

*As a second test of the timings of millenial-scale events, we use our method to fit filtered versions of the ATS2 and WAIS Divide $CO_2$ data. A Savitsky-Golay filter, designed to have an approximate cutoff periodicity of 500 years, is applied to the two records. In the two filtered series, the sub-millenial scale AR(1) noise present in the original series should be essentially removed. As such, fitting change points to these two series, assuming the residuals to be uncorrelated, provides a second form of verification of the appropriateness of the covariance matrix we use to fit the raw data.*

*Figure 4 shows the Savitsky-Golay filtered $CO_2$ and ATS2 time series, and the corresponding change point histograms. The four major changes identified in both series, at the T1 onset, the ACR onset, the ACR end, and Holocene onset, are similar in shape and center to the fits of the raw data. However, there are two notable differences between the two fits. First, the spikes representing centennial-scale $CO_2$ rises before the ACR and Holocene onsets are entirely removed. This is not surprising, given that the Savitsky-Golay filters are designed to remove all variability with periodicities less than 500 years, whereas the covariance matrix applied to the fits of the raw data only treats AR(1) correlated noise. Finally, the probability of the post-ACR change in ATS2 is considerably smaller for the filtered series. Savitsky-Golay filtering has its own drawbacks–data reinterpolation is required, for example, and propagating measurement uncertainty becomes difficult. However, the similarity of the two results supports our fits of the raw data.*

[Figure]

**Figure 4.** Upper panel: Savitsky-Golay filtered atmospheric $CO_2$ (black) and ATS2 (red) placed on a common time scale, with the normalized histograms of probable change points (8 points). Histograms are plotted downward-oriented when the rate of change decreases and upward-oriented when it increases (same colors, y-axis not shown, probabilities range from 0 (center) to 0.0024 (top/bottom)). The 0.0003 probability threshold is marked by dotted blue lines for reference, but is not applied here. Lower panel: Chronological uncertainty, taken as the sum of the $\Delta$age uncertainties and the uncertainty estimate for our volcanic synchronization.

**3.2 Change point timings for individual temperature records**

735 *Fits of each of the regional temperature records, corrected for source isotopic variations (Bintanja et al., 2013) are shown in figure 5. These fits should still be interpreted cautiously, as additional information included in the isotopic records used to calculate temperature–the signal of ice sheet elevation change, for example–are not corrected for. The comparison of these fits provides an initial, exploratory picture of potential regional differences in climate change during the last termination.*

*Of the four changes identified as coherent between the temperature stack and $CO_2$, those at the deglaciation onset, the ACR*
740 *end, and the Holocene onset are expressed as significant probability peaks in all five records. Some ambiguity appears to exist about the timing of the ACR onset in the EDML record. It is expressed by a rather broad, non-significant probability mode extending between 16ka and 14ka, though a significant spike at 14ka marks the downturn seen in the other records. The ACR onset is significant and well-defined in all of the five other records.*

*The WAIS Divide record is, notably, the only isotopic record in our stack from the West Antarctic ice sheet. We could thus*
745 *reasonably expect it to show considerably different trends from the other records. Indeed, a significant change in the WD temperature record occurs at 22 ka. This early change in the isotopic record was identified and confirmed to indeed be a*

[Figure]

**Figure 5.** Atmospheric $CO_2$ (black) and source-corrected temperature records (red) placed on a common time scale, with the normalized histograms of probable change points (8 points) for each ice core used in the ATS2 stack; the locations of the drill sites are shown in the center. Details of the histogram plots are as in Figure 3.

*temperature signal by Cuffey et al. (2016) using a borehole temperature record, though their study places the change at 21 ka. We confirm that the onset of the deglacial temperature rise in West Antarctica likely began as much as 4 ka before the onset of temperature rise in East Antarctica. Interestingly, the WD record also shows a temperature change point around 17.8 ka, ex-*

750 *pressed slightly later than in the other records and more synchronous with $CO_2$. This apparent acceleration of the temperature rise is followed by a significant downward-facing change point not seen in any of the other records. A difference appears to exist in timing at the Holocene onset as well, with temperature change at WD apearing to slightly precede temperature in the East Antarctic records, and the DF temperature change in particular appearing to occur more synchronously with $CO_2$.*

**3.3  Leads and lags**

755  The probability densities of leads and lags at the coherent change points between ATS2 and $CO_2$ are shown in Figure 6. We then report the 1 $\sigma$ standard deviation of the lead/lag, but this estimate must be applied with care where the lead probability is still multimodal, as is the case at the Holocene and ACR onsets.

[Figure]

**Figure 6.** Probability density $\rho$ (y-axis, normalized) of an ATS lead (x-axes, in years) at each of the selected change point intervals (noted on subfigures). Negative x-axis values indicate a $CO_2$ lead. In the text in each box, the name of the period, the time period in which the lead is calculated, the mean and standard deviation of the lead/lag density ($\mu$ and $1\sigma$), and the leading variable are given.

*ATS2 led $CO_2$ by $449 \pm 257$ years at the T1 onset. Given the large range of uncertainty, we cannot exclude the possibility of synchrony.* *At the ACR onset, the range of uncertainty ($145 \pm 328$ years) that includes both the large relative chronological uncertainty and the ambiguity concerning the timing of the $CO_2$ change point, related to the large centennial scale variability near this point, does not allow us to identify a lead or lag .* *At the ACR end, $CO_2$ led ATS2, by $188 \pm 154$ years, but again, the possibility of synchrony cannot be excluded within $2\,\sigma$.*

*At the Holocene onset, a $CO_2$ lag is certain. Calculating the phasing between 12.0 ka and 11.0 ka, we obtain an ATS2 lead of $531 \pm 126$ years.*

*If the end of the centennial-scale change in $CO_2$ coincides with the true millenial scale change, which appears visually plausible, the ATS2 lead is much smaller. However, considerable uncertainty with respect to the millenial-scale change is expressed by the second mode, and the estimate of $574 \pm 143$ is statistically more appropriate.*

**3.4 Discussion**

Our results refine and complicate the timings and leads and lags identified by the most recent comparable studies (Parrenin et al., 2013; Pedro et al., 2012). We identify a $CO_2$ change point not treated in these studies at 16 ka, associated with the end of the centennial-scale rapid rise identified by Marcott et al. (2014), and an Antarctic temperature change point at 14 ka, neither of which have a counterpart in the other series.

During the major, multi-millenial scale changes which occur at T1 and Holocene onsets, Antarctic temperature likely led $CO_2$ by several centuries. However, during the complex, centennial-scale change at the ACR onset, ATS was most likely synchronous with $CO_2$, and at the end of the ACR, $CO_2$ leads temperature. Further, we do not identify an analog in $CO_2$ of the marked temperature decrease in Antarctica after the ACR onset, or a temperature analog for the $CO_2$ change at 16ka, indicating at least some degree of decoupling during these changes. Additionally, the $CO_2$ changes at the ACR and Holocene onsets are overlayed with centennial-scale substructures. Finally, synchrony is within the $2\sigma$ uncertainty range for each of the phasings, with the exception of the Holocene onset.

[revised manuscript text omitted]

---

## Referee Report (RR1)

The manuscript has had major revisions and is improved. However, more work is in my view still needed to justify some of the conclusions (see major points). I also have a rather long list of more technical points that need to be clarified or corrected.

**Major points**
- Regarding the phasing differences between temperature and CO2 at different Antarctic sites. The discussion of these results is very brief and the statement in the conclusions "we confirm that the deglacial temperature rise did not occur homogeneously across the Antarctic continent" is in my opinion not yet justified. It would help a great deal to include a table which shows the phasing results and uncertainties for the stack and for the individual cores for each change point. Then we could more easily see if the phasing differences are indeed significant and at what level.
- I am not completely convinced by the method to assess uncertainty in the phasing estimates. The additional comments in the supplement on significance testing largely repeat what is already in the main text. One way to test this and convince this reviewer and readers would be to trial the method on some artificial data with known change points and AR(1) noise.
- In comparing the lag results with other studies the authors use qualitative language like 'nearly.. roughly consistent.. etc'. Please use more quantitative language. For example the phasing at T1 in Parrenin et al was -10/pm 160 years and here it is 449 \pm 257 years. This needs some more explanation than 'only roughly consistent'. If these results are indeed to be a target for carbon cycle modeling, then we need to be confident that this number will not change again in several years, can we be? Please address this concern in the revised text.

**Technical points**
- Abstract. List the lead/lag and uncertainties for the four coherent changes instead of using ambiguous terms like 'nearly' and 'most likely led'. The note in the supplement indicates that the lags are near Gaussian so I can't see why not to report as such. Or, if you have a strong rationale for avoiding quantitative descriptions then make that argument clearly in the text.
- Line 9. For clarity drop "During the large, millenial-scale changes".
- Line 11. Again, give the lead/lag and uncertainty.
- Line 30. The ACR occurs midway (in time) through the deglaciation, not 'near the end'.
- P2 L35: missing citation.
- P2 L38: Marcott 2014 does not present evidence of increased Southern Ocean upwelling during the deglaciation as far as I'm aware. Better would be Anderson et al., Science 2009 and Skinner et al., Science, 2010.
- P3L35-36: No. The study used Byrd and Siple CO2 data not Law Dome CO2 data.

- P3L39: "Roughly in phase.. etc'. No, instead list the lag and uncertainty for the intervals mentioned in this sentence.
- P4L5: and others?
- P4L6: You mean during the satellite era?
- P4L24: "The standard deviation of the records at each timestep is assumed to be representative of the uncertainty concerning the conversion from isotopes to temperature,"... No I do not agree with this. Clarify or drop.
- P5L35: This sentence is unclear, please revise.
- P7L36: Python
- Fig 3 caption, L3: Probabilities.
- Fig 3: The dotted lines are a good addition. The caption should refer the reader to the relevant section of the text to understand where this threshold comes from.
- P8L18: The choice of the .0003 threshold over the..
- Fig 6. Caption: is the blue text including the result for the equivalent change point in Parrenin et al.,? Clarify in the caption.
- The result for deglacial onset appears substantially different to Parrenin et al. i.e. 10+- 160 yr $CO_2$ lead (Parrenin) to 449+_257 yr ATS2 lead (this work). Some specific comments on the main source of this timing difference are needed.
- P14L37: 'Though the T1 onset and the ACR end are both thought to originate in AMOC reductions (Marcott et al., 2014), our results allow for the $CO_2$ - ATS2 phasing to be reversed during the two events (i.e. with temperature leading at T1 and $CO_2$ leading at the ACR end).'
- P14L37: "CH4 changes nearly synchronously with $CO_2$ at both points, but the phasings are opposite in direction and different in magnitude." What is the basis for this statement? You did not assess CH4 phasing.
- Pl5 L4:" Within the range of uncertainty, our lead-lag estimates are only roughly consistent with those of Pedro et al. (2012) and Parrenin et al. (2013)." They are either consistent or they are not - use precise language.
- P14L35... Add some words to clarify this result: "However, the cumulative probability of the ATS2 change point is much greater before 17.7 ka than after (*see Figure 7*); hence our results are do not support McConnell et al's proposed volcanic forcing of the temperature change.
- P14L47: suggest to cite here Buizert et al., Nature, 2018 (https://doi.org/10.1038/s41586-018-0727-5) which finds that EDML has a consistently different atmospheric response to AMOC perturbations than other Antarctic records. Being geographically closer to the Atlantic does not necessarily imply EDML should resolve Atlantic temperature anomalies with more fidelity than other cores, the reason is the ACC barrier, which anomalies must mix across to enter the polar ocean; this likely happens to a large extent down-stream of the Atlantic sector (see Pedro et al., 2018, https://doi.org/10.1016/j.quascirev.2018.05.005).
- P16 L15: "This variability suggests complex mechanisms of coupling that can be modulated by external forcing". Expand on what you mean by this, why

should the modulation be external and not internal? As it stands this statement is not justified and not convincing.

- Figure 5: This was a very good addition, but plotting time series around the map, makes the panels much too small. I suggest a standard layout with the map above or below.
- P16 L17: .. between West and East Antarctica…
- P16 L18: what is meant by 'regional external influences'? Influences other than regional temperature?
- P16 L20: be more precise about what these differences are and why you think they are significant and at what level (a Table comparing the results for all sites would help).
- P16 L24: drop 'as is the investigation of the role $CO_2$ in global temperature change.' It is repeated further down.
- Supplement: The note on Gaussian uncertainties should be moved to the main text. The note on assessing significance is central to the results and should also be integrated into the main text (and see major comments above).

---

## Author Response (AR2)

Dear editor, dear reviewers,

We thank the two reviewers for their second round of comments, and the responses to these comments are given below along with a marked-up version of the text. Two changes are notable: first, in responding to Reviewer 1's first major comment, we have implemented the Bayesian Information Criterion to select a coherent number of change points. We also eliminate the use of a constant probability threshold, which Reviewer 1 found to be a design flaw, in favor of selecting the $n-2$ (or fewer) contiguous probability peaks with greatest integrated probability (and apart from some small regions, there are generally only $n-2$ or less visible peaks).

In addition, we have updated our results to include the d18O stack of Buizert et al. (2018), mentioned by one of the reviewers, which was made with more up-to-date volcanic synchronizations between the four East Antarctic cores and the WAIS Divide cores, using Sulfate instead of ECM data, is more reliable, and presents less chronological uncertainty.

These changes, importantly, do not significantly change the results of our paper.

Finally, we thank the editor as well for his detailed comments, which we have also taken into account.

All the best, on behalf of all co-authors,

Jai Chowdhry Beeman

Reviewer 1

*The authors have addressed several of my concerns regarding the methodology and improved their manuscript significantly. However, there remain a number of points in the methodology that in my opinion require further consideration:*

*1. I disagree with the authors' statement that they can investigate more change-points than the number defined in the model (which is 6+2). Each realization of the MCMC fits the data with n change-points. Hence, the PDF is also only true for the defined n. In their reply to my previous comment, the authors state that they generate one continuous PDF for the likelihood of a given x being a change-point, and that this PDF can have multiple (more than n) peaks. It is of course true, that there can be more than n peaks in the PDF, but some of them will be modes of the same change-point and thus, mutually exclusive. It would thus be more informative to generate one PDF per change-point. The way MCMC works, I am surprised that this posterior distribution for each change-point is not standard output of the method. This would affect the error estimates on the change-points and thus, the inferred delays. Furthermore, it would affect the significance test. Currently, the absolute probability threshold does not measure significance but precision. We can easily imagine how few iterations placing a change-point in the same location create a high and narrow peak in the PDF. On the other hand, many iterations placing a change-point into a slightly more variable location will result in a wider and thus, lower peak in the PDF. The cumulative probability is however higher in the latter example. Eventually, it is not a question of how many significant change-points there are: By defining n change-points, the answer is n. The question would then be whether a model with more/less change-points leads to a significantly improved fit to the data. Ideally, I would like the authors to address this and show posterior distribution per change-point. However, given the tests with more (SI) and less (SI of previous version of the manuscript) change-points I can also see how treating the change-points separately would probably not change the results drastically. Thus, I would appreciate if the authors could include both figures (more and less change-points, the latter also updated with the AR-model of this version of the manuscript) in the supplementary materials and add a short section in the main text, where they discuss the differences with respect to change-points and the resulting delays. I would appreciate if the authors could be more explicit/quantitative in their discussion of the differences than in this version of the manuscript (P8, L5-6: "probability distributions with 8 and 10 points are rather similar.") since the choice of number of change-points is ultimately arbitrary and represents an uncertainty in the method.*

— First, let us address the idea of treating the change points individually. A few examples can illustrate how this approach, while it would be practical for the method, is not correct.

Unlike previously developed change point fitting methods (i.e. Parrenin et al., 2013), we do not specify an initial position for the change points. Instead, we randomly initialize multiple walkers over a distribution uniform in time. The randomized initialization helps us reduce subjectivity in the method, and the use of multile walkers helps prevent the simulation from becoming stuck in local minima. As the MCMC simulation proceeds, the walkers converge towards the posterior distribution.

However, importantly, this does not mean that they converge toward one "best fit" solution. Rather, multiple approximate solution should be accepted proportionally to their probability.

Consider two fits accepted during the MCMC simulation with a gradual change. It is probable to accept a change point during this gradual change, and probable to not accept one at all. If in the first case the 5th point was fit in the gradual change and the 6th point at the next change point, in the second case the 5th point will be fit at the next change point, and the 6th afterward.

If, for example, the gradual change occurs at 10 ky BP and the next change at 5 ky BP, it is not reasonable to treat these as modes of the same change point, even though they are both fit by the 5th point. Similarly, it is not reasonable to treat the two points fitted at 5 ky BP as mutually exclusive, simply because one is fitted by the second change point and one the third. We cannot neglect the physical reality of there being only one change here for statistical convenience.

The reviewer's statement "eventually, it is not a question of how many significant change-points there are: By defining n change-points, the answer is n" has considerable value. Indeed, we can take the $n - 2$ (or fewer) contiguous regions of highest cumulative probability for a given number of points, which provides a consistent solution. Unlike the maximum probability threshold used in the previous iteration of this manuscript, this allows us to treat contiguous regions of high cumulative probability consistently, whether the maximum histogram bin value is high or low. In the case that there are less than $n$ contiguous regions of cumulative probability, we can assume that there are indeed less than $n$ significant change points. The marked tests of various numbers of points to fit the temperature stack are an example of this.

Selecting n is then the next challenge. We do this by applying the Bayesian Information Criterion (BIC) to our two series. The normalized, summed BIC for the linear models of both series suggests that n=8 is the most parsimonious solution (Supplementary Figure 2, shown below as well). However, because the BIC is merely a criterion, we still include fits with 5,6,7,8 and 9 points in the supplement. We also provide timings and lead-lag estimates for n=7 points in the supplementary spreadsheet.

The BIC is treated in detail in section 2.4 of the manuscript, and in supplementary Figure 7.

*2. The AR-model should also be applied to the filtered data. In the manuscript the authors state (P 10, L11-12): "In the two filtered series, the sub-millenial scale AR(1) noise present in the original series should be essentially removed. As such, fitting change points to these two series, assuming the residuals to be uncorrelated, provides a second form of verification of the appropriateness of the covariance matrix we use to fit the raw data." This is not true. Savitsky-Golay filters are smoothing filters that do not remove, but introduce autocorrelation. It is obvious that data filtered with a 500yr cutoff but supplied at 200yr resolution cannot be white noise. Furthermore, the method itself estimates autocorrelation (equation 2 and 3) and thus, if "a" in eq. 2 and 3 was indeed zero, then the method should detect it, set "a" to zero and hence, treat it as white noise automatically. Please, redo the analysis and figure 4 accordingly.*

Accepted. This figure is redone (Figure 4).

*3. The authors point out correctly, that the PDFs of the change-points are multimodal and skewed and should hence not be reported as $\mu \pm \sigma$. Looking at figure 6 I believe the same is true for the lead/lag PDFs. Please, throughout the manuscript, provide the most likely values (probability peak) and the 68.2/95.4% probability intervals.*

Accepted. Sigmas are changed to 68/95% confidence intervals in the abstract, section 3.2, section 3.3, the new table 1, and in the conclusions.

*4. I apologize for not commenting on this earlier: Could you rearrange the figures so that the time-series and the histograms do not overlap with each other, but are connected with X-gridlines? As it is now, some features discussed in the manuscript are impossible to see in the figures.*

Accepted. Figures 3 and 4 are revised so that the series and histograms do not overlap.

Specific Comments:

*P1, L7: Delete "abrupt". The method makes no estimate of the abruptness of changes.*

Accepted

*P1, L11: replace "nearly synchronously" with a more precise "within xx years [68.2% probability]" – the uncertainty is quite big for this change.*

Accepted (as done in general.)

*P1, L11: "250 years": Figure 6 reads -188±154? Please correct (also according to main comment 3)*

Accepted

*P1, L12: "after the ACR onset in the temperature record": I think one of the interesting results of this paper is, that this change point is actually only present in East Antarctica (EDC and TD), which could be worthwhile mentioning in the abstract?*

This point may be present in WD as well, though it occurs slightly later. It is worth noting that the probability of this change point is lower in the new stack. This observation is perhaps too complex to include clearly in the abstract.

*P2, L3: enter spaces before the two en dash uses.*

Accepted

*P4, L23 and P11, L. 24: Bintanja et al. 2013 is cited as a reference for the correction of water isotope records for source isotopic variations. Is this the correct reference? They do not study isotopes, and focus on the last 30 years? Please check.*

Bintanja et al. (2015) is the correct citation, though this is now removed because of the use of the Buizert et al. (2018) stack. The replacement of the stack does not significantly change the results.

*Page 5, figure caption 2: "ratio of the age difference". I still don't entirely understand this formulation. Do you mean the ratio of the age difference between neighbouring tie-points on WDC2014 and the same tie-points on EDC (on AICC12?). I.e., if the ratio is 1.01, then the duration between tie-points differs by 1% between the two independent timescales? Please clarify.*

The stack is now replaced, and uncertainty calculated as in Buizert et al. (2018).

*Page 5, figure caption 2: The 20% uncertainty is assumed to be 1 sigma? If so please change to: "… which is determined as 20% (1$\sigma$) of the distance to the nearest tie point, …"*

The stack is replaced, and uncertainty calculated as in Buizert et al. (2018).

*P6, L6: "… and can improve the balance of precision and accuracy of the fits." What do you mean? Delete?*

Deleted

*P6, L7: "related with" replace "related to"*

Accepted

*P6, L9: "first source of uncertainty" replace with "measurement uncertainty"*

Accepted

*P6, L15-17: "These challenges can be circumvented..." That would only be true if we knew that the resolution was lower than the decorrelation time. Is this the case? Looking at figure 4 in Parrenin et al. also there the residuals are autocorrelated.*
25 *Delete?*

Accepted

*P7, L32-33: replace "confirm" with "test" and "is accurate" with "cannot be rejected".*

Accepted

*P7, L33: "(Supplement)" what does this refer to? On that note: in the supplementary P1, L15, it reads that "a" is set to "2"? How do these two sections relate to each other? Is "a" estimated as described in eq. 2-3 or is it set to a constant value? Why would it be constant? I cannot find the motivation to set "a" to 2 in Goodman and Weare 2010.*

5 Not in Goodman and Weare, but in Foreman-Mackey et al. This is changed in the supplement.

*P7, L37: "$(IQR(R)/1.349)^2$": Please explain IQR (Inter-Quartile-Range?) and where the 1.349 is coming from.*

Inter-Quartile Range. 1.349 is an adjustment factor to calculate sample standard deviation from the Inter-Quartile Range
10 (Ghosh 2018, Silverman 1986).

*P8, L4-6: See main comment 1. Yes, there can be more modes, but eventually there can only be as many change-points as allowed by the user (6+2). Please rephrase, and provide a short section (here or elsewhere in the manuscript) discussing differences arising from the choice of allowed change-points.*

See the response above, to the longer comment. We discuss the difference arising from the choice of 7 or 8 change points at the ACR in the second paragraph of section 3.3, and provide fits with 5-9 points in the supplement. We also provide probability maxima and 65% / 95% confidence intervals for both 7 and 8 point fits in the supplementary spreadsheet.

20 *P8, L8: "second derivative of the fits": I don't understand. The fits are straight lines, so the second derivative is always 0?*

At the change points the first derivative is discontinuous and changes abruptly. We take into account the direction of this change. This is now worded "by the sign of the change in slope of the fits".

25 *P8, L8-16: I do not understand what is done here. 1000 change points at once? Or 1000x1 random change-point? Or 1000 x (6+2) random change-points? How do the histograms reflect the approximate slope? Please clarify this section.*

This section is no longer included, as we have removed the vertical probability threshold.

5 *P9, Figure caption 3: Line 3, typo "Rrobability"; Line 5: "the sum" - linear? Quadrature?*

Accepted. Changed to quadratic sum.

*P9, L36: "one at the ACR onset": see main comment 4 – this is impossible to see.*

Revised. The histograms in the figures now do not overlap with the time series.

*P10, L44-45: see main comment 4 – impossible to see.*

Revised. The histograms in the figures now do not overlap with the time series.

*P10, L48-49: "roughly concurrent with the downward CO2 change point" replace with "concurrent with the second mode of the downward CO2 change point" Also, this statement is at odds with P12, L16, where you state, that you do not identify a CO2 change-point. Please check for consistency.*

In the 7-point fit, there is only one mode of the CO2 change point.

*P10, L1 (50?, line-numbering is off): "No corresponding change point is detected for CO2". This is briefly discussed in section 3.2 but maybe it is already here worthwhile mentioning that this change point is only present in East Antarctic temperature records?*

We prefer to keep the analysis of the stack and individual records separate, to avoid confusion for the reader.

*P10, L4: "The Holocene onset is well defined". I suppose the way the Holocene is defined it cannot be seen in Antarctic isotope records. Maybe replace with "The end of the deglacial warming is well-defined.."*

Accepted "The end of the deglacial warming in Antarctica." This is replaced, in general, with the terminology "deglaciation end" or "T1 end".

*P10, L10-13: See main comment 2.*

Revised

*P11, L24: See earlier – is Bintanja et al. 2013 the correct reference?*

As above, we now fit individual d18O records, in accordance with Buizert et al. (2018). This does not significantly change the result.

*P11, L30: "rather broad, non-significant". That is by design. See comment 1. Maybe change to: "rather broad and hence, non-significant."*

The probability threshold no longer used; hence, we no longer assess significance.

*P12, L41: "...with temperature change at WD appearing to slightly precede temperature in the East Antarctic records,...". Omit the second use of "temperature" or change to "temperature change"*

Revised (now d18O)

*Figure 6: The PDFs in the top-left and bottom-right panels are the same plot. Please check/replace.*

These are replaced with new PDFs.

*Figure 6 and throughout this section: See main comment 3.*

30    Revised

*P13, L4-5: Mention again, that interestingly this is not the case for Dome Fuji?*

Accepted. This is included in section 3.2.

*P13, L12: Mention again that the 14k change-point is not present everywhere?*

5     Accepted. Included in section 3.2.

*P14, L15-16: "was most likely synchronous" change to: "occurred within xx years of the $CO_2$ change"*

Replaced with "we cannot calculate a clear lead of either ATS3 or $CO_2$" due to the large uncertainty regarding this change.
10
*P14, L16-17: "Further, we do not identify an analog in $CO_2$ of the marked temperature decrease…". But figure 3 and 4 do show downward histograms there? As mentioned in main comment 1, I think this is related to how significance is defined. There seems to be a downward change-point that is just not very well constrained in time, and hence, yields a low and wide probability distribution. See also earlier comment – check for consistency with P10, L48-49.*

5     This is an error, now reads "of the temperature stabilization"

*P14, L37: Within error also the ACR onset "allows" for the $CO_2$-ATS phasing to be reversed. And for both events, phasing is zero within error.*

10    This is the case at the 95% level. In accordance with a comment from Reviewer 2, revised to "our results allow for the $CO_2$-ATS3 phasing to be different during the two events, with the maximum probabilities reversed in directionality (i.e. with temperature leading at T1 and $CO_2$ leading at the ACR end, though zero phasing is within 95% error)"

*P14, L38: "But the phasings are opposite in direction and different in magnitude": This sentence is unclear. Do you mean*
15  *that $CO_2$ and $CH_4$ change in opposite directions? Why is it relevant that the magnitude of $CO_2$ and $CH_4$ change differs, and how do you define the magnitude for 2 gases that have very different background concentrations?*

Rephrased to: "Though $CH_4$ appears to change alongside $CO_2$ during both intervals, the phasings between $CO_2$ and Temperature during the intervals are opposite in direction and different in slope."
20
*P14, L42: "None of the five isotopic records show significant probability in this region" change to: "We do not detect a significant change-point for any of the isotope records during this period."*

Accepted

*Figure 7: Please also add a dashed vertical line to the oldest change-point/Mt. Takahe eruption that extends through all panels.*

*P15, L3: "only roughly consistent" Can you please be more precise?*

We have replaced this with an assessment of each change point.

10  *P15, L11, "ice-air shift" Please replace with "delta-age"*

Accepted.

*P16, L21. Also add, that the end of deglacial warming occurs later in DF?*

Accepted

Reviewer 2

Review of Chowdhry Beeman et al. *The manuscript has had major revisions and is improved. However, more work is in my view still needed to justify some of the conclusions (see major points). I also have a rather long list of more technical points that need to be clarified or corrected.*

*Major points*

*Regarding the phasing differences between temperature and CO2 at different Antarctic sites. The discussion of these results is very brief and the statement in the conclusions "we confirm that the deglacial temperature rise did not occur homogeneously across the Antarctic continent" is in my opinion not yet justified. It would help a great deal to include a table which shows the phasing results and uncertainties for the stack and for the individual cores for each change point. Then we could more easily see if the phasing differences are indeed significant and at what level.*

Accepted. We have now included such a table (timings and phasings for each core) in the supplementary spreadsheet.

*I am not completely convinced by the method to assess uncertainty in the phasing estimates. The additional comments in the supplement on significance testing largely repeat what is already in the main text. One way to test this and convince this reviewer and readers would be to trial the method on some artificial data with known change points and AR(1) noise.*

This is now present in the supplementary materials.

*In comparing the lag results with other studies the authors use qualitative language like 'nearly.. roughly consistent.. etc'. Please use more quantitative language. For example the phasing at T1 in Parrenin et al was -10± 160 years and here it is 449 ± 257 years. This needs some more explanation than 'only roughly consistent'. If these results are indeed to be a target for carbon cycle modeling, then we need to be confident that this number will not change again in several years, can we be? Please address this concern in the revised text.*

The phasings will be addressed in terms of probability (now included in the abstract, section 3.3, Table 1, and the supplementary spreadsheet).

However, we must accept that the numbers and uncertainty ranges for phasings may indeed change in several years (and so should modelers of the carbon cycle). As the reviewers know very well, paleoclimatology is a dynamic field, and new CO2 measurement methods, used on high-resolution cores, better source corrections for temperature isotopes, the inclusion of more highly-resolved temperature records from new drilling sites, and last but certainly not least better methods for addresing phasing! could all change the lead-lag estimates. Our ranges of uncertainty are by design relatively large (we attempt to account for as many sources of uncertainty as possible, including AR1 noise that may obstruct the detection change points), so new estimates will very likely fall within our ranges. These rather large ranges can provide an appropriate target for carbon cycle modelers. But advances and improvements are at the core of scientific investigation. So, at least the first author of this paper profoundly disagrees, in principle, that the numbers regarding the phasing between CO2 and Antarctic temperature records should not change again. Indeed, they should be revised as new records are developed, although hopefully the confidence intervals should stay compatible.

*Technical points*

*Abstract. List the lead/lag and uncertainties for the four coherent changes instead of using ambiguous terms like 'nearly' and 'most likely led'. The note in the supplement indicates that the lags are near Gaussian so I can't see why not to report as such. Or, if you have a strong rationale for avoiding quantitative descriptions then make that argument clearly in the text.*

The probability ranges are asymmetric, as noted by the first reviewer. Inserted.

Some description, though, helps a probability distribution become understandable, so we don't want to categorically give up on describing the distributions quantitatively.

*Line 9. For clarity drop "During the large, millenial-scale changes".*

Accepted (in the abstract).

*Line 11. Again, give the lead/lag and uncertainty.*

Accepted (in the abstract).

*Line 30. The ACR occurs midway (in time) through the deglaciation, not 'near the end'.*

Accepted (in the introduction). Now reads "midway through".

*P2 L35: missing citation.*

Added Anklin et al., 1995.

*P2 L38: Marcott 2014 does not present evidence of increased Southern Ocean upwelling during the deglaciation as far as I'm aware. Better would be Anderson et al., Science 2009 and Skinner et al., Science, 2010.*

Accepted

*P3L35-36: No. The study used Byrd and Siple CO2 data not Law Dome CO2 data.*

Modified accordingly.

*P3L39: "Roughly in phase.. etc'. No, instead list the lag and uncertainty for the intervals mentioned in this sentence.*

Accepted, but we still think it is important to give the reader an idea of the phasing that is easy to retain in the introduction. So descriptions will be left in the text alongside the phasing.

*P4L5: and others?*

Latex compilation error, deleted.

*P4L6: You mean during the satellite era?*

Yes, the directly observed (not paleoclimate proxy) record. Accepted.

*P4L24: "The standard deviation of the records at each timestep is assumed to be representative of the uncertainty concerning the conversion from isotopes to temperature,"... No I do not agree with this. Clarify or drop.*

Deleted

*P5L35: This sentence is unclear, please revise.*

Changed to:
We identify likely change points using piecewise linear functions. Residuals are calculated between linear functions with a fixed number of stochastically proposed change points, which are free to explore the entire temporal range of the time series, and the raw data (similarly to Parrenin et al. (2013)). These residuals form a cost function, which allows us to perform a Bayesian analysis of the probable timing of change points.

*P7L36: Python*

Accepted

*Fig 3 caption, L3: Probabilities.*

Accepted

*Fig 3: The dotted lines are a good addition. The caption should refer the reader to the relevant section of the text to understand where this threshold comes from.*

The threshold is no longer used, in accordance with the second reviewer's new round of comments.

*P8L18: The choice of the .0003 threshold over the..*

The threshold is no longer used, in accordance with the second reviewer's new round of comments.

*Fig 6. Caption: is the blue text including the result for the equivalent change point in Parrenin et al.,? Clarify in the caption.*

Accepted, this is now moved to Table 1.

*The result for deglacial onset appears substantially different to Parrenin et al. i.e. 10±160 yr CO2 lead (Parrenin) to 449±257 yr ATS2 lead (this work). Some specific comments on the main source of this timing difference are needed.*

The following will be included:

The considerable difference at T1 between our result and that of Parrenin et al. (2013) is most likely due to the much higher resolution of the WAIS Divide CO2 time series. It is also possible, that the result of Parrenin et al. (2013) was limited to a local probability maximum of this change point in the CO2 series.

*P14L37: 'Though the T1 onset and the ACR end are both thought to originate in AMOC reductions (Marcott et al., 2014), our results allow for the CO2 - ATS2 phasing to be reversed during the two events (i.e. with temperature leading at T1 and CO2 leading at the ACR end).'*

Accepted. In accordance with a comment from reviewer 1 This is now "our results allow for the CO2-ATS3 phasing to be different during the two events, with the maximum probabilities reversed in directionality (i.e. with temperature leading at T1

35    and CO2 leading at the ACR end, though zero phasing is within 95% error)"

*P14L37: "CH4 changes nearly synchronously with CO2 at both points, but the phasings are opposite in direction and different in magnitude." What is the basis for this statement? You did not assess CH4 phasing.*

Rephrased to: "Though CH4 appears to change alongside CO2 during both intervals, the phasings between CO2 and Temperature during the intervals are opposite in direction and different in slope."

*Pl5 L4:" Within the range of uncertainty, our lead-lag estimates are only roughly consistent with those of Pedro et al. (2012) and Parrenin et al. (2013)." They are either consistent or they are not - use precise language.*

Worded as below.

"Within the range of uncertainty, the mean of our lead-lag estimates is consistent with the boundaries proposed by Pedro et al. (2012). Our results are consistent with those of Parrenin et al. (2013) for three out of the four change points addressed, but differ considerably at the T1 onset."

15    *P14L35... Add some words to clarify this result: "However, the cumulative probability of the ATS2 change point is much greater before 17.7 ka than after (\*see Figure 7\*); hence our results are do not support McConnell et al's proposed volcanic forcing of the temperature change.*

Accepted.

*P14L47: suggest to cite here Buizert et al., Nature, 2018 (https://doi.org/10.1038/s41586-018-0727-5) which finds that EDML has a consistently different atmospheric response to AMOC perturbations than other Antarctic records. Being geographically closer to the Atlantic does not necessarily imply EDML should resolve Atlantic temperature anomalies with more fidelity than other cores, the reason is the ACC barrier, which anomalies must mix across to enter the polar ocean; this likely happens*
25    *to a large extent down-stream of the Atlantic sector (see Pedro et al., 2018, https://doi.org/10.1016/j.quascirev.2018.05.005).*

Accepted, we also now use the d18O stack from Buizert et al. (2018). This now reads "EDML indeed appears to record changes in AMOC differently than the other isotopic records (Landais et al., 2018; Buizert et al., 2018).

30    *P16 L15: "This variability suggests complex mechanisms of coupling that can be modulated by external forcing". Expand on what you mean by this, why should the modulation be external and not internal? As it stands this statement is not justified and not convincing.*

Reworded:

35

"This variability of phasings indicates that the mechanisms of coupling are complex. We propose three possibilities: I) the mechanisms by which CO2 and Antarctic Temperature were coupled were consistent throught the deglaciation, but can be modulated by external forcings or background conditions that impact heat transfer and oceanic circulation (and hence CO2 release); II) these mechanisms can be modulated by internal feedbacks that change the response timings of the two series; and/or III) multiple, distinct mechanisms might have provoked similar responses in both series, but with accordingly different lags."

*Figure 5: This was a very good addition, but plotting time series around the map, makes the panels much too small. I suggest*
485    *a standard layout with the map above or below.*

This figure is revised.

490

*P16 L17: .. between West and East Antarctica. . .*

Can the comment be clarified?

*P16 L18: what is meant by 'regional external influences'? Influences other than regional temperature?*

495

"is complicated by influences other than regional temperature, including source temperature and ice sheet elevation change"

*P16 L20: be more precise about what these differences are and why you think they are significant and at what level (a Table comparing the results for all sites would help).*

500

Accepted, a spreadsheet is now included in the supplement.

*P16 L24: drop 'as is the investigation of the role CO2 in global temperature change.' It is repeated further down.*

Accepted

505

*Supplement: The note on Gaussian uncertainties should be moved to the main text. The note on assessing significance is central to the results and should also be integrated into the main text (and see major comments above).*

These points have now been changed, in accordance with reviewer one's comments.

[revised manuscript text omitted]

---

## Author Response (AR3)

Dear editor,

We appreciate the latest round of revisions, and these have been taken into account in this version. Point-by-point responses and a marked-up manuscript are included in this file.

On behalf of all co-authors, all the best,

Jai Chowdhry Beeman

—

Point-by-point responses

"I don't think parsimonious is the best expression also in the main text!"
(Previous responses, in Section 2.4)
*Parsimonious is indeed the expression we wish to use, see Box and Jenkins (1976) who define parsimony as the "smallest possible number of parameters for adequate representation" and Findley (1991) who gives: "When the log-likelihood-ratio sequence of two models with different numbers of parameters is bounded in probability, then model selection criteria like BIC, whose penalty for estimated parameters becomes infinite with N, will obey the principle of parsimony in the strong sense that the probability of selecting the model with fewer parameters approaches one as N increases."*

"[...] In the revised supplement, it says you selected a to be 2 but in fact you fit the autocorrelation coefficient. Please clarify and correct!" (Regarding the variable a which was used mistakenly in two unrelated equations, in the Supplement)

*The variable a is changed to $A_Z$ in the supplement to differentiate between the two. This variable is used only in the MCMC routine and is unrelated with the autocorrelation.*

"A general comment. The late change point in CO2 at the end of T1 needs more commenting in the main text. Looking at the figures nobody would put the change point there by eye, so what is the reason? Is it a resolution or noise problem? You should also qualify your conclusions on the CO2 lag at the end of the T1 accordingly!"

*We had treated this issue in a previous round of the manuscript, but one of the reviewers had preferred that we only address the region of highest probability, so the commentary was removed. We appreciate the editor's observation and have made an effort to re-introduce this discussion in a manner that still emphasizes that the region of much greater probability occurs later, in order to make a compromise of sorts between the two viewpoints. We add the following to the section on change point timing (3.1):*
*Visually, we might question why the CO2 change point is deemed to most probably occur at 11.15 ka, when a kink in the series appears to occur closer to 11.5 ka. It is important to note that two minor spikes in probability appear to fit the rapid rise that occurs close to 11.5 ka, and that the downward spike at the end of this rise indeed indicates that a small number of fits does indeed fit a change point here.*
*Since the resolution of the WAIS Divide dataset decreases considerably after the Holocene onset, and only three points account for the majority of the rapid rise that occurs before the Holocene onset, most of the weight is given to the obvious line beginning at the ACR end. Adding an additional point should allow us to fit this slightly better–the CO2 series is slightly better fit with 9 points, according to the individual BIC values, and there is more probability mass around the rapid rise in the 9-point fit, though the peak at 11.15 ka is still dominant. In any case, some methodological uncertainty exists regarding the location of this point, and the probability estimate is possibly biased by the quick change in resolution. Better resolution around this point will help identify the true location of the change.*

*In section 3.3, about phasing:*

*This estimate is complicated, though, if we consider the small possibility that the true $CO_2$ change point occurs closer to 11.5 ka BP, at the end of the rapid rise. In this case, the phasing is reduced to 174 years (68% central probability range of 65 to 280 years) and synchrony is within the 95% central probability interval (-71 to 411 years).*

"here and throughout the manuscript at BP or B1950 if you refer to ages!!!" (Abstract)

*Revised to BP throughout.*

"name???" (LU, section 2.3)

*Revised to Lower/Upper.*

"wrong expression" (parsimony)

*We still believe this is the correct expression, see above.*

"unclear, please change text" (End of section 2.4)

*Changed to:*
*"We avoid comparing incoherent modes by separating changes by the sign of the change in slope of the fits. If the slope decreases at a change point, the change in slope is negative, or concave-down. These changes are indicated by the downward-facing part of the histogram graphs. Note that while this part of the histogram appears 'negative' probabilities cannot be negative, and this simply indicates that the probability is for a concave-down change point. If the slope increases at a change point, the change in slope is positive, or concave-up. The probabilities of these changes are indicated by the upward-facing, or 'positive' part of the histogram. When we calculate leads and lags, we only do so for either a region in which there is a probability peak for a concave-down change point in both series, or for a concave-up change point in both series, but we do not treat concave-up probability and concave-down probability together."*

"combine Figure 3 and 7, they are largely redundant"

*Accepted!*

"why "fits", Please clarify wording" (Beginning of section 3.2)

*Changed to "Histograms calculated for..."*

"name the events as in Fig 3" (for figure 5)

*Accepted.*

"you don't provide mu and sigma, you provide the 68 and 95 percentiles. Please correct!" (Figure 6)

*Corrected!*

Supplement:

"See comment in the main text, There is something wrong here, either it is set to 2 or the autocorrelation is fitted."

*This is revised, see above.*

5     "isn't it 5,6,7,8 and 9 change points?"

*Corrected.*

"In the supplementary xls table provided with the submission the phasing of ATS3 with 7 change points is missing. Please
10    add!"

*It is to the right of the phasing with 8 points in sheet 1.*

References for responses:

Findley, D. F. (1991). Counterexamples to parsimony and BIC. Annals of the Institute of Statistical Mathematics, 43(3), 505-514.

Box, G. E. P. and Jenkins, G. M. (1976). Time Series Analysis: Forecasting and Control, 2nd ed., Holden-Day, San Francisco.

[revised manuscript text omitted]

---

## Editor Decision (ED3)

Dear editor, dear reviewers,

We thank the two reviewers for their second round of comments, and the responses to these comments are given below along with a marked-up version of the text. Two changes are notable: first, in responding to Reviewer 1's first major comment, we have implemented the Bayesian Information Criterion to select a coherent number of change points. We also eliminate the use of a constant probability threshold, which Reviewer 1 found to be a design flaw, in favor of selecting the $n-2$ (or fewer) contiguous probability peaks with greatest integrated probability (and apart from some small regions, there are generally only $n-2$ or less visible peaks).

In addition, we have updated our results to include the d18O stack of Buizert et al. (2018), mentioned by one of the reviewers, which was made with more up-to-date volcanic synchronizations between the four East Antarctic cores and the WAIS Divide cores, using Sulfate instead of ECM data, is more reliable, and presents less chronological uncertainty.

These changes, importantly, do not significantly change the results of our paper.

Finally, we thank the editor as well for his detailed comments, which we have also taken into account.

All the best, on behalf of all co-authors,

Jai Chowdhry Beeman

Reviewer 1

*The authors have addressed several of my concerns regarding the methodology and improved their manuscript significantly. However, there remain a number of points in the methodology that in my opinion require further consideration:*

*1. I disagree with the authors' statement that they can investigate more change-points than the number defined in the model (which is 6+2). Each realization of the MCMC fits the data with n change-points. Hence, the PDF is also only true for the defined n. In their reply to my previous comment, the authors state that they generate one continuous PDF for the likelihood of a given x being a change-point, and that this PDF can have multiple (more than n) peaks. It is of course true, that there can be more than n peaks in the PDF, but some of them will be modes of the same change-point and thus, mutually exclusive. It would thus be more informative to generate one PDF per change-point. The way MCMC works, I am surprised that this posterior distribution for each change-point is not standard output of the method. This would affect the error estimates on the change-points and thus, the inferred delays. Furthermore, it would affect the significance test. Currently, the absolute probability threshold does not measure significance but precision. We can easily imagine how few iterations placing a change-point in the same location create a high and narrow peak in the PDF. On the other hand, many iterations placing a change-point into a slightly more variable location will result in a wider and thus, lower peak in the PDF. The cumulative probability is however higher in the latter example. Eventually, it is not a question of how many significant change-points there are: By defining n change-points, the answer is n. The question would then be whether a model with more/less change-points leads to a significantly improved fit to the data. Ideally, I would like the authors to address this and show posterior distribution per change-point. However, given the tests with more (SI) and less (SI of previous version of the manuscript) change-points I can also see how treating the change-points separately would probably not change the results drastically. Thus, I would appreciate if the authors could include both figures (more and less change-points, the latter also updated with the AR-model of this version of the manuscript) in the supplementary materials and add a short section in the main text, where they discuss the differences with respect to change-points and the resulting delays. I would appreciate if the authors could be more explicit/quantitative in their discussion of the differences than in this version of the manuscript (P8, L5-6: "probability distributions with 8 and 10 points are rather similar.") since the choice of number of change-points is ultimately arbitrary and represents an uncertainty in the method.*

— First, let us address the idea of treating the change points individually. A few examples can illustrate how this approach, while it would be practical for the method, is not correct.

Unlike previously developed change point fitting methods (i.e. Parrenin et al., 2013), we do not specify an initial position for the change points. Instead, we randomly initialize multiple walkers over a distribution uniform in time. The randomized initialization helps us reduce subjectivity in the method, and the use of multile walkers helps prevent the simulation from becoming stuck in local minima. As the MCMC simulation proceeds, the walkers converge towards the posterior distribution.

However, importantly, this does not mean that they converge toward one "best fit" solution. Rather, multiple approximate solution should be accepted proportionally to their probability.

Consider two fits accepted during the MCMC simulation with a gradual change. It is probable to accept a change point during this gradual change, and probable to not accept one at all. If in the first case the 5th point was fit in the gradual change and the 6th point at the next change point, in the second case the 5th point will be fit at the next change point, and the 6th afterward.

If, for example, the gradual change occurs at 10 ky BP and the next change at 5 ky BP, it is not reasonable to treat these as modes of the same change point, even though they are both fit by the 5th point. Similarly, it is not reasonable to treat the two points fitted at 5 ky BP as mutually exclusive, simply because one is fitted by the second change point and one the third. We cannot neglect the physical reality of there being only one change here for statistical convenience.

The reviewer's statement "eventually, it is not a question of how many significant change-points there are: By defining n change-points, the answer is n" has considerable value. Indeed, we can take the $n - 2$ (or fewer) contiguous regions of highest cumulative probability for a given number of points, which provides a consistent solution. Unlike the maximum probability threshold used in the previous iteration of this manuscript, this allows us to treat contiguous regions of high cumulative probability consistently, whether the maximum histogram bin value is high or low. In the case that there are less than $n$ contiguous regions of cumulative probability, we can assume that there are indeed less than $n$ significant change points. The marked tests of various numbers of points to fit the temperature stack are an example of this.

Selecting n is then the next challenge. We do this by applying the Bayesian Information Criterion (BIC) to our two series. The normalized, summed BIC for the linear models of both series suggests that n=8 is the most parsimonious solution (Supplementary Figure 2, shown below as well). However, because the BIC is merely a criterion, we still include fits with 5,6,7,8 and 9 points in the supplement. We also provide timings and lead-lag estimates for n=7 points in the supplementary spreadsheet.

The BIC is treated in detail in section 2.4 of the manuscript, and in supplementary Figure 7.

*2. The AR-model should also be applied to the filtered data. In the manuscript the authors state (P 10, L11-12): "In the two filtered series, the sub-millenial scale AR(1) noise present in the original series should be essentially removed. As such, fitting change points to these two series, assuming the residuals to be uncorrelated, provides a second form of verification of the appropriateness of the covariance matrix we use to fit the raw data." This is not true. Savitsky-Golay filters are smoothing filters that do not remove, but introduce autocorrelation. It is obvious that data filtered with a 500yr cutoff but supplied at 200yr resolution cannot be white noise. Furthermore, the method itself estimates autocorrelation (equation 2 and 3) and thus, if "a" in eq. 2 and 3 was indeed zero, then the method should detect it, set "a" to zero and hence, treat it as white noise automatically. Please, redo the analysis and figure 4 accordingly.*

Accepted. This figure is redone (Figure 4).

*3. The authors point out correctly, that the PDFs of the change-points are multimodal and skewed and should hence not be reported as $\mu \pm \sigma$. Looking at figure 6 I believe the same is true for the lead/lag PDFs. Please, throughout the manuscript, provide the most likely values (probability peak) and the 68.2/95.4% probability intervals.*

Accepted. Sigmas are changed to 68/95% confidence intervals in the abstract, section 3.2, section 3.3, the new table 1, and in the conclusions.

*4. I apologize for not commenting on this earlier: Could you rearrange the figures so that the time-series and the histograms do not overlap with each other, but are connected with X-gridlines? As it is now, some features discussed in the manuscript are impossible to see in the figures.*

Accepted. Figures 3 and 4 are revised so that the series and histograms do not overlap.

Specific Comments:

*P1, L7: Delete "abrupt". The method makes no estimate of the abruptness of changes.*

Accepted

*P1, L11: replace "nearly synchronously" with a more precise "within xx years [68.2% probability]" – the uncertainty is quite big for this change.*

Accepted (as done in general.)

*P1, L11: "250 years": Figure 6 reads -188±154? Please correct (also according to main comment 3)*

Accepted

*P1, L12: "after the ACR onset in the temperature record": I think one of the interesting results of this paper is, that this change point is actually only present in East Antarctica (EDC and TD), which could be worthwhile mentioning in the abstract?*

This point may be present in WD as well, though it occurs slightly later. It is worth noting that the probability of this change point is lower in the new stack. This observation is perhaps too complex to include clearly in the abstract.

*P2, L3: enter spaces before the two en dash uses.*

Accepted

*P4, L23 and P11, L. 24: Bintanja et al. 2013 is cited as a reference for the correction of water isotope records for source isotopic variations. Is this the correct reference? They do not study isotopes, and focus on the last 30 years? Please check.*

Bintanja et al. (2015) is the correct citation, though this is now removed because of the use of the Buizert et al. (2018) stack. The replacement of the stack does not significantly change the results.

*Page 5, figure caption 2: "ratio of the age difference". I still don't entirely understand this formulation. Do you mean the ratio of the age difference between neighbouring tie-points on WDC2014 and the same tie-points on EDC (on AICC12?). I.e., if the ratio is 1.01, then the duration between tie-points differs by 1% between the two independent timescales? Please clarify.*

The stack is now replaced, and uncertainty calculated as in Buizert et al. (2018).

*Page 5, figure caption 2: The 20% uncertainty is assumed to be 1 sigma? If so please change to: "... which is determined as 20% (1$\sigma$) of the distance to the nearest tie point, ..."*

The stack is replaced, and uncertainty calculated as in Buizert et al. (2018).

*P6, L6: "... and can improve the balance of precision and accuracy of the fits." What do you mean? Delete?*

Deleted

*P6, L7: "related with" replace "related to"*

Accepted

*P6, L9: "first source of uncertainty" replace with "measurement uncertainty"*

Accepted

*P6, L15-17: "These challenges can be circumvented..." That would only be true if we knew that the resolution was lower than the decorrelation time. Is this the case? Looking at figure 4 in Parrenin et al. also there the residuals are autocorrelated.*

25 *Delete?*

Accepted

*P7, L32-33: replace "confirm" with "test" and "is accurate" with "cannot be rejected".*

Accepted

*P7, L33: "(Supplement)" what does this refer to? On that note: in the supplementary P1, L15, it reads that "a" is set to "2"? How do these two sections relate to each other? Is "a" estimated as described in eq. 2-3 or is it set to a constant value? Why would it be constant? I cannot find the motivation to set "a" to 2 in Goodman and Weare 2010.*

5 Not in Goodman and Weare, but in Foreman-Mackey et al. This is changed in the supplement.

*P7, L37: "$(IQR(R)/1.349)^2$": Please explain IQR (Inter-Quartile-Range?) and where the 1.349 is coming from.*

Inter-Quartile Range. 1.349 is an adjustment factor to calculate sample standard deviation from the Inter-Quartile Range
10 (Ghosh 2018, Silverman 1986).

*P8, L4-6: See main comment 1. Yes, there can be more modes, but eventually there can only be as many change-points as allowed by the user (6+2). Please rephrase, and provide a short section (here or elsewhere in the manuscript) discussing differences arising from the choice of allowed change-points.*

See the response above, to the longer comment. We discuss the difference arising from the choice of 7 or 8 change points at the ACR in the second paragraph of section 3.3, and provide fits with 5-9 points in the supplement. We also provide probability maxima and 65% / 95% confidence intervals for both 7 and 8 point fits in the supplementary spreadsheet.

20 *P8, L8: "second derivative of the fits": I don't understand. The fits are straight lines, so the second derivative is always 0?*

At the change points the first derivative is discontinuous and changes abruptly. We take into account the direction of this change. This is now worded "by the sign of the change in slope of the fits".

25 *P8, L8-16: I do not understand what is done here. 1000 change points at once? Or 1000x1 random change-point? Or 1000 x (6+2) random change-points? How do the histograms reflect the approximate slope? Please clarify this section.*

This section is no longer included, as we have removed the vertical probability threshold.

5 *P9, Figure caption 3: Line 3, typo "Rrobability"; Line 5: "the sum" - linear? Quadrature?*

Accepted. Changed to quadratic sum.

*P9, L36: "one at the ACR onset": see main comment 4 – this is impossible to see.*

Revised. The histograms in the figures now do not overlap with the time series.

*P10, L44-45: see main comment 4 – impossible to see.*

Revised. The histograms in the figures now do not overlap with the time series.

*P10, L48-49: "roughly concurrent with the downward CO2 change point" replace with "concurrent with the second mode of the downward CO2 change point" Also, this statement is at odds with P12, L16, where you state, that you do not identify a CO2 change-point. Please check for consistency.*

In the 7-point fit, there is only one mode of the CO2 change point.

*P10, L1 (50?, line-numbering is off): "No corresponding change point is detected for CO2". This is briefly discussed in section 3.2 but maybe it is already here worthwhile mentioning that this change point is only present in East Antarctic temperature records?*

We prefer to keep the analysis of the stack and individual records separate, to avoid confusion for the reader.

*P10, L4: "The Holocene onset is well defined". I suppose the way the Holocene is defined it cannot be seen in Antarctic isotope records. Maybe replace with "The end of the deglacial warming is well-defined.."*

Accepted "The end of the deglacial warming in Antarctica." This is replaced, in general, with the terminology "deglaciation end" or "T1 end".

*P10, L10-13: See main comment 2.*

Revised

*P11, L24: See earlier – is Bintanja et al. 2013 the correct reference?*

As above, we now fit individual d18O records, in accordance with Buizert et al. (2018). This does not significantly change the result.

*P11, L30: "rather broad, non-significant". That is by design. See comment 1. Maybe change to: "rather broad and hence, non-significant."*

The probability threshold no longer used; hence, we no longer assess significance.

*P12, L41: "…with temperature change at WD appearing to slightly precede temperature in the East Antarctic records,…". Omit the second use of "temperature" or change to "temperature change"*

Revised (now d18O)

*Figure 6: The PDFs in the top-left and bottom-right panels are the same plot. Please check/replace.*

These are replaced with new PDFs.

*Figure 6 and throughout this section: See main comment 3.*

30    Revised

*P13, L4-5: Mention again, that interestingly this is not the case for Dome Fuji?*

Accepted. This is included in section 3.2.

*P13, L12: Mention again that the 14k change-point is not present everywhere?*

5     Accepted. Included in section 3.2.

*P14, L15-16: "was most likely synchronous" change to: "occurred within xx years of the CO2 change"*

Replaced with "we cannot calculate a clear lead of either ATS3 or CO2" due to the large uncertainty regarding this change.
10
*P14, L16-17: "Further, we do not identify an analog in CO2 of the marked temperature decrease…". But figure 3 and 4 do show downward histograms there? As mentioned in main comment 1, I think this is related to how significance is defined. There seems to be a downward change-point that is just not very well constrained in time, and hence, yields a low and wide probability distribution. See also earlier comment – check for consistency with P10, L48-49.*

5     This is an error, now reads "of the temperature stabilization"

*P14, L37: Within error also the ACR onset "allows" for the CO2-ATS phasing to be reversed. And for both events, phasing is zero within error.*

10    This is the case at the 95% level. In accordance with a comment from Reviewer 2, revised to "our results allow for the CO2-ATS3 phasing to be different during the two events, with the maximum probabilities reversed in directionality (i.e. with temperature leading at T1 and CO2 leading at the ACR end, though zero phasing is within 95% error)"

*P14, L38: "But the phasings are opposite in direction and different in magnitude": This sentence is unclear. Do you mean*
15  *that CO2 and CH4 change in opposite directions? Why is it relevant that the magnitude of CO2 and CH4 change differs, and how do you define the magnitude for 2 gases that have very different background concentrations?*

Rephrased to: "Though CH4 appears to change alongside CO2 during both intervals, the phasings between CO2 and Temperature during the intervals are opposite in direction and different in slope."
20
*P14, L42: "None of the five isotopic records show significant probability in this region" change to: "We do not detect a significant change-point for any of the isotope records during this period."*

Accepted

*Figure 7: Please also add a dashed vertical line to the oldest change-point/Mt. Takahe eruption that extends through all panels.*
5
*P15, L3: "only roughly consistent" Can you please be more precise?*

We have replaced this with an assessment of each change point.

10  *P15, L11, "ice-air shift" Please replace with "delta-age"*

Accepted.

*P16, L21. Also add, that the end of deglacial warming occurs later in DF?*

Accepted

Reviewer 2

Review of Chowdhry Beeman et al. *The manuscript has had major revisions and is improved. However, more work is in my view still needed to justify some of the conclusions (see major points). I also have a rather long list of more technical points that need to be clarified or corrected.*

*Major points*

*Regarding the phasing differences between temperature and CO2 at different Antarctic sites. The discussion of these results is very brief and the statement in the conclusions "we confirm that the deglacial temperature rise did not occur homogeneously across the Antarctic continent" is in my opinion not yet justified. It would help a great deal to include a table which shows the phasing results and uncertainties for the stack and for the individual cores for each change point. Then we could more easily see if the phasing differences are indeed significant and at what level.*

Accepted. We have now included such a table (timings and phasings for each core) in the supplementary spreadsheet.

*I am not completely convinced by the method to assess uncertainty in the phasing estimates. The additional comments in the supplement on significance testing largely repeat what is already in the main text. One way to test this and convince this reviewer and readers would be to trial the method on some artificial data with known change points and AR(1) noise.*

This is now present in the supplementary materials.

*In comparing the lag results with other studies the authors use qualitative language like 'nearly.. roughly consistent.. etc'. Please use more quantitative language. For example the phasing at T1 in Parrenin et al was -10± 160 years and here it is 449 ± 257 years. This needs some more explanation than 'only roughly consistent'. If these results are indeed to be a target for carbon cycle modeling, then we need to be confident that this number will not change again in several years, can we be? Please address this concern in the revised text.*

The phasings will be addressed in terms of probability (now included in the abstract, section 3.3, Table 1, and the supplementary spreadsheet).

However, we must accept that the numbers and uncertainty ranges for phasings may indeed change in several years (and so should modelers of the carbon cycle). As the reviewers know very well, paleoclimatology is a dynamic field, and new CO2 measurement methods, used on high-resolution cores, better source corrections for temperature isotopes, the inclusion of more highly-resolved temperature records from new drilling sites, and last but certainly not least better methods for addresing phasing! could all change the lead-lag estimates. Our ranges of uncertainty are by design relatively large (we attempt to account for as many sources of uncertainty as possible, including AR1 noise that may obstruct the detection change points), so new estimates will very likely fall within our ranges. These rather large ranges can provide an appropriate target for carbon cycle modelers. But advances and improvements are at the core of scientific investigation. So, at least the first author of this paper profoundly disagrees, in principle, that the numbers regarding the phasing between CO2 and Antarctic temperature records should not change again. Indeed, they should be revised as new records are developed, although hopefully the confidence intervals should stay compatible.

*Technical points*

*Abstract. List the lead/lag and uncertainties for the four coherent changes instead of using ambiguous terms like 'nearly' and 'most likely led'. The note in the supplement indicates that the lags are near Gaussian so I can't see why not to report as such. Or, if you have a strong rationale for avoiding quantitative descriptions then make that argument clearly in the text.*

The probability ranges are asymmetric, as noted by the first reviewer. Inserted.

Some description, though, helps a probability distribution become understandable, so we don't want to categorically give up on describing the distributions quantitatively.

*Line 9. For clarity drop "During the large, millenial-scale changes".*

Accepted (in the abstract).

*Line 11. Again, give the lead/lag and uncertainty.*

Accepted (in the abstract).

*Line 30. The ACR occurs midway (in time) through the deglaciation, not 'near the end'.*

Accepted (in the introduction). Now reads "midway through".

*P2 L35: missing citation.*

Added Anklin et al., 1995.

*P2 L38: Marcott 2014 does not present evidence of increased Southern Ocean upwelling during the deglaciation as far as I'm aware. Better would be Anderson et al., Science 2009 and Skinner et al., Science, 2010.*

Accepted

*P3L35-36: No. The study used Byrd and Siple CO2 data not Law Dome CO2 data.*

Modified accordingly.

*P3L39: "Roughly in phase.. etc'. No, instead list the lag and uncertainty for the intervals mentioned in this sentence.*

Accepted, but we still think it is important to give the reader an idea of the phasing that is easy to retain in the introduction. So descriptions will be left in the text alongside the phasing.

*P4L5: and others?*

Latex compilation error, deleted.

*P4L6: You mean during the satellite era?*

Yes, the directly observed (not paleoclimate proxy) record. Accepted.

*P4L24: "The standard deviation of the records at each timestep is assumed to be representative of the uncertainty concerning the conversion from isotopes to temperature,". . . No I do not agree with this. Clarify or drop.*

Deleted

*P5L35: This sentence is unclear, please revise.*

Changed to:
We identify likely change points using piecewise linear functions. Residuals are calculated between linear functions with a fixed number of stochastically proposed change points, which are free to explore the entire temporal range of the time series, and the raw data (similarly to Parrenin et al. (2013)). These residuals form a cost function, which allows us to perform a Bayesian analysis of the probable timing of change points.

*P7L36: Python*

Accepted

*Fig 3 caption, L3: Probabilities.*

Accepted

*Fig 3: The dotted lines are a good addition. The caption should refer the reader to the relevant section of the text to understand where this threshold comes from.*

The threshold is no longer used, in accordance with the second reviewer's new round of comments.

*P8L18: The choice of the .0003 threshold over the..*

The threshold is no longer used, in accordance with the second reviewer's new round of comments.

*Fig 6. Caption: is the blue text including the result for the equivalent change point in Parrenin et al.,? Clarify in the caption.*

Accepted, this is now moved to Table 1.

*The result for deglacial onset appears substantially different to Parrenin et al. i.e. $10\pm160$ yr CO2 lead (Parrenin) to $449\pm257$ yr ATS2 lead (this work). Some specific comments on the main source of this timing difference are needed.*

The following will be included:

The considerable difference at T1 between our result and that of Parrenin et al. (2013) is most likely due to the much higher resolution of the WAIS Divide CO2 time series. It is also possible, that the result of Parrenin et al. (2013) was limited to a local probability maximum of this change point in the CO2 series.

*P14L37: 'Though the T1 onset and the ACR end are both thought to originate in AMOC reductions (Marcott et al., 2014), our results allow for the CO2 - ATS2 phasing to be reversed during the two events (i.e. with temperature leading at T1 and CO2 leading at the ACR end).'*

Accepted. In accordance with a comment from reviewer 1 This is now "our results allow for the CO2-ATS3 phasing to be different during the two events, with the maximum probabilities reversed in directionality (i.e. with temperature leading at T1

35  and CO2 leading at the ACR end, though zero phasing is within 95% error)"

*P14L37: "CH4 changes nearly synchronously with CO2 at both points, but the phasings are opposite in direction and different in magnitude." What is the basis for this statement? You did not assess CH4 phasing.*

Rephrased to: "Though CH4 appears to change alongside CO2 during both intervals, the phasings between CO2 and Temperature during the intervals are opposite in direction and different in slope."

*Pl5 L4:" Within the range of uncertainty, our lead-lag estimates are only roughly consistent with those of Pedro et al. (2012) and Parrenin et al. (2013)." They are either consistent or they are not - use precise language.*

Worded as below.

"Within the range of uncertainty, the mean of our lead-lag estimates is consistent with the boundaries proposed by Pedro et al. (2012). Our results are consistent with those of Parrenin et al. (2013) for three out of the four change points addressed, but differ considerably at the T1 onset."

15  *P14L35... Add some words to clarify this result: "However, the cumulative probability of the ATS2 change point is much greater before 17.7 ka than after (\*see Figure 7\*); hence our results are do not support McConnell et al's proposed volcanic forcing of the temperature change.*

Accepted.

*P14L47: suggest to cite here Buizert et al., Nature, 2018 (https://doi.org/10.1038/s41586-018-0727-5) which finds that EDML has a consistently different atmospheric response to AMOC perturbations than other Antarctic records. Being geographically closer to the Atlantic does not necessarily imply EDML should resolve Atlantic temperature anomalies with more fidelity than other cores, the reason is the ACC barrier, which anomalies must mix across to enter the polar ocean; this likely happens*
25  *to a large extent down-stream of the Atlantic sector (see Pedro et al., 2018, https://doi.org/10.1016/j.quascirev.2018.05.005).*

Accepted, we also now use the d18O stack from Buizert et al. (2018). This now reads "EDML indeed appears to record changes in AMOC differently than the other isotopic records (Landais et al., 2018; Buizert et al., 2018).

30  *P16 L15: "This variability suggests complex mechanisms of coupling that can be modulated by external forcing". Expand on what you mean by this, why should the modulation be external and not internal? As it stands this statement is not justified and not convincing.*

Reworded:

"This variability of phasings indicates that the mechanisms of coupling are complex. We propose three possibilities: I) the mechanisms by which CO2 and Antarctic Temperature were coupled were consistent throught the deglaciation, but can be modulated by external forcings or background conditions that impact heat transfer and oceanic circulation (and hence CO2 release); II) these mechanisms can be modulated by internal feedbacks that change the response timings of the two series; and/or III) multiple, distinct mechanisms might have provoked similar responses in both series, but with accordingly different lags."

*Figure 5: This was a very good addition, but plotting time series around the map, makes the panels much too small. I suggest*
485  *a standard layout with the map above or below.*

This figure is revised.

490

*P16 L17: .. between West and East Antarctica. . .*

Can the comment be clarified?

*P16 L18: what is meant by 'regional external influences'? Influences other than regional temperature?*

495 "is complicated by influences other than regional temperature, including source temperature and ice sheet elevation change"

*P16 L20: be more precise about what these differences are and why you think they are significant and at what level (a Table comparing the results for all sites would help).*

500 Accepted, a spreadsheet is now included in the supplement.

*P16 L24: drop 'as is the investigation of the role CO2 in global temperature change.' It is repeated further down.*

Accepted
505

*Supplement: The note on Gaussian uncertainties should be moved to the main text. The note on assessing significance is central to the results and should also be integrated into the main text (and see major comments above).*

These points have now been changed, in accordance with reviewer one's comments.

[revised manuscript text omitted]

To conserve computational time, the individual temperature series are run with 10,000 initialization iterations and 100,000 optimization iterations, and the Savitsky-Golay filtered series with 1000 initialization iterations and 50,000 optimization iterations. Tests show that runs with 10,000 optimization iterations achieve reasonable convergence.

**Parallel MCMC methodology**

20  To propose updates to the walkers, we apply what Goodman and Weare (2010) refer to as a "stretch move". Consider the ensemble of walkers, in this case representing potential piecewise linear fits $\mathbf{X}$ and an individual walker $X_k^j$ in the ensemble at proposal step $j$. We select another walker $X_h^j$ from the complementary ensemble $\mathbf{X}_{[k]}^j$, composed of all of the other walkers. Then, a proposal is made to update $\boldsymbol{X}_k$ to $W$:

$$X_k^j \rightarrow W = \boldsymbol{X}_h^j + Z\left(\boldsymbol{X}_k^j - \boldsymbol{X}_h^j\right) \tag{1}$$

[Figure]

**Figure 1.** Schematic of the calculation of residuals between a data series and its change point representation. Here, the data series is shown as black points, with $1\sigma$ error bars. Change points are shown in pink, and the interpolations between them as black lines. The differences between the data points and the interpolations are shown as red arrows; the residual vector is composed of this distance divided by the uncertainty $\sigma$ at each point.

GW define the following probability distribution to generate stochastic variable $Z$:

$$g(Z) \propto \left\{ \frac{1}{Z} \ if \ Z \in \left[\frac{1}{a_Z}, a_Z\right]; \ 0 \ otherwise \right. \tag{2}$$

where $a_Z$ is a user-defined constant. Proposals are accepted or rejected with acceptance probability

$$P_{X_k^j \rightarrow X_k^{j+1}=W} = \min\left\{1, \ Z^{j-1}\frac{exp(-J(W)))}{exp(-J(X_k^j)))}\right\}. \tag{3}$$

**5 BIC and change point sensitivity tests**

We calculate the BIC for both series individually, and an additive, normalized BIC to pick the number of change points used. The additive BIC gives 7 as the best number of points to coherently fit both series. Using the individual criteria, 6 points would have been chosen for temperature, and 9 for $CO_2$, but comparing different numbers of points makes it less clear if the changes reoresented are actually coherent in timescale. These are shown in Figure 2.

10  We run tests to test the sensitivity of our results to the use of 6,7,8,9 and 10 change points rather than 7 (Figure 3). It is clear from these figures that 5-point fits cannot accurately represent either series; at 6 points, the temperature series begins to be

[Figure]

**Figure 2.** Individual (left, $CO_2$ in black and ATS3 in red) and cumulative BIC (right) for 5,6,7,8,9 and 10 point (x-axis) runs.

[Figure]

**Figure 3.** Atmospheric $CO_2$ (black) and ATS3 histograms of probable change points from the 5,6,7, 8 and 9 (from top left to bottom right) runs of LinearFit.

well-represented, and at 7 points the $CO_2$ series is appropriately represented. The 7, 8 and 9 point fits do not show significant differences, except at the ACR onset.

**Test of AR(1) residuals**

We test our AR(1) model by plotting adjacent residuals of the $CO_2$ series against their predictions, using the robust AR1 model

5   adapted to unevenly spaced time series described in the methods section. The results are shown in figure 4.

[Figure]

**Figure 4.** Plot of residuals $r_i$ (x axis) against the predictions made using our AR(1) model $r_i = r_{i-1} \cdot a^{t_i - t_{i-1}}$ (y axis) shown as blue dots, for the $CO_2$ series after the initialization MCMC procedure. The orange line represents a perfect model fit.

**A note on Gaussian Estimates**

We avoid providing estimates of change point timings in Gaussian form, i.e. as a mean $\pm$ a standard deviation. This is because the change point histograms we calculate are often skewed, and sometimes even multimodal, with multiple, separate probability peaks in a given time period where change is likely. These peaks can be due to sub-millenial scale variations at these points in the two series.

**A test with known lags and covariance**

.

We use artificially generated series to test the capacity of our method to fit change points in series with four known change points (plus two end points). For each series, a covariance matrix is used to generate noise with a red component, with a correlation coefficient of 0.8 at 50 years and 0.64 at 100 years away from the central point, and a uniform white component. The noise is scaled to 10% of the standard deviation of the change point values. The results of this test, along with the original change points, are shown in Figure 5.

In Figure 5, note that all of the change points are correctly identified in the histograms, with uncertainties on the order of 100 years. However, a small, but incorrect probability peak is generated around 13 ka. We conclude that gradual changes, such as that around 12 ka, are slightly more difficult to extract from correlated noise even when this noise is modeled, adding some methodological uncertainty which is reflected in the histograms.

[Figure]

**Figure 5.** Artificially generated series and generated change point distributions.

---

## Author Response (AR4)

Dear editor,

The final round of revisions are included in this marked-up file. The references to "significance" were left in from a previous version of the manuscript in which we set a probability threshold, which we no longer use – we have removed them except for two cases which do not refer to the histograms. We send our sincere thanks for your continued, detailed attention to this manuscript.

On behalf of all co-authors, all the best,

Jai Chowdhry Beeman

[revised manuscript text omitted]